EMBO
Molecular Medicine

# Modulating phosphatase DUSP22 with BML-260 ameliorates skeletal muscle wasting via Akt independent JNK-FOXO3a repression

Sang-Hoon Lee[1], Hyun-Jun Kim[1], Seon-Wook Kim[1], Hyunju Lee [iD][2], Da-Woon Jung [iD][1,✉] & Darren Reece Williams [iD][1,✉]

## Abstract

**Skeletal muscle wasting results from numerous conditions, such as sarcopenia, glucocorticoid therapy or intensive care. It prevents independent living in the elderly, predisposes to secondary diseases, and ultimately reduces lifespan. There is no approved drug therapy and the major causative mechanisms are not fully understood. Dual specificity phosphatase 22 (DUSP22) is a pleiotropic signaling molecule that plays important roles in immunity and cancer. However, the role of DUSP22 in skeletal muscle wasting is unknown. In this study, DUSP22 was found to be upregulated in sarcopenia patients and models of skeletal muscle wasting. DUSP22 knockdown or treatment with BML-260 (a small molecule previously reported to target DUSP22) prevented multiple forms of muscle wasting. Mechanistically, targeting DUSP22 suppressed FOXO3a, a master regulator of skeletal muscle wasting, via downregulation of the stress-activated kinase JNK, which occurred independently of aberrant Akt activation. DUSP22 targeting was also effective in human skeletal muscle cells undergoing atrophy. In conclusion, phosphatase DUSP22 is a novel target for preventing skeletal muscle wasting and BML-260 treatment is therapeutically effective. The DUSP22-JNK-FOXO3a axis could be exploited to treat sarcopenia or related aging disorders.**

**Keywords** Skeletal Muscle Wasting; DUSP22; Myofiber Atrophy; FOXO3a; BML-260
**Subject Category** Musculoskeletal System

## Introduction

Skeletal muscle wasting is a complex multifactorial disorder. A major type of muscle wasting is the loss of mass and strength due to aging, which was termed sarcopenia ("flesh" and "poverty" in Greek) by Irwin Rosenberg in 1989 (Ding et al, 2018). It is estimated that skeletal muscle loses around 1% mass and 3% strength annually from middle age, leading to an overall mass reduction that can reach 50% by the 8–9th decade of life (Oikawa et al, 2019; Wilkinson et al, 2018). The decreased physical function and increased frailty progressively leads to impaired mobility, loss of independence, and increased mortality. Due to demographic aging, sarcopenia has become a major health issue with an economic burden estimated at $40.4 billion per year in the United States alone (Deane et al, 2021; Dos Santos et al, 2017; Goates et al, 2019; Xu et al, 2022).

Apart from aging, many other disorders can produce skeletal muscle wasting. Some examples include side effects from commonly prescribed drugs (e.g., glucocorticoid therapy), infections (such as HIV-AIDS), various degenerative diseases (including heart failure, chronic kidney disease and diabetes), stroke, burns, or intensive care (also known as ICU-acquired weakness) (Ding et al, 2018). Currently, there are no clinically approved drugs for treating skeletal muscle wasting and the causative molecular mechanisms are not completely understood (Sartori et al, 2021).

Dual-specificity phosphatases (DUSPs) are a family of signaling enzymes that dephosphorylate serine/threonine and tyrosine residues (Li et al, 2022). Numerous protein and non-protein substrates, such as lipids and glucans, are modified by DUSPs (Thompson and Stoker, 2021; Zandi et al, 2022). DUSP22 (also known as JSP1-1 or JKAP) was recently linked to aging processes in kidney tissue (Oh et al, 2021). DUSP22 is also known to be expressed in skeletal muscle. A major target of DUSP22 is the stress-activated c-Jun N-terminal kinase (JNK) signaling cascade, which is associated with a number of diseases, such as cancer, dementia and autoimmunity (An et al, 2021; Chen et al, 2002; Gao et al, 2021; Li et al, 2022; Li et al, 2014). The JNK pathway has also been linked to some types of muscle wasting (Mulder et al, 2020). JNK signaling is increased in cancer-associated skeletal muscle loss and upregulated the atrogenes atrogin-1 (Fbxo32) and MuRF-1 (Trim63) (Mulder et al, 2020). JNK signaling has been shown to induce the activation of FOXO3a, which is a master regulator of skeletal muscle wasting and transcriptional activator of MuRF-1 and atrogin-1 (Chaanine et al, 2012; Chi et al, 2022; Nho and Hergert, 2014; Wang et al, 2012). Thus, JNK could be a drug target for this disorder. Unfortunately, despite much research progress, no JNK inhibitor compound has been approved for clinical use

[1]New Drug Targets Laboratory, Department of Life Sciences, College of Life Sciences and Medical Engineering, Gwangju Institute of Science and Technology, Gwangju 61005, Republic of Korea. [2]AI Graduate School, Gwangju Institute of Science and Technology, Gwangju 61005, Republic of Korea. ✉E-mail: jung@gist.ac.kr; darren@gist.ac.kr

(Zhang et al, 2012). FOXO3a signaling can also be downregulated via activation of the important signaling molecule Akt (protein kinase B), although Akt activating compounds could increase the incidence of aging-related pathologies and may be unsuitable for therapeutic applications (Campins et al, 2017; Chen et al, 2019; Nojima et al, 2013). Therefore, discovering novel signaling molecules and targets that suppress FOXO3a could facilitate therapeutic development for skeletal muscle wasting. However, the potential role of DUSP22-JNK signaling in the pathogenesis of skeletal muscle wasting is unknown.

In this study, DUSP22 expression levels were analyzed in sarcopenia patient samples and experimental models of skeletal muscle wasting. The effect of DUSP22 overexpression, knockdown, and pharmacological targeting with BML-260 were assessed in cell-based models of muscle atrophy. Skeletal muscle knockdown of DUSP22 and pharmacological inhibition was then investigated in three models of muscle wasting. Mechanism of action was elucidated by gene expression analysis of atrophy-related genes and whole genome transcriptome sequencing. Applicability to humans was validated using donor-derived skeletal muscle cells undergoing atrophy.

# Results

## DUSP22 is upregulated in skeletal muscle wasting and overexpression disrupts myogenesis

The expression of DUSP22 in humans with skeletal muscle wasting was investigated using a sarcopenia gene expression database of human muscle biopsies from elderly individuals over 70 years old and across ethnicities (Singapore Sarcopenia Study; (GEO accession no. GSE111016 (Migliavacca et al, 2019)). DUSP22 expression was found to be upregulated in sarcopenia patients (Fig. 1A). The dexamethasone (Dex) treatment model of myotube atrophy was used to study the effect of DUSP22 on muscle wasting in vitro, as previously described (Lee et al, 2021). DUSP22 expression was upregulated in myotubes undergoing Dex-induced atrophy, as assessed by RNA Seq (Fig. 1B) and qPCR (Fig. 1C). Atrogin-1 expression was also upregulated in myotubes undergoing Dex-induced atrophy, as assessed by RNA Seq (Appendix Fig. S1). DUSP22 expression was also investigated in the tibialis anterior (TA) muscle of three models of skeletal muscle wasting: Dex treatment, aged mice (27 months-old), and immobilization (Fig. 1C). The mean DUSP22 expression level was found to be upregulated in all three models, and reached statistical significance in the aging and immobilization models.

DUSP22 was overexpressed in C2C12 myoblasts using the endogenous CRISPR-cas9 editing tool and confirmed by qPCR (Fig. 1D). Overexpression disrupted myotube formation and significantly reduced the fusion and differentiation indexes (Fig. 1E–G). Expression analysis was carried out for genes linked to mitochondrial function, autophagy, ubiquitin-proteasome system (UPS), myosin heavy chain (MHC) isoforms, and FOXO3a signaling. DUSP22 overexpression downregulated peroxisome proliferator-activated receptor gamma coactivator 1-alpha (PGC-1α), a master regulator of mitochondrial biogenesis, and upregulated the mitochondrial uncoupling protein UCP-3, although acyl-CoA synthetase long chain family member 1 (Acyl), which

regulates mitochondrial fatty acid metabolism, was unaffected (Fig. 1H). The autophagy genes microtubule-associated proteins 1A/1B light chain 3B (LC-3B) and cathepsin L1 (Ctsl) were both upregulated by DUSP22 overexpression (Fig. 1I). The UPS gene ubiquitin protein ligase E3 component N-recognin 2 (UBR2) was upregulated by DUSP22 overexpression, while proteasome 26S subunit, non-ATPase 11 (Psmd11) expression was unaffected (Fig. 1J). The MHC isoforms slow myosin (MYH7) and fast myosin (MYH1) were downregulated by DUSP22 overexpression (Fig. 1K). 10 genes associated with FOXO3a signaling were analyzed (MuRF-1, atrogin-1, FOXO3a, sequestosome 1 (p62), TGFB induced factor homeobox 1 (TGIF), activating transcription factor 4 (ATF4), BCL2/adenovirus E1B 19 kDa protein-interacting protein 3 (Bnip3), growth arrest and DNA damage inducible alpha (Gadd45a), specific of muscle atrophy and regulated by transcription (SMART), and muscle ubiquitin ligase of SCF complex in atrophy-1 (MUSA1)). Seven genes were found to be upregulated by DUSP22 overexpression (MuRF-1, atrogin-1, FOXO3a, p62, TGIF, Gadd45a, and SMART) (Fig. 1L). Western blot analysis also confirmed the upregulation of UBR2, MuRF-1, and atrogin-1 by DUSP22 overexpression (Appendix Fig. S2).

## DUSP22 expression correlates with atrogene expression and knockdown prevents myotube atrophy

The potential relationship between DUSP22 and atrogene expression levels was investigated using Correlation AnalyzeR [29] for expression array profiling of a database obtained from muscle biopsies. The E3 ubiquitin ligase atrogenes MuRF-1 (TRIM63) and atrogin-1 (FBXO32, MAFbx) were selected because they are commonly measured in studies of skeletal muscle wasting and are transcriptionally activated by FOXO3a [30], whereas UBR2 was included as a potential target of DUSP22. Expression profiling confirmed a significant positive correlation between the expression of DUSP22 and MuRF-1 or atrogin-1, whereas UBR2 expression showed no significant change (Fig. 2A,B).

siRNA was used to investigate the effect of DUSP22 knockdown on myotube atrophy and atrogene expression. Gene knockdown in the myotubes was confirmed by qPCR (Fig. 2C). Knockdown prevented atrophy in the Dex model, as shown by increased myotube diameter and a greater proportion of larger-sized myotubes (Fig. 2D–F). Knockdown also increased the myotube fusion and differentiation indexes (Fig. 2G,H).

## DUSP22 knockdown lowers atrogene expression, downregulates JNK, and enhances myogenesis

The activation of FOXO3a and Akt after DUSP22 knockdown was assessed by western blotting. Knockdown in normal and Dex-treated myotubes decreased FOXO3a protein levels, leading to an increase in the overall ratio of phosphorylated FOXO3a: total FOXO3a (Fig. 3A,B). Knockdown had no significant effect on Akt phosphorylation (Fig. 3A–C). qPCR and western blotting showed that DUSP22 knockdown reduced atrogin-1 and MuRF-1 levels in Dex-treated myotubes (Fig. 3D–F). A second DUSP22 siRNA oligo was used for gene knockdown validation, as shown by reduced DUSP22 and atrogin-1 expression (Appendix Fig. S3). FOXO3a gene knockdown alone was also sufficient to reduce atrogin-1 and MuRF-1 levels (Appendix Fig. S4).

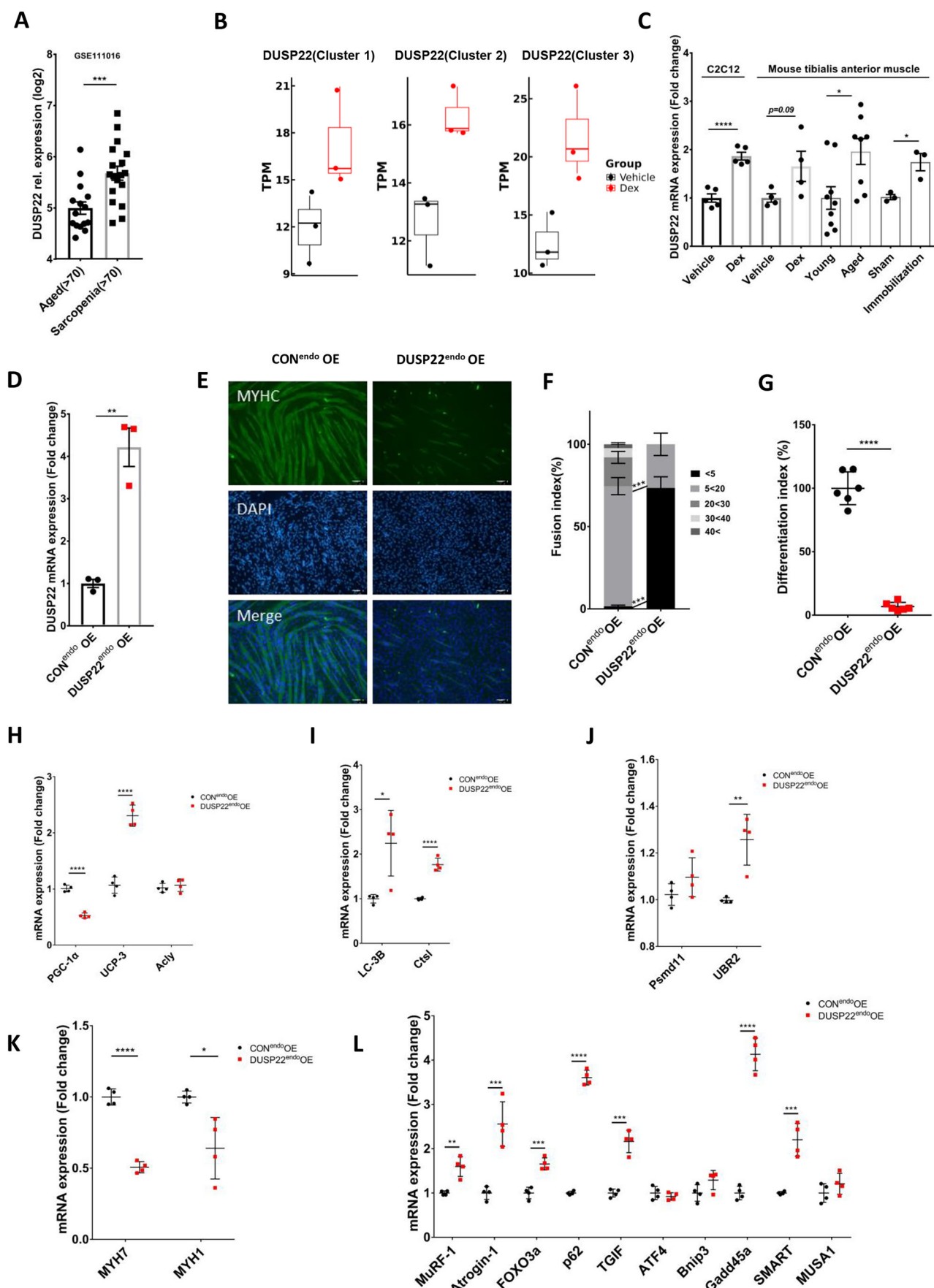

◄

**Figure 1.  DUSP22 is upregulated in skeletal muscle wasting and overexpression disrupts myogenesis.**

(A) DUSP22 expression in aged individuals (over 70 years, $n = 15$) and aged individuals diagnosed with sarcopenia (over 70 years, $n = 18$), $p = 0.0009$ (obtained from the Singapore Sarcopenia Study; (GEO accession no. GSE111016 (Migliavacca et al, 2019)). (B) DUSP22 expression in C2C12 murine myotubes treated with vehicle or dexamethasone (Dex) to induce atrophy ($n = 3$ each) Custer 1 ($p = 0.024$), Cluster 2 ($p = 0.13$), Cluster 3 ($p = 0.0246$). Expression was measured using RNA Seq. TPM=transcript per million. (C) qPCR analysis of DUSP22 expression in four models of muscle atrophy: (1) C2C12 myotubes treated with Dex ($n = 5$), $p = 7.67E{-}05$, (2) the TA muscle of C57BL/6 mice treated with Dex ($n = 4$), $p = 0.09$, (3) the TA muscle of young (5 months-old, $n = 9$) and geriatric (27 months-old, $n = 8$) C57BL/6 mice, $p = 0.0324$, (4) the TA of C57BL/6 mice after hind limb immobilization ($n = 3$), $p = 0.018$. (D) qPCR of DUSP22 expression in C2C12 myoblasts transfected with a DUSP22 CRISPR activation plasmid (DUSP22$^{endo}$OE) or control plasmid (CON$^{endo}$OE) ($n = 3$), $p = 0.0022$. (E) Fast myosin (MYH2) immunocytochemistry of CON$^{endo}$OE and DUSP22$^{endo}$OE myoblasts after 96 h culture in DM (scale bar = 100 μm). (F) Fusion index ($n = 6$). (G) Differentiation index ($n = 6$), $p = 9.91E{-}09$. (H–L) qPCR analysis of gene expression related to the following: (H) Mitochondrial homeostasis (PGC-1α (peroxisome proliferator-activated receptor gamma coactivator 1-alpha, $p = 1.44E{-}05$), UCP-3 (mitochondrial uncoupling protein 3, $p = 4.61E{-}05$), Acly (ATP citrate lyase, $p = 0.5141$)) ($n = 4$). (I) Autophagy (LC-3B (microtubule-associated proteins 1 A/1B light chain 3B, $p = 0.0152$), CtsL (cathepsin L, $p = 4.76E{-}05$)) ($n = 4$). (J) Ubiquitin-proteasome system (UPS) (UBR2 (ubiquitin protein ligase E3, $p = 0.0031$), Psmd11 (proteasome 26S subunit, non-ATPase 11, $p = 0.1718$) ($n = 4$). (K) Myosin heavy chain levels (slow myosin MYH7, $p = 7.23E{-}06$, fast myosin MYH1, $p = 0.0171$) ($n = 4$), and (L) FoxO3a-related signaling (FoxO3a ($p = 0.0004$), MurF-1 ($p = 0.0019$), atrogin-1 ($p = 0.001$), p62 ($p = 8.67E{-}08$), TGIF (TGFB induced factor homeobox 1, $p = 0.0001$), ATF4 (activating transcription factor 4, $p = 0.3974$), Bnip3 (BCL2/adenovirus E1B 19 kDa protein-interacting protein 3, $p = 0.0887$); Gadd45a (growth arrest and DNA damage inducible alpha, $p = 4.19E{-}06$), SMART (specific of muscle atrophy and regulated by transcription, $p = 0.0006$), MUSA1 (muscle ubiquitin ligase of SCF complex in atrophy-1, $p = 0.2357$)) ($n = 4$). Box plots represent the distribution of DUSP22 expression levels. The center line indicates the median (50th percentile, Q2), representing the middle value of the dataset. The box bounds correspond to the interquartile range (IQR), extending from the 25th percentile (Q1, lower bound) to the 75th percentile (Q3, upper bound). Whiskers extend to the smallest and largest values within $1.5 \times$ IQR from Q1 and Q3, representing the minimum (lower whisker) and maximum (upper whisker) values within this range. Data points that fall beyond this range are considered outliers and are displayed as individual points outside the whiskers. *$p < 0.05$, **$p < 0.01$, ***$p < 0.001$, and ****$p < 0.0001$ indicate significantly increased or decreased. n represents biological replicates. Error bars represent the standard error of the mean (SEM). Source data are available online for this figure.

The effects of DUSP22 knockdown was further investigated in differentiating myoblasts under normal conditions. Knockdown increased the expression of myosin heavy chain in the myotubes (Fig. 3G). qPCR analysis showed that knockdown increased expression of the myosin heavy chains slow myosin (MYH7), fast myosin (MYH2), fast myosin (MYH1), and fast myosin (MYH4) (Fig. 3H). In addition, DUSP22 knockdown increased the expression of myogenin (MyoG), a master inducer of myogenesis, and decreased the expression of FOXO3a (Fig. 3H). Western blotting indicated that DUSP22 knockdown increased fast myosin (MYH2) mean expression, although it did not reach statistical significance (Fig. 3I,J). Western blotting also confirmed that DUSP22 downregulates the JNK pathway in muscle cells. Knockdown decreased JNK levels and the phosphorylation and expression of the downstream JNK pathway mediator, c-jun (Fig. 3I,J). As previously observed in Fig. 3E, DUSP22 knockdown also reduced atrogin-1 and MuRF-1 expression levels in normal myotubes (Fig. 3I,J).

## DUSP22 pharmacological targeting prevents myotube atrophy

BML-260 is a rhodanine-based small molecule previously reported to target DUSP22 (Cutshall et al, 2005). BML-260 was originally characterized by Cutshall et al, from a screen of rhodanine derivatives for DUSP22 inhibition using an epidermal growth factor receptor peptide P$^{32}$-based assay (Cutshall et al, 2005). BML-260 was found to be a competitive inhibitor of DUSP22 with an IC$_{50}$ in the low micromolar range. BML-260 specificity was demonstrated by showing no inhibitory effect against VH1-related (VHR) phosphatase, which is a related, atypical DUSP (Cutshall et al, 2005). Molecular docking analysis indicated that BML-260 non-covalently binds to the active site of human DUSP22 at residue Cys88 with a Vina score of -5.8 (Fig. 4A). A DUSP22 phosphatase activity assay was carried out to further validate inhibition by BML-260. It was observed that BML-260 dose dependently inhibited DUSP22 activity with an IC$_{50}$ of 54 μM

(Appendix Fig. S5A). Additional molecular docking analysis was also undertaken to investigate BML-260 binding to the active site of DUSP22. BML-260 was shown to bind the active site region of DUSP22, whereas DUSP22 with an active site mutation (at C88S) altered BML-260 binding. DUSP15, which is a DUSP member with the highest homology with DUSP22 (as assessed by sequence alignment (Huang and Tan, 2012)), showed BML-260 binding at the interface of the A and B chain, distinct from the predicted active site at position 85 (Appendix Fig. S5B).

C2C12 mouse myotubes were treated with Dex in the presence or absence of BML-260. Dex-treated myotubes underwent atrophy, as shown by a reduction in myotube diameter. BML-260 treatment prevented myotube atrophy and maintained the proportion of larger-sized myotubes (Fig. 4B–D). BML-260 also recovered the myotube fusion and differentiation indexes (Fig. 4E,F). In addition, BML-260 prevented the reduction in protein synthesis caused by Dex (Fig. 4G,H). qPCR analysis showed that atrogin-1 and MuRF-1 were upregulated by Dex treatment and downregulated by BML-260 (Fig. 4I). Western blotting confirmed that BML-260 reduced atrogin-1, MuRF-1 and DUSP22 levels in the Dex-treated myotubes (Fig. 4J–M). The role of JNK in mediating the effects of BML-260 was validated using the small molecule inhibitor, SP600125. JNK inhibitor and BML-260 treatment together produced similar effects on Dex-induced myotube atrophy compared to SP600125 or BML-260 alone (Appendix Fig. S6). Moreover, BML-260 treatment had no significant additional effect on Dex-induced atrogene expression in the presence of DUSP22 gene knockdown (Appendix Fig. S10).

## DUSP22 knockdown in skeletal muscle ameliorates wasting

DUSP22 siRNA was delivered to the TA muscle of Dex-treated mice (Fig. 5A). DUSP22 knockdown in the TA was confirmed by western blotting and qPCR (Fig. 5B,C). DUSP22 knockdown prevented TA muscle wasting (Fig. 5D). Histological analysis indicated that knockdown increased myofiber cross-sectional area (CSA) and the proportion of larger sized myofibers (Fig. 5E–G).

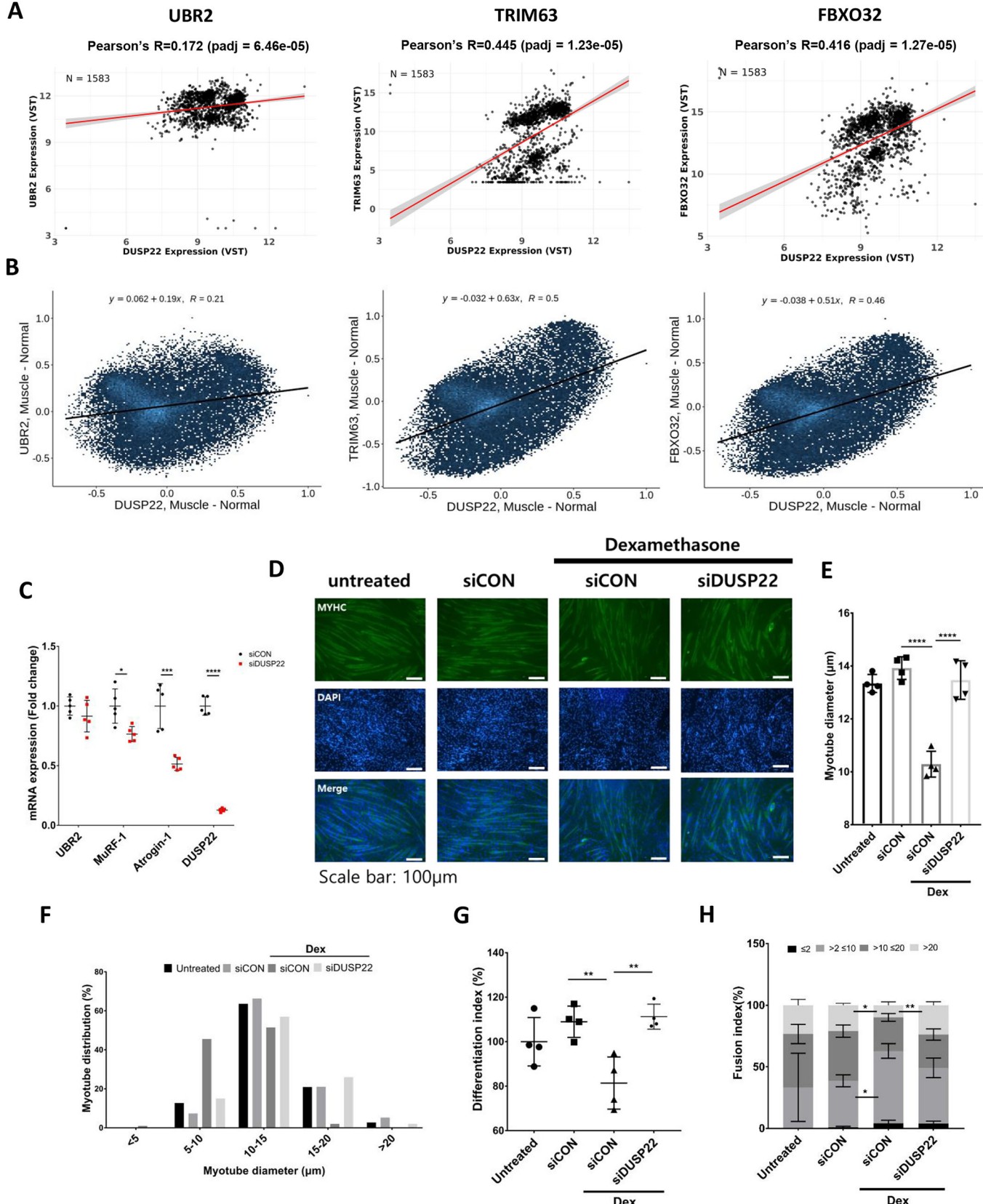

**Figure 2.  DUSP22 expression positively correlates with MuRF-1 and atrogin-1 expression and knockdown inhibits myotube atrophy.**

(A, B) Expression array profiling of DUSP22, MuRF-1, atrogin-1, and UBR2 expression levels (VST) in a database obtained from muscle biopsies, analyzed using Correlation AnalyzeR. (C) qPCR analysis of DUSP22 ($p = 7.22E-09$), UBR2 ($p = 0.25$), MuRF-1 ($p = 0.0101$), atrogin-1 ($p = 0.0005$) expression in C2C12 myotubes cultured treated with DM and control siRNA or DUSP22 siRNA for 48 h ($n = 5$). (D) Fast myosin (MYH2) immunocytochemistry of C2C12 myoblasts cultured as follows: (1) 120 h incubation with DM; (2) 72 h incubation with DM and 48 h incubation with DM plus control, DUSP22 siRNA; (3) Following 72 h incubation with DM, 24 h incubation in with DM plus control, scrambled siRNA and additional 24 h treatment with 10 μM Dex plus siRNA; (4) Following 72 h incubation with DM, 24 h incubation in with DM plus DUSP22 siRNA and additional 24 h treatment with 10 μM Dex plus siRNA (scale bar = 100 μm). (E) Myotube diameter ($n = 4$, $p=$ siCON (3.01E−05), siDUSP22+Dex (0.0003)). (F) Myotube distribution. (G) Differentiation index of C2C12 myoblasts treated as in part (D) ($n = 4$, $p =$ siCON (0.0068), siDUSP22+Dex (0.0037))). (H) Fusion index ($n = 4$). *$p < 0.05$, **$p < 0.01$, ***$p < 0.001$ and ****$p < 0.0001$ indicate significantly increased or decreased. n represents biological replicates. Error bars represent the standard error of the mean (SEM). Source data are available online for this figure.

Fast twitch type 2 myofibers are known to be more susceptible to sarcopenia (Livingstone et al, 1981). DUSP22 knockdown preserved the CSA of type 2a and type 2b myofibers (Fig. 5E,H,I). Western blotting showed that FOXO3a protein levels increased after Dex treatment and were reduced by DUSP22 knockdown, which increased the overall ratio of phosphorylated FOXO3a: total FOXO3a (Fig. 5J,K). DUSP22 knockdown also downregulated JNK and the downstream effector, c-jun (Fig. 5J–L). The upregulation of MuRF-1 by Dex was also inhibited by DUSP22 knockdown (Fig. 5M,N). A similar result was observed for p62, a key autophagy-related gene induced by FOXO3a (Moller et al, 2019) (Fig. 5M,N). Western blot analysis of AKT phosphorylation indicated that DUSP22 knockdown did not activate AKT signaling (Fig. EV1). To assess whether DUSP22 knockdown affects the strength of muscle contraction, the TA muscles of siRNA injected mice (24 months-old or 15 months-old) were analyzed using an 820MS muscle strip system. It was observed that DUSP22 knockdown produced an increase in tetanic and twitch force (Fig. EV2).

## DUSP22 pharmacological targeting ameliorates skeletal muscle wasting

The Dex treatment mouse model was used to investigate DUSP22 pharmacological targeting with BML-260 (Fig. 6A). Dex significantly reduced mean body weight, which was recovered by BML-260 treatment (Fig. 6B). BML-260 treatment increased the mean grip strength value, but it did not achieve statistical significance, possibly due to variation in the Dex alone group (Fig. 6C). Rotarod testing showed that BML-260 significantly recovered muscle performance in both the acceleration and constant models (Fig. 6D). Histological analysis of the TA muscle showed that BML-260 significantly increased myofiber CSA (Fig. 6E,F). BML-260 also shifted the myofiber area distribution towards larger sized fibers (Fig. 6G). TA muscle mass was increased by BML-260 treatment (Fig. 6H), and the upregulation of atrogin-1, MuRF-1 and DUSP22 was inhibited (Fig. 6I,J). Pharmacokinetic analysis of BML-260 in the plasma and skeletal muscle (TA) showed that BML-260 could be detected in the plasma, and then the TA muscle, after IP delivery (Fig. EV3).

## DUSP22 knockdown and pharmacological inhibition prevents aging-induced skeletal muscle wasting

DUSP22 siRNA was delivered to the TA muscle of geriatric mice (27 months-old) (Fig. 7A). DUSP22 knockdown in the TA was confirmed by western blotting (Fig. 7B,C). DUSP22 knockdown

also reduced the upregulation of MuRF-1 (Fig. 7B). TA muscle mass was increased in the DUSP22 knockdown group (Fig. 7D). Histological analysis revealed that DUSP22 knockdown increased TA myofiber CSA (Fig. 7E,F). Western blot analysis showed that atrogin-1, MuRF-1, and FOXO3a levels tended to be reduced in the siRNA treated TA muscle compared to the contralateral leg (7G,H).

Geriatric mice (24–26 months-old) were treated with BML-260 for 4 weeks (Fig. 7I). Body weight was not significantly affected by BML-260 treatment (Fig. 7J). The grip strength mean value was increased by BML-260 treatment but did not reach statistical significance (Fig. 7K). Latency to fall in the rotarod test was statistically increased by BML-260 treatment (Fig. 7K). BML-260 also increased the mean mass value of the quadriceps, gastrocnemius, TA and soleus muscles, and reached statistical significance for the TA and gastrocnemius muscles (Fig. 7L). qPCR analysis indicated that BML-260 reduced the expression of atrogin-1 and MuRF-1 in the TA muscle (Fig. 7M). BML-260 also reduced the expression of myostatin and increased the expression of PGC-1α, which are critical promoters and inhibitors of muscle aging, respectively (Fig. 7N). Western blot analysis showed that BML-260 treatment reduced DUSP22 levels in the TA muscle (Fig. EV4A). Immunostaining of the TA and gastrocnemius muscles showed that BML-260 increased the CSA and Feret's diameter of the fast twitch type 2 A, type 2B and 2X myofibers in the aged mice (Fig. 7O–Q). BML-260 treatment did not significantly affect the proportion of slow type 1 myofibers (Appendix Fig. S7).

DUSP22 knockout mice have been shown to develop inflammatory and autoimmune disorders (Chen et al, 2023; Li et al, 2014). To gain some insight into the potential toxicity of long-term BML-260 treatment, 15-month-old mice were treated with BML-260 for 6 weeks. Assessment of kidney, liver and heart appearance and mass indicated no major difference compared to vehicle-treated mice (Appendix Fig. S8A,B).

## DUSP22 pharmacological targeting in aged skeletal muscle regulates FOXO3a signaling, muscle cell differentiation and myofiber development

The mechanisms by which DUSP22 targeting prevents aging-induced skeletal muscle atrophy were characterized by whole genome RNA seq of the TA muscle in geriatric mice (27 months-old). Principal component analysis showed that pharmacological targeting with BML-260 produced similar overall effects on gene expression in the aged mice (Fig. 8A). The volcano plot showed a total of 350 differentially expressed genes (DEGs) (Fig. 8B). DUSP22 pharmacological targeting downregulated genes linked to FOXO3a signaling (including atrogin-1 and MuRF-1), as

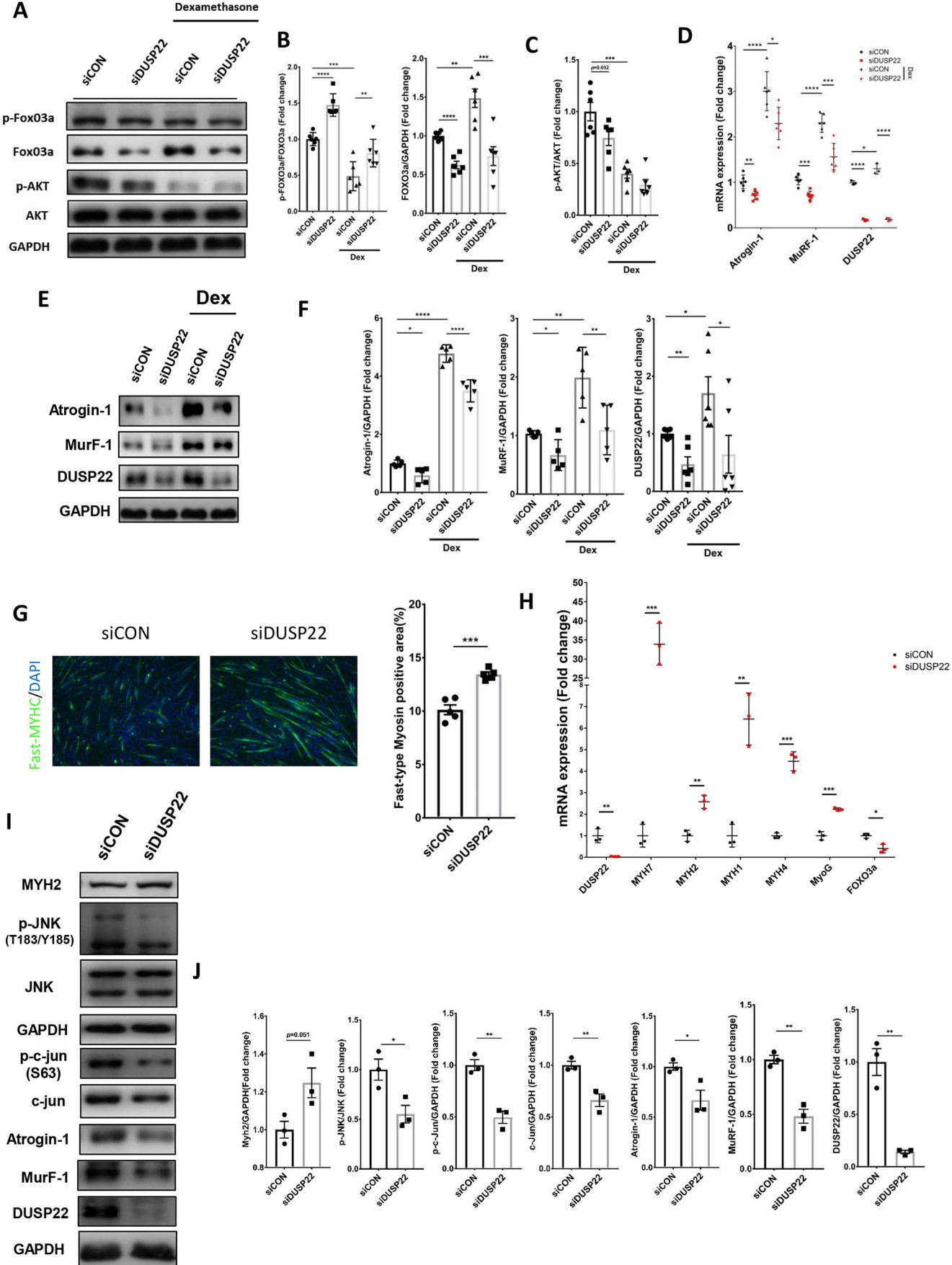

**Figure 3. DUSP22 knockdown inhibits atrogene expression and promotes myogenesis.**

(A) Western blot analysis of FoxO3a and Akt phosphorylation in C2C12 myotubes treated with control or DUSP22 siRNA in the presence or absence of Dex. (B) Quantification of FOXO3a phosphorylation, which is inversely proportional to activity ($n = 6$, p-FOXO3A/FOXO3A $p=$siDUSP22 (7.8E−05), siCON (0.0002), siDUSP22+Dex (0.018), FOXO3A $p=$ siDUSP22 (3.37E−05), siCON (0.0024), siDUSP22+Dex (0.0013)). (C) Quantification of Akt phosphorylation, which is directly proportional to activity ($n = 6$, p-AKT/AKT $p =$ siDUSP22 (7.8E−05), siCON (0.0002), siDUSP22+Dex (0.018)). (D) qPCR analysis of atrogin-1 ($p =$ siDUSP22 (0.0187), siCON (0.0002), siDUSP22+Dex (0.0451)), MuRF-1 ($p =$ siDUSP22 (0.0037), siCON (0.0001), siDUSP22+Dex (0.0044)), and DUSP22 ($p=$siDUSP22 (1.8E−05), siCON (0.0067), siDUSP22+Dex (2.91E−05)) expression ($n=6,3$). (E) Western blot of atrogin-1 and MuRF-1 expression levels. (F) Quantification of atrogin-1 ($p =$ siDUSP22 (0.0125), siCON (0.5.05E−09), siDUSP22+Dex (0.0003)), MuRF-1 ($p =$ siDUSP22 (0.0226), siCON (0.0033), siDUSP22+Dex (0.0098)), and DUSP22 ($p =$ siDUSP22 (0.0023), siCON (0.0356), siDUSP22+Dex (0.0353)) levels ($n = 5$). (G) Fast-type myosin immunostaining of C2C12 myoblasts after culture in DM for 24 h and treatment with control, scrambled siRNA or DUSP22 siRNA for 72 h. Quantification of the fast-type myosin positive myotubes is also shown ($n = 5$, $p = 0.0001$). (H) qPCR analysis of DUSP22 ($p = 0.0065$), myosin heavy chains (slow myosin MYH7 ($p = 0.0004$), fast myosin MYH2 ($p = 0.0024$), fast myosin MYH1 ($p = 0.002$), and fast myosin MYH4 ($p = 0.0002$)), myogenin (MyoG) ($p = 0.0005$) and FOXO3a ($p = 0.0171$) expression ($n = 3$). (I) Western blot analysis of MYH2 ($p = 0.015$), p-JNK ($p = 0.0316$), c-jun ($p = 0.0086$), c-jun phosphorylation ($p = 0.0031$), atrogin-1 ($p = 0.0333$), MuRF-1 ($p = 0.0021$), and DUSP22 ($p = 0.0026$) ($n = 3$). (J) Quantification of expression and phosphorylation levels. *$p < 0.05$, **$p < 0.01$, ***$p < 0.001$, and ****$p < 0.0001$ indicate significantly increased or decreased. n represents biological replicates. Error bars represent the standard error of the mean (SEM). Source data are available online for this figure.

assessed by overall TPM fold change (Fig. 8C) and normalized counts from the raw read (Fig. 8D). Interestingly, pharmacological targeting also upregulated the expression of musclin (OSTN), which is an exercise-induced myokine that protects against muscle wasting (Re Cecconi et al, 2019; Subbotina et al, 2015) (Fig. 8E). Gene ontology (GO) analysis showed that pharmacological targeting upregulated genes linked to muscle cell differentiation and muscle fiber development (Fig. 8F). In addition to musclin, pharmacological targeting increased the expression of fast myosin (MYH4), which is a predominant myosin type in fast-twitch TA muscle (Fig. 8F). The genes Six1 and Six4, which are known to activate the fast-type muscle gene program (Niro et al, 2010), were also upregulated (Fig. 8F). Genes linked to the PI3K-Akt pathway were also downregulated by pharmacological targeting (Appendix Fig. S9A). Gene set enrichment analysis (GSEA) combined with GO analysis indicated that skeletal muscle genes involved in the negative regulation of hypertrophy and response to inactivity were under-represented after DUSP22 targeting (Appendix Fig. S9B). In addition, genes involved in skeletal muscle cell differentiation and proliferation were enriched after targeting (Appendix Fig. S9C).

### DUSP22 pharmacological targeting downregulates atrogenes in immobilized muscle and ameliorates atrophy in human skeletal myotubes

A hind limb immobilization model was used to assess whether DUSP22 pharmacological targeting is effective for multiple types of skeletal muscle wasting. Immobilization decreased the mean mass of the quadriceps, gastrocnemius, TA and soleus muscles, and reached statistical significance for the gastrocnemius (Fig. 9A). Analysis of the TA muscle showed an upregulation of DUSP22 expression, which was inversely correlated with TA mass (Fig. 9B–D). BML-260 treatment increased the mean TA mass, although it did not reach statistical significance (Fig. 9E). Atrogin-1 and MuRF-1 expression was upregulated in the immobilized TA and downregulated by BML-260 treatment (Fig. 9F). Western blot analysis also showed that BML-260 treatment reduced DUSP22 levels in the TA and gastrocnemius muscles (Fig. EV4B). Analysis of the TA muscle also showed an upregulation of MYH4 (also known as myosin heavy chain 2B) after BML-260 treatment, along with a reduction in the phosphorylation level of the DUSP22 target, JNK (Figs. EV4C and EV5).

The clinical potential of DUSP22 targeting for skeletal muscle wasting was tested in cultures of human donor myotubes undergoing Dex-induced atrophy. BML-260 treatment recovered the mean myotube diameter (Fig. 9G,H). DUSP22 gene knockdown also prevented atrophy in human myotubes, as shown by a significant increase in myotube diameter (Fig. 9I,J). qPCR confirmed the knockdown of DUSP22 using siRNA (Fig. 9K). qPCR analysis also showed that DUSP22 gene knockdown inhibited MuRF-1 expression in the Dex-treated myotubes (Fig. 9L).

## Discussion

Skeletal muscle wasting has become major socioeconomic problem and there is currently no clinically approved drug for this disorder. In the current study, pharmacological and genetic targeting of the pleiotropic signaling phosphatase DUSP22 was shown to be effective at ameliorating skeletal muscle wasting in multiple models via the suppression of JNK and FOXO3a signaling.

DUSP22 pharmacological targeting was investigated with the previously reported DUSP22 targeting small molecule, BML-260. This compound is based on the rhodanine (2-thioxothiazolidin-4-one) chemical scaffold, which is becoming more prominent in medicine and pharmacy (Mousavi et al, 2019). Rhodanine is a privileged heterocyclic compound that is amenable for structural modification, and has been classified as a highly decorated scaffold with an important position for drug development (Chaurasyia et al, 2022). Consequently, BML-260 should be suitable for further rounds of structural optimization to improve its drug-like properties, biological stability, and targeting of DUSP22.

This study has shown that BML-260 or DUSP22 gene knockdown prevents skeletal muscle atrophy. Knockdown or BML-260 treatment also downregulated FOXO3a-related atrogene expression without stimulating Akt activity. BML-260 was originally identified in a screen of rhodanine-based compounds for DUSP22 inhibition (Cutshall et al, 2005). BML-260 inhibitory activity was shown to be specific for DUSP22, because there was no activity against the related atypical DUSP, VH1-related (VHR) phosphatase (Cutshall et al, 2005). However, other non-specific effects of BML-260 cannot be discounted. For example, Feng et al, reported that BML-260 treatment upregulates the expression of uncoupling protein-1 (UCP-1) in adipocytes (Feng et al, 2019). This effect was found to

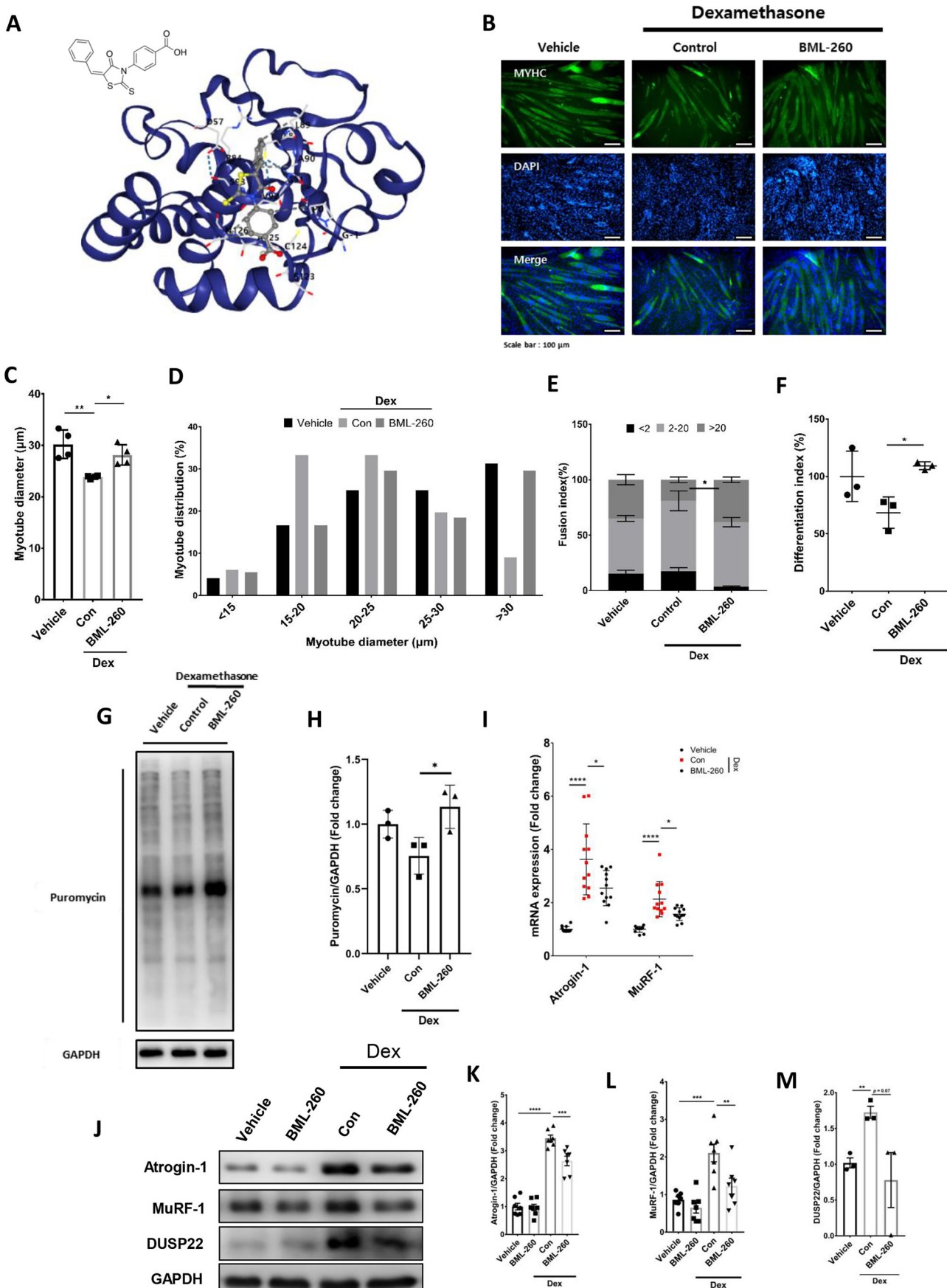

◄ **Figure 4. DUSP22 pharmacological targeting prevents myotube atrophy.**

(A) Chemical structure of BML-260 and CB-Dock2 modeling of BML-260 binding to the active site of human DUSP22 (Pocket C5 and score -5.8. Chain A: GLY-1, PRO0, ASP57, CYS88, LEU89, ALA90, GLY91, VAL92, SER93, ARG94, SER123, CYS124, ALA125, ASN126, ASN128). (B) Fast myosin (MYH2) immunostaning of C2C12 myoblasts cultured as follows: (1) DM for 120 h (untreated); (2) DM for 96 h and DM plus 10 µM Dex for 24 h; (3) DM for 96 h and DM plus 10 µM Dex and 12.5 µM BML-260 for 24 h (scale bar = 100 µm). (C) Mean myotube diameter ($n = 4$, $p =$ (Vehicle = 0.0038), (BML-260 = 0.0051)). (D) Myotube diameter distribution. (E) Fusion index ($n = 3$). (F) Differentiation index ($n = 3$, $p =$ (Vehicle = 0.0731), (BML-260 = 0.0282)). (G) SUnSET assay of protein synthesis rate measuring puromycin incorporation ($n = 3$). (H) Quantification of the SUnSET assay ($p =$ (Vehicle = 0.0747), (BML-260 = 0.0401)). (I) qPCR analysis of atrogin-1 ($p =$ (Vehicle = 7.65E−07), (BML-260 = 0.0194)) and MuRF-1($p =$ (Vehicle = 7.01E−06), (BML-260 = 0.112)) expression ($n = 12$). (J) Western blot analysis of atrogin-1, MuRF-1, and DUSP22. (K–M) Quantification of atrogin-1 ($p =$ Vehicle (3.74E−09), Dex+BML-260 (0.0016)), MuRF-1 ($p =$ Vehicle (0.0002), Dex+BML-260 (0.0025)) ($n = 6$) and DUSP22 ($p =$ Vehicle (0.0029), Dex+BML-260 (0.0737)) ($n = 3$). *=$p < 0.05$, **=$p < 0.01$, and ****=$p < 0.0001$ indicate significantly increased or decreased. n represents biological replicates. Error bars represent the standard error of the mean (SEM). Source data are available online for this figure.

be independent of DUSP22 activity and the precise mechanism could not be fully elucidated. It should be noted that while UCP-1 upregulation can be beneficial for treating obesity via mitochondrial uncoupling, reduced ATP production and heat generation in adipocytes, this may not be the case for sarcopenia, which should benefit from more efficient mitochondrial function and ATP generation in skeletal muscle (Harper et al, 2021). In addition, increased physical activity has been shown to decrease UCP-1 expression (Brandao et al, 2019). It is also interesting to observe that in the present study DUSP22 overexpression in myotubes upregulated the expression of the uncoupling protein, UCP-3 (Fig. 1H). The pharmacokinetics data for BML-260 presented in this study also suggests that, although BML-260 is detectable in skeletal muscle after dosing, it may not reach levels where it can function as an enzymatic inhibitor (Fig. EV3). Possible explanations for this discrepancy include off target effects or the presence of DUSP22 in intracellular protein complexes that have a greater affinity for BML-260.

Rhodanine-based compounds have been described as problematic for medicinal chemistry due to non-specificity, aggregation and potential toxicity (Tomasic and Peterlin Masic, 2012). However, Epalrestat, a rhodanine-based drug for diabetic neuropathy, is safe, well-tolerated, and the only commercially available inhibitor of aldose reductase (Ramirez and Borja, 2008). More recently, Montaño et al, demonstrated that rhodanine compounds could be developed as antibiotics that specifically target bacterial thymidylate kinase (Montano et al, 2021). Progress in the medicinal chemistry of rhodanine derivative scaffolds also suggest that these could be viewed as privileged structures that may be exploited for future rational design and discovery (Chaurasyia et al, 2022). Moreover, Mendgen et al, assessed the biological activity of 163 rhodanine or rhodanine-like compounds and found that only 2 compounds displayed potential aggregation, with the α-carbon attachment decoration being important in conferring selectivity and reducing promiscuous binding (Jones, 2017; Mendgen et al, 2012). Therefore, with a cautious approach to structural modifications and biological assays for activity, it may be possible to further develop BML-260 as a clinically relevant DUSP22 inhibitor. Further development may be significant for BML-260, which does not appear to have undergone any additional structural optimization since the initial publication by Cutshall et al (Cutshall et al, 2005). Consequently, pleiotropic off target effects that protect against muscle wasting cannot be ruled out when using this early stage inhibitor. Indeed, some of the effects of BML-260 treatment may overlap with the DUSP22-JNK-FOXO3a pathway to produce epistatic results.

The relationship between JNK activation and FOXO3a-mediated signaling has previously been reported. In addition, the effect of DUSP22 activation on JNK signaling is already known. The novel aspect of this current study is the demonstration that DUSP22 targeting can downregulate FOXO3a in skeletal muscle and produce beneficial effects on muscle atrophy. As major signaling molecules in cells, both JNK and FOXO3a activity can be regulated by numerous factors (reviewed in (Nho and Hergert, 2014; Zeke et al, 2016)). The results herein demonstrate that targeting DUSP22 alone in skeletal muscle is sufficient to downregulate FOXO3a and produce therapeutically desirable effects in myofibers.

It was observed that DUSP22 overexpression increased the expression of UBR2 in murine myotubes. Interestingly, a previous study reported that DUSP22 downregulates UBR2 proteins in human T lymphocytes (Shih et al, 2024). There are a number of possible explanations for the difference in these research findings. For example, the MAPK pathway has been show to maintain UBR2 expression, and DUSP22 is known to activate MAPK signaling (Ju et al, 2016; Villa et al, 2020). MAPK signaling status was not assessed in the human T lymphocytes, meaning the effect of DUSP22 activation on this pathway and its possible role in UBR2 expression in lymphocytes could not be ascertained. In skeletal muscle tissue, UBR2 upregulation has been previously linked to muscle atrophy and UBR2 expression was shown to be increased by FOXO3a in cancer-related muscle wasting (Hockerman et al, 2014; Judge et al, 2014). The results presented herein show that DUSP22 overexpression upregulated FOXO3a in muscle cells, which could explain the observed increase in UBR2 expression. Additionally, in skeletal muscle tissue additional signaling mechanisms, such as the N-end rule pathway, have been shown to maintain high UBR2 expression in conditions of muscle atrophy (where UBR2 functions as a substrate recognition component) (Gao et al, 2022; Kwak et al, 2004). It is possible that DUSP22 overexpression may also upregulate this pathway in muscle cells to increase UBR2 expression.

The results in this study show that DUSP22 expression is upregulated in Dex-treated mice. It is known that a number of DUSPs are upregulated by glucocorticoids (Clark et al, 2008). This regulatory mechanism has mainly been studied for DUSP1, but it is possible that similar mechanisms govern DUSP22 expression. Moreover, the upstream DNA enhancer sequence of the DUSP22 gene contains binding sites for the transcription factor Krueppel-like factor 9 (KLF9). Glucocorticoids have been shown to induce KLF9 expression in human epithelial cells (Mostafa et al, 2021). It may be possible that muscle cells also upregulate KLF9 expression

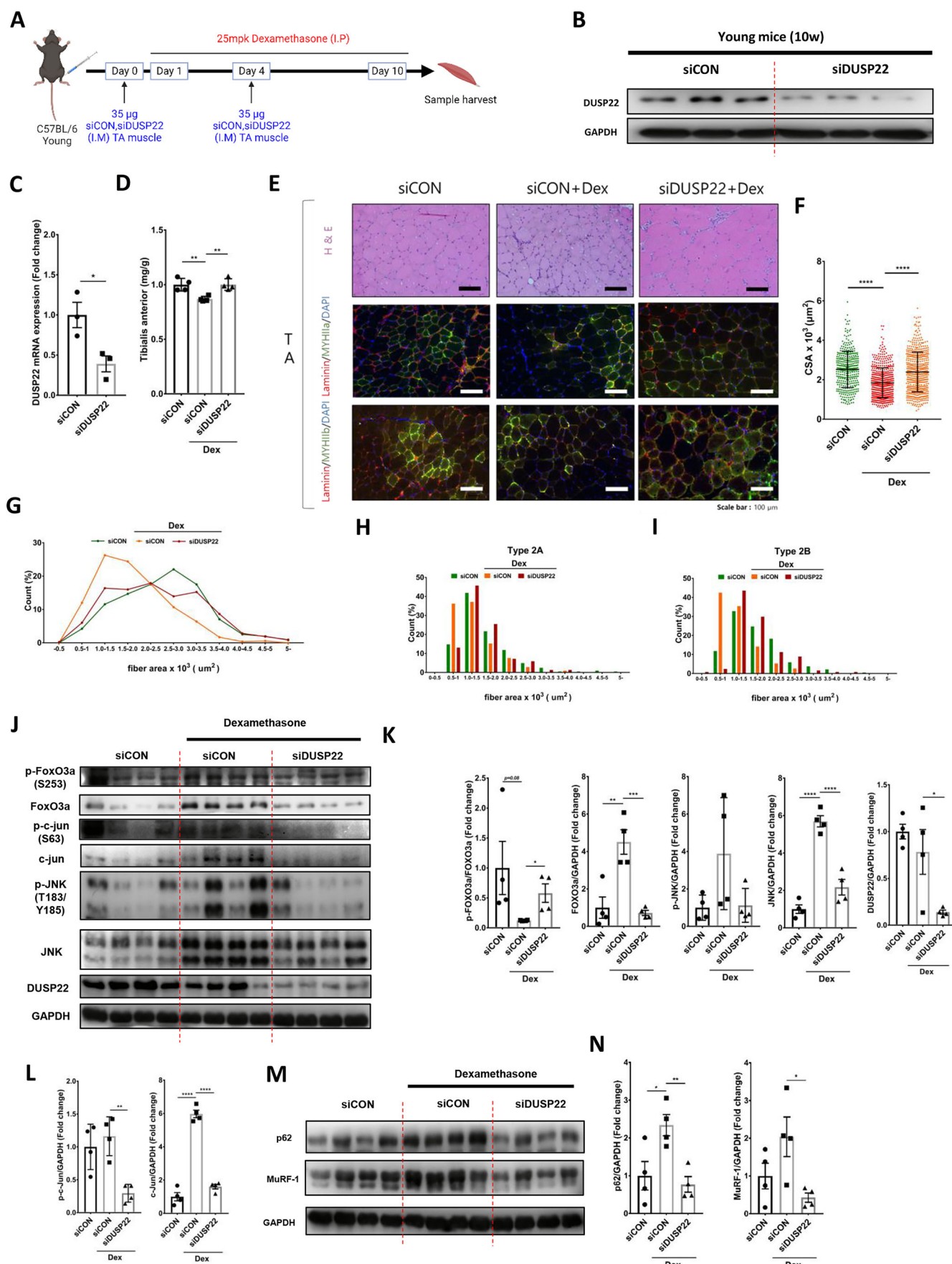

**Figure 5. DUSP22 knockdown prevents Dex-induced skeletal muscle atrophy.**

(A) Schematic of the experimental protocol. (B) Western blot analysis of DUSP22 in the Dex-treated tibialis anterior (TA) muscle 3 d after delivery of control or DUSP22 siRNA ($n = 3$). (C) Quantification of DUSP22 expression ($n = 3$, $p = 0.0308$). (D) TA muscle mass ($n = 4$, $p = $ (siCON = 0.006, siDUSP22+Dex = 0.0042)). (E) Representative H&E staining, and myosin heavy chain IIa (MYHIIa; type 2 A) and IIb (MYHIIb; type 2B) immunostaining of the TA muscle. (F) TA myofiber CSA ($n = 4$, $p = $ ((siCON = 1.34E−30, siDUSP22+Dex = 1.31E−22)). At least 80 fibers were measured for each sample (G) TA myofiber area distribution. (H) Type 2A myofiber distribution. (I) Type 2B myofiber distribution. (J) Western blot analysis of phosphorylated FOXO3a (p-FOXO3a), FOXO3a, phosphorylated c-jun (p-c-jun), c-jun, phosphorylated JNK (p-JNK), JNK in the TA muscle ($n = 4$). (K) Quantification of p-FOXO3a ($p = $ ((siCON = 0.0958, siDUSP22+Dex = 0.0255)), FOXO3a ($p = $ ((siCON = 0.0068, siDUSP22+Dex = 0.0.001)), P-JNK ($p = $ ((siCON = 0.1087, siDUSP22+Dex = 0.1271)), and JNK ($p = $ ((siCON = 1.61E−05, siDUSP22+Dex = 0.0004)). (L) Quantification of P-c-jun ($p = $ ((siCON = 0.5016, siDUSP22+Dex = 0.0017)) and c-jun ($p = $ ((siCON = 6.27E−06, siDUSP22+Dex = 3.52E−06)). GAPDH was used for the normalization of FOXO3a, p-JNK, JNK, p-c-Jun and c-Jun expression. FOXO3a was used for the normalization of p-FOXO3a expression. (M) Western blot analysis of p62 ($p = $ ((siCON = 0.0297, siDUSP22+Dex = 0.0042)) and MuRF-1 ($p = $ ((siCON = 0.1484, siDUSP22+Dex = 0.0245)) ($n = 4$). GAPDH was used for normalization of expression. (N) Quantification of p62 and MuRF-1. *$p < 0.05$, **$p < 0.01$, ***$p < 0.001$ and ****$p < 0.0001$ indicate significantly increased or decreased. n represents biological replicates. Error bars represent the standard error of the mean (SEM). Source data are available online for this figure.

in response to Dex treatment and KLF9 binds to the DUSP22 upstream enhancer to induce gene expression. Further experiments would be needed to validate this hypothesis, which may be a rewarding area for future investigation.

The effect of long-term administration of BML-260 in mice has not previously been reported. However, a previous study by Chen et al, observed that DUSP22 knockout mice spontaneously developed syndesmophytes (inflammatory growths in the spinal vertebrae and may produce joint fusion) (Chen et al, 2023). In addition, Li et al, reported that aged JKAP-knockout mice can spontaneously develop inflammation and autoimmunity, with higher serum levels of anti-nuclear antibodies and anti-dsDNA, and expansion of white pulps in the spleen (Li et al, 2014). Ideally, BML-260 should target DUSP22 without recapitulating the immune-related phenotype observed in knockout mice. It can be noted DUSP22 has additional functions in cells that are independent of its enzyme activity, such as a scaffold protein for JNK regulation by signaling proteins, including apoptosis signal-regulating kinase 1 (ASK1) (Ju et al, 2016). These non-enzymatic functions should still be preserved in the presence of BML-260 treatment. In this study, a general assessment of the effect of 6 weeks BML-260 treatment in 15-month-old mice on liver, heart and kidney mass (three major organs assessed for potential drug toxicity (Lin and Will, 2012)) indicated no gross significant change after the treatment period. However, further in-depth studies would be required to fully assess BML-260 toxicity during long-term treatment.

Pharmacological targeting or gene knockdown of DUSP22 inhibited signaling by JNK (also known as stress-activated kinase) and FOXO3a (a key regulator of skeletal muscle atrophy). DUSP22 is also known as JNK-stimulating phosphatase 1 (JSP1) and JNK activation has been shown to promote FOXO3a activation in numerous cell types (Chaanine et al, 2012; Chi et al, 2022; Nho and Hergert, 2014; Wang et al, 2012). To confirm the ability of DUSP22 to stimulate JNK in this study, we showed that gene knockdown in skeletal muscle downregulated JNK and the downstream mediator, c-jun. The IGF-1/PI3K/Akt pathway is also known to suppress FOXO3a activity in skeletal muscle and have a major role in the prevention of wasting (Chen et al, 2022). However, we consistently found that Akt activity was not upregulated by DUSP22 knockdown or BML-260 treatment. This may be advantageous for developing therapeutic approaches that target FOXO3a independently of the IGF-1-PI3K-Akt pathway due to potential side effects associated with PI3k-Akt activation (Nunnery and Mayer, 2019;

Wong et al, 2010). In addition, over-activation of protein synthesis, which is a downstream target of IGF-1-PI3K-Akt, has recently been observed in aging-related skeletal muscle wasting (Crombie et al, 2023; Fuqua et al, 2023). Strategies to further increase protein synthesis in geriatric muscle may worsen proteostatic stress (Fuqua et al, 2023). Thus, compounds such as BML-260, which inhibit FOXO3a signaling via alternative pathways, such as JNK, may be an attractive option for future drug development. A schematic of the effect of DUSP22 targeting in skeletal muscle wasting is shown in Fig. 9M.

An interesting aspect of the current study is the discovery of DUSP22 as an upstream regulator of the relationship between JNK and FOXO3a in skeletal muscle tissue, along with the availability of BML-260 as a novel small molecule inhibitor that prevents skeletal muscle atrophy. A number of small molecule JNK inhibitors have been developed and are under investigation for disorders such as stroke, cancer and Parkinson's disease (Cui et al, 2007; Wu et al, 2020). However, although there has been much research progress, no JNK targeting drug is approved for clinical use. DUSP22 is a relatively unexplored drug target and further investigation may produce lead compounds that can enter clinical trials as indirect JNK inhibitors for a number of disease applications. The in vivo RNA Seq data indicates that Six1 and Six4, genes linked to fast twitch myofiber formation (Niro et al, 2010), are upregulated by DUSP22 pharmacological targeting in aged muscle. This could explain the increased proportion of fast twitch type 2B and 2X myofibers. The mechanism by which DUSP22 targeting increases fast myofiber number in aged mice (preservation of pre-existing myofibers versus upregulation of transcriptional activators of fast fiber type gene expression) can be an interesting subject for future research. The discovery that musclin is upregulated after DUSP22 targeting may also warrant further investigation, because this myokine has been shown to be critical for cardiac conditioning (Harris et al, 2023). Cardiac hypertrophy and skeletal muscle wasting are known to share similar molecular mechanisms, and cardio-sarcopenia has recently been described as a syndrome of concern in aging (Loh et al, 2022; Rausch et al, 2021). In addition, skeletal muscle-secreted musclin has recently been shown to alleviate depression in animal models (Ataka et al, 2023; Szaroszyk et al, 2022). To our knowledge, there is no previous report of DUSP22 targeting in cardiac disease or depressive disorders. Whether musclin levels are directly regulated by DUSP22, or via downstream effects on JNK-FOXO3a inhibition, may also be an interesting topic for myokine research.

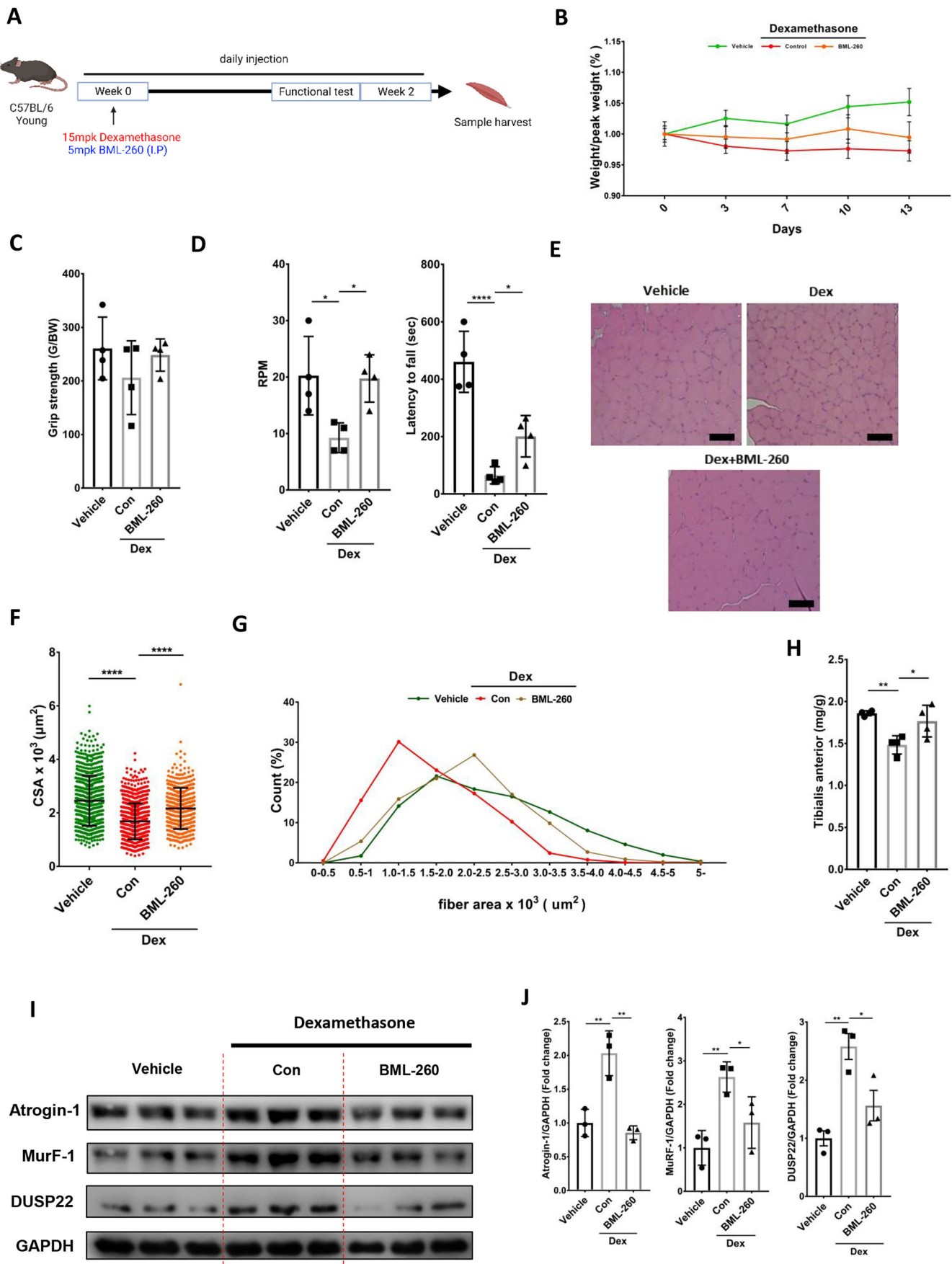

**Figure 6.  DUSP22 pharmacological targeting prevents Dex-induced skeletal muscle atrophy.**

(**A**) Schematic of the experimental protocol. (**B**) Body weight during 13 d treatment with vehicle, 15 mg/kg Dex, or 15 mg/kg Dex and 5 mg/kg BML-260. (**C**) Grip strength ($n = 4$, $p =$ (Vehicle $= 0.3426$, BML-260 $= 0.224$). (**D**) Rotarod performance in the constant (RPM) ($p =$ (Vehicle $= 0.05$, BML-260 $= 0.0263$)) and acceleration (latency to fall) ($p =$ (Vehicle $= 0.0003$, BML-260 $= 0.0132$)) models ($n = 4$). (**E**) Representative H&E staining of the gastrocnemius muscle. (**F**) Myofiber CSA ($n = 4$, $p =$ (Vehicle $= 1.98E{-}84$, BML-260 $= 2.93E{-}32$)). (**G**) Myofiber area distribution. At least 100 fibers were measured for each sample (**H**) TA muscle mass ($n = 4$, $p =$ (Vehicle $= 0.0042$, BML-260 $= 0.0208$)). (**I, J**) Western blot analysis of atrogin-1 ($p =$ (Vehicle $= 0.0029$, BML-260 $= 0.0014$)), MuRF-1 ($p =$ (Vehicle $= 0.0086$, BML-260 $= 0.0498$)), and DUSP22 ($p =$ (Vehicle $= 0.0034$, BML-260 $= 0.0258$)) expression in the TA muscle ($n = 3$). GAPDH was used for normalization of expression. *$p < 0.05$, **$p < 0.01$, and ****$p < 0.0001$ indicate significantly increased or decreased. n represents biological replicates. *n* represents biological replicates. Error bars represent the standard error of the mean (SEM). Source data are available online for this figure.

BML-260 was developed as an enzyme inhibitor of DUSP22 activity. In myotubes undergoing atrophy, BML-260 had no additional effect on atrogene downregulation compared to DUSP22 gene knockdown, suggesting that BML-260 activity is dependent on the presence of DUSP22. In addition, the observation of reduced levels of DUSP22 after BML-260 treatment in the skeletal muscle of Dex-treated, aged, and immobilized mice indicates that BML-260 also suppresses DUSP22 expression. The effect on DUSP22 levels, rather than reduced enzyme activity, may also be responsible for the therapeutic effects of BML-260 in muscle atrophy. This aspect of BML-260 bioactivity could be an interesting area for future investigation in other disease contexts linked to DUSP22 and JNK suppression.

In summary, this study demonstrates that DUSP22 is a regulator of skeletal muscle wasting and therapeutic target that functions by inhibiting a DUSP22-JNK-FOXO3a signaling axis. These results increase our understanding of the molecular mechanisms driving skeletal muscle wasting and provide a novel candidate for drug development. Previous research has linked aberrant DUSP22 to other diseases, such as T-cell lymphoma and Alzheimer's disease (Pedersen et al, 2017). The results presented herein may also be applicable to therapeutic approaches for these diseases. In addition, the identification of DUPS22 as a mediator of aging-related skeletal muscle wasting suggests that it may be worthwhile to investigate its role in other aging-related disorders, such as osteoarthritis, heart failure and type 2 diabetes. Overall, these findings provide new insight into the functions of DUSP22, the mechanisms responsible for producing skeletal muscle wasting, and novel treatment approaches for this debilitating condition.

## Methods

### Reagents and tools table

| Reagent/resource | Reference or source | Identifier or catalog number |
| --- | --- | --- |
| **Experimental models** | | |
| C57BL/6J (*M. musculus*) | DAMOOL SCIENCE | B6.129P2Gpr37tm1Dgen/J |
| C2C12 | ATCC | |
| Human adult skeletal primary cell | Gibco | A11440 |
| **Recombinant DNA** | | |
| DUSP22 CRISPR Activation Plasmid (m) | Santa Cruz | sc-430587-ACT |
| Control CRISPR Activation Plasmid | Santa Cruz | sc-437275 |

| Reagent/resource | Reference or source | Identifier or catalog number |
| --- | --- | --- |
| **Antibodies** | | |
| α-Tubulin | INVITROGEN | PA5-29444 |
| Atrogin-1/MAFbx | Abcam | Ab168372 |
| MuRF-1/Trim63 | Santa Cruz | SC-398608 |
| FoxO3a | Cell Signaling | #12829 |
| Phospho-FoxO3a | Cell Signaling | #9466 |
| AKT | Cell Signaling | #2983 |
| Phospho-AKT | Cell Signaling | #2971 |
| JNK | Cell Signaling | SC-7345 |
| Phospho-JNK | Cell Signaling | #4668 |
| c-Jun | Cell Signaling | #9165 |
| Phospho-c-Jun | Cell Signaling | #2361 |
| STAT3 | SANTA CRUZ | SC-8019 |
| Phospho-STAT3 | SANTA CRUZ | SC-8059 |
| UBR2 | Abcam | Ab217069 |
| Myosin heavy chain 2 | SANTA CRUZ | SC-53095 |
| DUSP22 | Abcam | Ab70124 |
| Myosin Heavy Chain type 1 | DSHB | BA-D5 |
| Myosin Heavy Chain type 2A | DSHB | SC-71 |
| Myosin Heavy Chain type 2B | DSHB | BF-F3 |
| Laminin | sigma | L9393 |
| Fast-type skeletal muscle | Santa Cruz | sc-32732 |
| Goat anti-Mouse IgG (H + L) | Abcam | ab6789 |
| Goat anti-Rabbit IgG HRP | Cell signaling | #7074S |
| Alexa Fluor™ 488 Goat anti-mouse IgG(H + L) | INVITROGEN | A11001 |
| Alexa Fluor™ 555 Goat anti-mouse IgG(H + L) | INVITROGEN | A21422 |
| Alexa Fluor™ 488 Goat anti-mouse IgG(H + L) | INVITROGEN | A28175 |
| Alexa Fluor™ 350 Goat anti-mouse IgG(H + L) | INVITROGEN | A11045 |
| **Oligonucleotides and other sequence-based reagents** | | |
| GAPDH primer | This study | F: CTC CAC TCA CGG CAA ATT CA<br>R: GCC TCA CCC CAT TTG ATG TT |
| Atrogin-1 primer | This study | F: CAG AGA GCT GCT CCG TCT CA<br>R: ACG TAT CCC CCG CAG TTT C |

| Reagent/resource | Reference or source | Identifier or catalog number |
|---|---|---|
| MuRF-1 primer | This study | F: CCG AGT GCA GAC GAT CAT CTC R: TGG AGG ATC AGA GCC TCG AT |
| Myh2 primer | This study | F: GAT CAC CAC GAA CCC ATA TGA TT R: TTC ATG TTC CCA TAA TGC ATC AC |
| Pax7 primer | This study | F: CAC AGA GGC AGA GCT GAT TGC R: CCA ATT GAG GAG AGT GAC AGG TT |
| Myf5 primer | This study | F: AGC TGG GCA GAA TAC GTG CTT R: AGA ACA GGC AGA GGA GAA TCC A |
| Myogenin primer | This study | F: AGC GCA GGC TCA AGA AAG TG R: CCG CCT CTG TAG CGG AGA T |
| MyoD1 primer | This study | F: TGT CCT TTC GAA GCC GTT CT R: TGC AGC CAG AGT GCA AGT G |
| DUSP22 primer | This study | F: GAT GCC TTG CAC ACT GTT CGT R: ACT GGT GCA CTT CAT GTT TCT CA |
| DUSP22 human primer | This study | F: GGT TTC TGT ACC TCG CTT GGA T R: AGG CGT TCA CAG AAA GCA A |
| Atrogin-1 human primer | This study | F: GGA ACT ACT CCA GAC CCT CTA CAC A R: CTC CAT CCG ATA CAC CCA CAT |
| MuRF-1 human primer | This study | F: TTG ACT TTG GGA CAG ATG AGG AA R: CCA GCT CCT TAC TGG TGT CCT T |
| GAPDH human primer | This study | F: CTG CAC CAC CAA CTG CTT AGC R: TCT TCT GGG TGG CAG TGA TG |
| PGC-1α primer | This study | F: CAG GGT GCA TGG CAG TTG T R: CAG AGG CCA TGC TAG TGA AAG A |
| UCP-3 primer | This study | F: GAT GTG GTG AAG GTC CGA TTT C R: CCC TGG CGA TGG TTC TGT AG |
| Acly primer | This study | F: ATG CCA AGA CCA TCC TCT CAC T R: GCG GCC ACA TTG TGT AAG |
| LC-3B primer | This study | F:CGT CCT GGA CAA GAC CAA GT R:ATT GCT GTC CCG AAT GTC TC |
| Cathepsin L primer | This study | F: CAG GGT CCG TGA GTG TGT CT R: CTG CAG TGC TGC CAG CTT T |
| Psmd11 primer | This study | F: TTT CTT ACG CCA AGC ATT GGA R: CTC CCG AAG CAG CTG AGA AC |
| UBR2 primer | This study | F: TAT TCT CCT CCT TAC CTT G R: CGA AAC CGC TCT TGG CAT A |
| MYH7 primer | This study | F: TGT TTT GTG CCC CGA TGA R: CAG TCA CCG TCT TGC CAT T |
| MYH1 primer | This study | F: CGG GAG AAC CAG TCT ATT TTG ATC R: CTC CCC AGT GAC TGC AAT TGT |
| MYH4 primer | This study | F: GTC GGC AAT GAG TAT GTC A R: TGA CCA TCC ATA GGA ACA TC |
| FoxO3a primer | This study | F: TGG AGT CCA TCA TCC GTA GTG A R: CTG GTA CCC AGC TTT GAG ATG AG |
| P62 primer | This study | F: CCC AGT GTC TTG GCA TTC TT R: AGG AAA GCA GAG GGA GC TC |
| TGIF primer | This study | F: TTT CCT CAT CAG CAG CCT CT R: CTT GCC ATC CTT TCA GC |
| ATF4 primer | This study | F: TCC TGA ACA GCG AAG TGT TG R: ACC CAT GAG GTT TCA AGT GC |
| Bnip3 primer | This study | F: TTC ACC TAG CAC CTT CTG ATG A R: GAA CAC CGC ATT TAC AGA ACA A |
| Gadd45a primer | This study | F: GAA AGT CGC TAC ATG GAT CAG T R: AAA CTT CAG TGC AAT TTG TTC |
| SMART primer | This study | F: TCA ATA ACC TCA GGC GTC R: GTT TTG CAC ACA AGC TCC A |
| MUSA1 primer | This study | F: TCG TGG AAT GGT AAT CTT GC R: CCT CCC GTT TCT CTA TCA CG |
| DUSP22 (Mouse) siRNA -1 | Invitrogen | 87545 |
| DUSP22 (Mouse) siRNA -2 | Invitrogen | 287290 |
| DUSP22 (Human) siRNA | Invitrogen | 287290 |
| FOXO3a (Mouse) siRNA | Invitrogen | 102011 |
| **Chemicals, Enzymes and other reagents** | | |
| BML-260 | Santa Cruz | SC-223822 |
| Dexamethasone | Santa Cruz | SC-204715A |
| Hygromycin B | Sigma-Alrich | H7772 |
| Blasticidin S HCl | Thermo Fisher Scientific | R210-01 |
| Puromycin | Abcam | ab141453 |
| SP600125 | Santa cruz | sc-200635 |
| Myoplasma removal agent | CAPRICORN | MYX-B |
| **Software** | | |
| *GraphPad7* | https://www.graphpad.com/ | |
| **Other** | | |
| Illumina Novaseq6000 | Illumina | |

## Molecular docking analysis

The binding of BML-260 to human DUSP22 was analyzed with the CB-Dock2 software (Liu et al, 2022). The Vina score was calculated using DUSP22 human (PDB: 6lvq) and BML-260 (CID: 1565747).

## DUSP22 activity assay

The phosphatase activity of purified DUSP22 was measured using the EnzChek® Phosphatase Assay Kit (E12020, ThermoFisher, MA, USA) following the manufacturer's instructions. In brief, for the assay a 200 μM 6,8-difluoro-4-methylumbelliferyl phosphate (DiFMUP) working solution was prepared. 50 μL DiFMUP solution was added to each well of a black-walled microplate, followed by a 50 μL sample containing GST-tagged-DUSP22 (SRP5021, Sigma-Aldrich, MO, USA) or 1X reaction buffer, to give a total reaction volume of 100 μL per well. BML-260 and 1X reaction buffer in a 50 μL volume were then added to each well. The microplate was incubated at room temperature for 45 min in the dark and fluorescence was measured at excitation 360 nm and emission 455 nm (SpectraMAX Gemini XS, Molecular Devices, CA, USA).

## Cell culture

C2C12 murine skeletal muscle myoblasts were purchased from Koram Biotech Corp. (Seoul, Republic of Korea). Prior to experiments, mycoplasma contamination was eliminated using the MycoXpert Mycoplasma Removal Reagent (MYX-B, CAPRI-CORN) following the manufacturer's protocol. To ensure complete removal, all cells were passaged three times after treatment before being used in subsequent experiments. No STR profiling was recently undertaken. Myoblasts were cultured in growth media (GM), containing Dulbecco's modified Eagle's medium (DMEM), 10% fetal bovine serum, and 1% penicillin and streptomycin (PenStrep). C2C12 myoblasts were differentiated into myotubes at more than 80% confluence by culture in differentiation media (DM), containing DMEM supplemented with 2% horse serum and PenStrep, for 96 h. To induce atrophy, the myotubes were treated with DM plus 10 μM Dex for 24 h. BML-260 and SP600125 were purchased from Santa Cruz Biotechnology, TX, USA (cat. no. sc-223822 and sc-200635).

Human skeletal myoblasts were purchased from Thermo-Fisher Scientific, USA, re-suspended in DM and seeded onto 12-well culture plates at a density of $4.8 \times 10^4$ cells/well. 72 h later, the differentiated myotubes were treated with 10 μM Dex for 24 h, stained with haematoxylin and eosin (H&E) using a kit (Merck, Germany) and myotube diameter measured by light microscopy analysis of DIC captured images (Olympus CKX41, Olympus Life Science, Japan) and the NIH imaging software Image J 64 (National Institutes of Health, MD, USA). At least 80 myotubes in 5 micrographs were measured in each experimental group.

## Myotube diameter

Fast myosin heavy chain 2 (MYH2) immunocytochemistry was used to visualize murine myotubes and measure myotube diameter, following the previously published protocol (Chen et al, 2016). After mounting, fluorescence images were captured in 5 different regions using a DMI 3000 B microscope (Leica, Germany), and analyzed using the NIH imaging software Image J 64. 100 myotubes containing at least three nuclei were measured in each experimental group. Myotubes were primarily examined in the central area of each well, as opposed to randomly selected fields. The mid-point of the myotube was used to measure the diameter. For nuclear counts, we chose myotubes containing three or more nuclei. The observer was aware of the treatment conditions.

## Myotube fusion and differentiation indexes

To assess myoblast fusion into multinucleate myotubes, nuclei number within fast myosin (MYH2)-positive cells were measured using the NIH imaging software Image J 64. The differentiation index was calculated as the percentage of nuclei in fast myosin (MYH2)-positive cells (nuclei in myotubes/total nuclei). Myotubes were classified as MYH-positive cells containing at least 4 nuclei. Myotubes were measured in groups of at least 100, with at least 7500 nuclei measured in each group.

## SUnSET assay

The surface sensing of translation (SUnSET) assay was used to measured protein synthesis, in accordance with the published protocol (Vinel et al, 2018). In brief, 1 μg/mL puromycin was added to the myotube cultures and cell lysates were then harvested 10 min later. Western blotting was carried out using an anti-puromycin 12D10 antibody (MABE343; Millipore).

## Western blotting

Western blotting was carried out using the previously published protocol (Lee et al, 2020). The NIH imaging software Image J 64 was used for densitometry analysis of the gel bands. The primary and secondary antibodies used in this study are shown in the reagent and tools table. Chemiluminescence was used to detect the secondary antibody.

## Real-time qPCR

The StepOnePlus Real Time PCR System (Applied Biosystems, UK) was used to measure the mRNA level of the genes of interest. An AccuPower RT PreMix (Bioneer, Republic of Korea) was used to synthesize the cDNA from total RNA. The real-time PCR (qPCR) was carried out according to the manufacturer's instructions, and the previously published protocol (Lee et al, 2020). Details of the primers used in this study are shown in Appendix Tables S1 and S2.

## siRNA-mediated gene knockdown in myotubes

Gene knockdown was carried out following the manufacturer's protocol using a 6-well culture plate format (Thermo Fischer Scientific, Waltham, USA). Lipofectamine 3000 was used for the transfection step (Thermo Fischer Scientific, Waltham, USA). siRNA transfection was performed on day 3 of myoblast differentiation into myotubes. C2C12 myotubes and primary human myotubes were transfected with 75 pmol siRNA duplex. Details of the siRNAs used in this study are shown in Appendix Table S3.

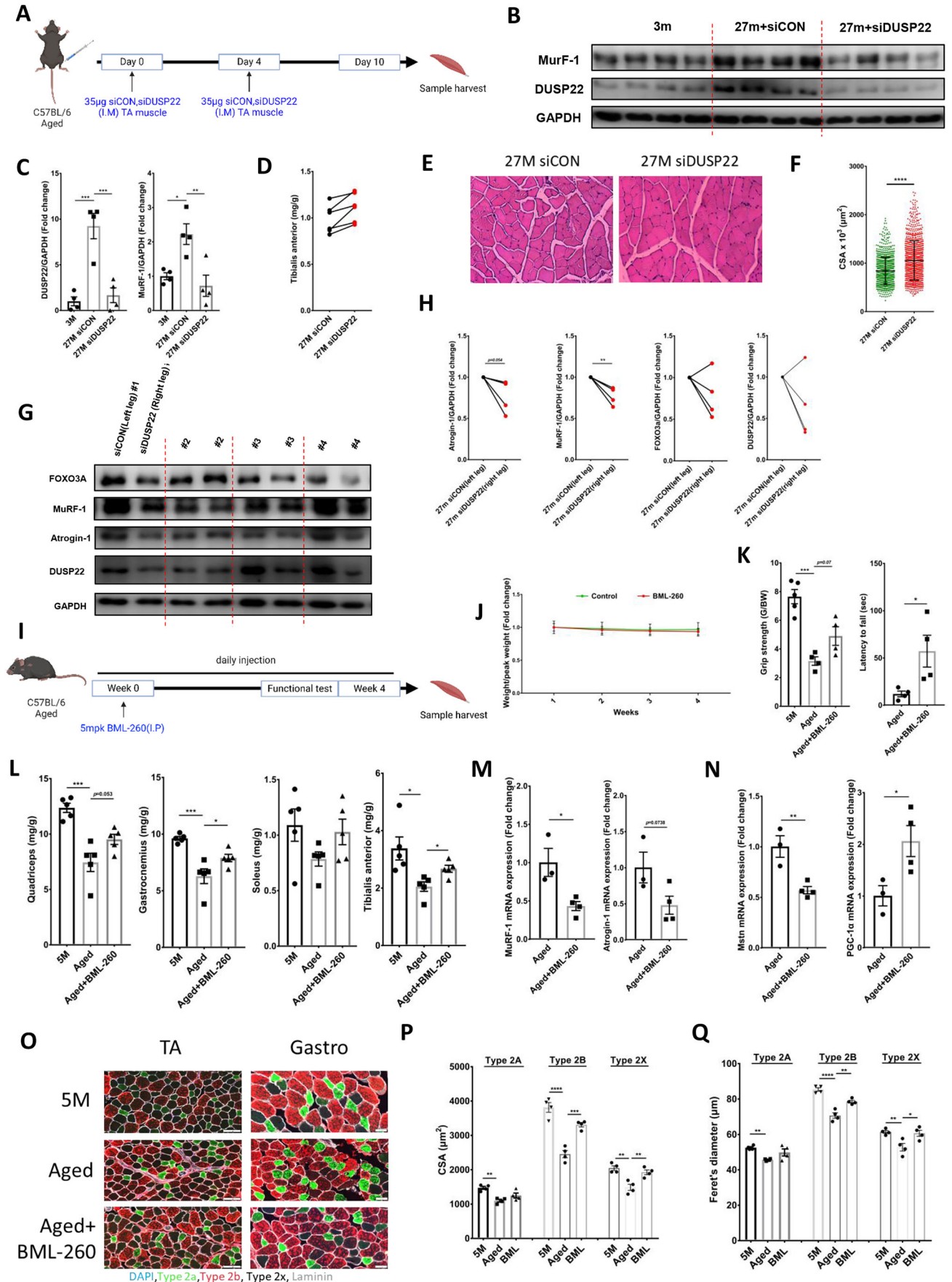

**Figure 7. DUSP22 knockdown or pharmacological protects against aging-induced skeletal muscle wasting.**

(A) Experimental protocol to knockdown DUSP22 expression in the TA of aged mice. (B) Western blot analysis of DUSP22 and MuRF-1 and expression levels ($n = 4$). GAPDH was used for normalization of expression. (C) Quantification of DUSP22 ($p = (3\,M = 0.0004,\ 27\,M+siDUSP22 = 0.0007)$) and MuRF-1 ($p = (3\,M = 0.0148,\ 27\,M +siDUSP22 = 0.0046)$). (D) Change in TA muscle mass ($p = 0.1749$) over the course of the experiment. (E) Representative H&E staining of the TA muscle. (F) TA myofiber CSA ($n = 5,\ p = 7.54E{-}35$). At least 150 fibers were measured for each sample (G) Western blot of FOXO3a, atrogin-1, MuRF-1, and DUSP22 levels in the TA muscle from the control siRNA treated left and DUSP22 siRNA treated right leg ($n = 4$). (H) Change in FOXO3a ($p = 0.1852$), atrogin-1 ($p = 0.0532$), and MuRF-1 ($p = 0.0052$), DUSP22 ($p = 0.1466$) expression relative to GAPDH. (I) Experimental protocol for DUSP22 pharmacological targeting in aged mice. (J) Body weight over the course of the experiment. (K) Grip strength ($p = (5\,M = 0.0002,\ Aged+BML\text{-}260 = 0.0702)$) and rotarod performance ($p = 0.0381$) in the latency to fall test ($n = 5,4$). (L) Mass of the quadriceps ($p = (5\,M = 0.0002,\ Aged+BML\text{-}260 = 0.0539)$, gastrocnemius ($p = (5\,M = 0.0003,\ Aged+BML\text{-}260 = 0.0383)$, TA ($p = (5\,M = 0.0229,\ Aged +BML\text{-}260 = 0.04919)$ and soleus ($p = (5\,M = 0.1484,\ Aged+BML\text{-}260 = 0.2699)$ muscles ($n = 5$). (M) qPCR analysis of MuRF-1($p = 0.0194$) and atrogin-1 ($p = 0.0737$) expression ($n = 3,4$). (N) qPCR analysis of myostatin (Mstn) ($p = 0.0072$) and PGC-1α ($p = 0.0423$) expression ($n = 3,4$). (O) MYHIIa (type 2a), MYHIIb (type 2b), and MYHIIx (type 2x), DAPI and laminin staining in the TA and gastrocnemius muscles (5 M = 5 months-old). Scale bar = 100 μm. (P) CSA of the type 2a ($p = (5\,M = 0.0016,\ Aged+BML\text{-}260 = 0.1403)$, 2b ($p = (5\,M = < 0.0001,\ Aged+BML\text{-}260 = 0.0004)$, and 2x ($p = (5\,M = 0.001,\ Aged+BML\text{-}260 = 0.005)$ myofibers ($n = 4$). (Q) Minimal Feret's diameter of the type 2a ($p = (5\,M = 0.0057,\ Aged+BML\text{-}260 = 0.0662)$, 2b ($p = (5\,M = < 0.0001,\ Aged+BML\text{-}260 = 0.0017)$, and 2x ($p = (5\,M = 0.0092,\ Aged +BML\text{-}260 = 0.0155)$ myofibers. *$p < 0.05$, **$p < 0.01$, ***$p < 0.001$ and ****$p < 0.0001$ indicate significantly increased or decreased. $n$ represents biological replicates. Error bars represent the standard error of the mean (SEM). Source data are available online for this figure.

## CRISPR/Cas9-mediated gene overexpression

To induce endogenous overexpression, C2C12 myoblasts were transfected with a DUSP22 CRISPR activation plasmid (Santa Cruz, SC-430587-ACT, sequence: TGCAGTTTGCGCACGCGCGC). Transfection was performed at 50% confluence. The myoblasts were transfected for 48 h. The transfection efficiency was approximately 5%. Selection was conducted using puromycin dihydrochloride (2 μg/mL), hygromycin B (200 μg/mL), and blasticidin S HCl (20 μg/mL). After approximately two weeks of selection, the selected myoblasts were differentiated into myotubes. The expression levels were then confirmed using qPCR to verify overexpression.

## Animal studies

All C57BL/6 mice (*Mus musculus*) were obtained from Damool Science (Seoul, Republic of Korea). Studies were carried out under the guidance provided by the Institute for Laboratory Animal Research Guide for the Care and Use of Laboratory Animals, and were approved by the Gwangju Institute of Science and Technology Animal Care and Use Committee (study approval number GIST-2021-117). All mice were housed in a temperature-controlled environment (22 °C ± 2 °C, 50% ± 10% humidity) with 12 h light/dark cycle (7 a.m./7 p.m.), 10–15 air changes per hour, and ad libitum access to food and water. Animal studies have been approved by the appropriate ethics committee and have therefore been performed in accordance with the ethical standards laid down in the 1964 Declaration of Helsinki and its later amendments. Animals were provided by Damool Science, Republic of Korea.

## Dex treatment model of skeletal muscle atrophy and DUSP22 knockdown

The Dex model of muscle atrophy was based on the previously published protocol (Kim et al, 2022). In brief, 12-week-old male C57BL/6J mice were treated as follows: 1) Vehicle (DMSO in PBS pH 7.4 with 5% Tween 80) alone, 2) 15 mg/kg Dex dissolved in vehicle, 3) 15 mg/kg Dex and 5 mg/kg BML-260 ($n = 5$ per group). Mice were treated daily by intraperitoneal (IP) injection for 14 d and then assessed for muscle condition. The 5 mg/kg in vivo dose

was selected based on a previous study that carried out bilateral injection of BML-260 into the fat pads of male 8-week-old C57BL/6J mice (Feng et al, 2019).

To knockdown DUSP22 expression, intramuscular delivery of DUSP22 siRNA (siRNA ID: 287290, Thermo Fisher Scientific, MA, USA) was carried out using the Invivofectamine 3.0 reagent ((Thermo Fisher Scientific) following the manufacturer's instructions and previously published protocols (Chemello et al, 2019; Militello et al, 2018). In brief, for the Dex model of muscle wasting, 35 μg siRNA complexes in a 50 μL volume were injected into the TA muscle of 12-week-old C57BL/6 mice ($n = 4$ per group), 24 h before IP treatment with 25 mg/kg Dex. Mice were treated with 25 mg/kg Dex every 24 h for 7 days, and received a second injection of 35 μg siRNA complex into the TA muscle at day 3 of the Dex treatment. Muscles were harvested for analysis after 7 days of Dex treatment.

## Assessment of potential organ toxicity

To assess the effect of BML-260 treatment on the liver, heart and kidney, 15-month-old C57BL/6J male mice were treated as follows: 1) Vehicle (DMSO in PBS pH 7.4 with 5% Tween 80) alone, 2) 5 mg/kg BML-260 ($n = 4$ per group). BML-260 was administered via IP delivery every 24 h for 6 weeks. The mice were then sacrificed and the liver, kidneys and heart were weighed and photographed.

## Skeletal muscle aging model and DUSP22 targeting

To knockdown DUSP22 expression in aging skeletal muscle, 27-month-old male C57BL/6 mice received intramuscular delivery of DUSP22 siRNA using the Invivofectamine 3.0 reagent ($n = 4$ per group). 35 μg siRNA complexes were injected into the TA muscle, followed by a second injection 4 days later. TA muscles were harvested for analysis at day 10 of the experiment.

To assess DUSP22 pharmacological targeting, BML-260 was treated to 24–26-month-old male C57BL/6 mice as follows: (1) vehicle alone (5% DMSO + 5% tween 80) and (2) 5 mg/kg BML-260 ($n = 5$ per group). Mice were treated via IP delivery every 24 h for 4 weeks and then assessed for skeletal muscle function and condition.

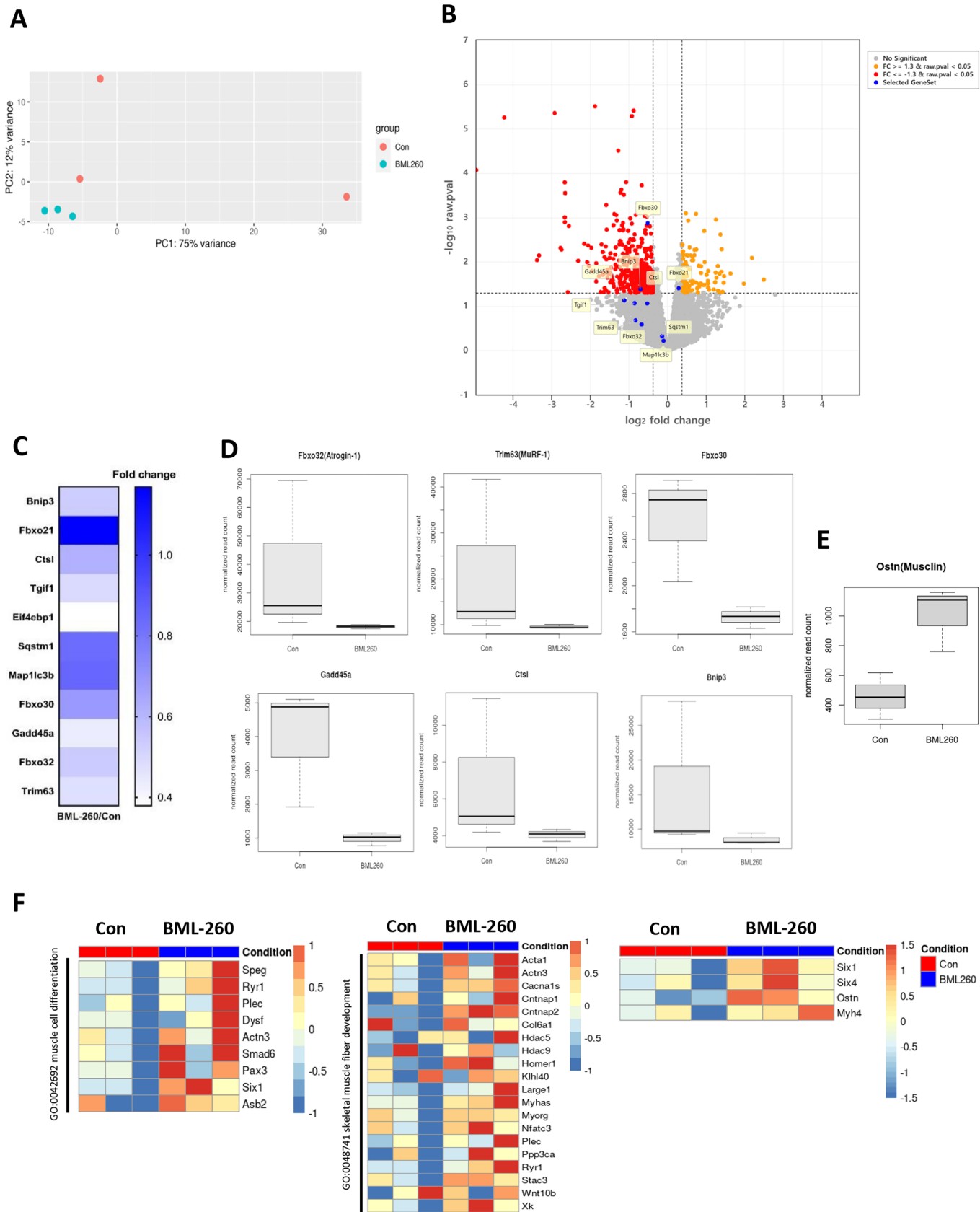

◄ **Figure 8. Whole genome RNA Seq analysis of DUSP22 pharmacological targeting in geriatric mouse TA muscle.**

(A) Principal component analysis of gene expression variance in the vehicle treated (Con) and BML-260 treated groups ($n = 3$). (B) Volcano plot of differentially expressed genes. Selected genes linked to FOXO3a signaling are indicated. (C) Expression analysis of genes related to FOXO3a signaling measured by overall TPM fold change. The control was set as the baseline value of 1. A value of 2 would indicate a 2-fold increase, and 0.5 would indicates a 2-fold decrease. (D) Expression of selected genes linked to FOXO3a signaling as normalized counts from the raw read. (E) Expression of the exercise-induced myokine musclin (Ostn). (F) Gene ontology (GO) analysis of the cellular components muscle cell differentiation and muscle fiber development. Also shown are Six1 and Six4 (which specify fast twitch myofiber development), Ostn and fast myosin (MYH4; the predominant myosin type in the fast-twitch TA muscle). Box plots represent the distribution of Fbxo32, Trim63, Fbxo30, Gadd45a, Ctsl, Bnip3, Ostn expression levels. The center line indicates the median (50th percentile, Q2), representing the middle value of the dataset. The box bounds correspond to the interquartile range (IQR), extending from the 25th percentile (Q1, lower bound) to the 75th percentile (Q3, upper bound). Whiskers extend to the smallest and largest values within 1.5 × IQR from Q1 and Q3, representing the minimum (lower whisker) and maximum (upper whisker) values within this range. Data points that fall beyond this range are considered outliers and are displayed as individual points outside the whiskers. n represents biological replicates. Error bars represent the standard error of the mean (SEM). Source data are available online for this figure.

## Immobilization model

To immobilize the hind limbs, 14-week-old male C57BL/6 mice were anesthetized with isoflurane and both hind limbs were fixed with plastic EP-tubes (Axygen, MCT-175-C). To prevent the tubes from slipping, the junction between the tubes and the hind limbs were wrapped with insulating tape. BML-260 was treated as follows: (1) vehicle alone (5% DMSO + 5% tween 80) and (2) 5 mg/kg BML-260 ($n = 3$ per group). The mice were monitored daily and sacrificed after 14 days.

## Grip strength test

Grip strength was measured using the BIO-GS3 device (Bioseb, FL, USA). Mice were placed onto the grid with all four paws attached and gently pulled backwards to measure the grip strength until the grid was released. The maximum grip value, used to represent muscle force, was calculated using 3 trials with an interval of 30 s.

## Muscle fatigue test

Muscle fatigue was measured the previously described protocol (Chiu et al, 2018), with two different models using the Rotarod. One was the constant model and the other was the accelerating model that is inherent in the rotarod device. In brief, the mice were accommodated to training before the commencement of the fatigue task using an accelerating rotarod (Ugo Basile, Italy). The mice were trained at a constant speed of 13 rpm for 15 min. After a recovery period of 15 min, the mice were placed on the rotarod adjusted to accelerate from 13 to 25 rpm in 3 min for 15 min. 24 h later, the muscle fatigue test was carried out with speeds ramping from 13 to 25 rpm in 3 min, and maintained at 25 rpm for a 30 min interval. Latency to fall off the rotarod was measured for each mouse. A fatigued mouse was classified as falling off 4 times within a 1 min period, which also terminated the test.

## Skeletal muscle dissection and histology

Mice were anesthetized using ketamine (22 mg/kg; Yuhan, Republic of Korea) and xylazine (10 mg/kg; Bayer, Republic of Korea) in saline by IP injection. The quadriceps, gastrocnemius, soleus and TA muscles were then dissected and weighed. For histological analysis, the muscles were fixed by overnight incubation with 4% paraformaldehyde in PBS pH 7.4 at 4 °C. Paraffin sectioning and H&E staining were provided by the Animal Housing Facility,

Gwangju Institute of Science and Technology, Republic of Korea (5 μm muscle sections using a Thermo Scientific HistoStar (Thermofisher). H&E staining was carried out with a kit (Merck, Germany)). For immunohistochemistry, TA muscle sections were permeabilized for 15 min (PBS with 0.5% Triton X-100), blocked for 1 h (5% BSA) and incubated with the primary antibody overnight at 4 °C (a list of primary and secondary antibodies used in this study are provided in Appendix Tables S4 and S5, respectively). After washing with PBS 3 times for 5 min, the sections were incubated with the secondary antibody at RT for 1 h. The nuclei were counterstained with DAPI in ProLong™ Gold Antifade Mountant (Invitrogen, P36935). Digital images were acquired with Leica DM 2500 (Leica Microsystems). Cross sectional area (CSA) and fiber size distribution was measured using the NIH imaging software Image J 64. To CSA and size distribution of different fiber types was calculated in 4 muscle section as follows: type 2 A: 30–40 fibers measured; type 2B: at least 100 fibers measured; type 2X: 30–40 fibers measured.

## Muscle force measurement

TA muscles were dissected from each leg of the siDUSP22 and siCON treated groups immediately after sacrifice. The isolated muscles were transferred to a Krebs-Henseleit buffered solution chamber (820MS, Danish Myo Technology, AnimaLab, Poland) for oxygenation and maintained at 25 °C. Both ends of the muscles were then fixed with clamps. Tetanic contraction force was measured by applying a 10 V stimulus and twitch force was measured by conducting at 5 V with frequencies ranging from 10 Hz to 200 Hz. 3 min rest time was allocated between each frequency measurement.

## RNA-Seq

RNA samples were harvested from the TA muscles of 24–26-month-old male C57BL/6 mice treated as follows: (1) vehicle alone (5% DMSO + 5% tween 80) and (2) 5 mg/kg BML-260. Mice were treated by IP delivery every 24 h for 4 weeks ($n = 3$ per group). RNA-Seq was provided by Macrogen, Republic of Korea. Prior to the commencement of sequencing, QC was carried out using FastQC v0.11.7 (http://www.bioinformatics.babraham.ac.uk/projects/fastqc/). Illumina paired ends or single ends in the sequenced samples were trimmed using Trimmomatic 0.38 with various parameters (http://www.usadellab.org/cms/?page=trimmomatic). The sequences of the samples were mapped and analyzed using HISAT2 version 2.1,0, Bowtie2 2.3.4.1 (https://ccb.jhu.edu/software/hisat2/index.shtml).

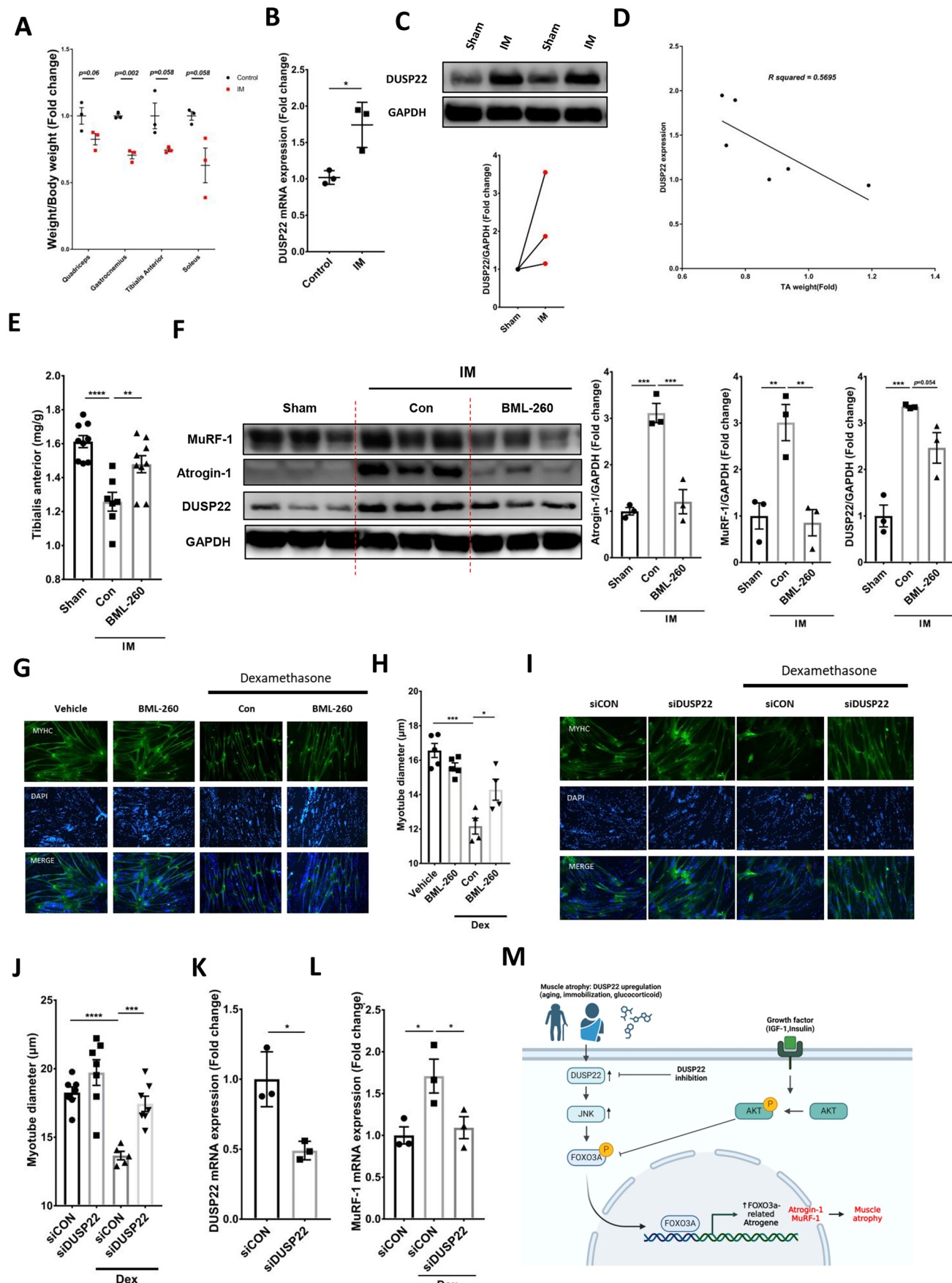

**Figure 9. DUSP22 targeting downregulates atrogene expression in immobilized skeletal muscle and inhibits atrophy in human donor myotubes.**

(A) Effect of immobilization on skeletal muscle mass using the plastic EP tube method ($n = 3$). (B) qPCR analysis of DUSP22 expression in the TA muscle (IM = immobilized) ($n = 3$) ($p = 0.018$). (C) Plot of DUSP22 expression in relation to TA mass ($n = 6$). (D) Western blot analysis of DUSP22 expression ($n = 3$) ($p = 0.1699$). (E) TA mass ($n = 9,7$) ($p = $ (Sham $ = <0.0001$, BML-260 $ = 0.007$)). (F) Western blot analysis of atrogin-1($p = $ (Sham $ = 0.0005$, BML-260 $ = 0.0009$)), MuRF-1 ($p = $ (Sham $ = 0.008$, BML-260 $ = 0.0057$)), and DUSP22 ($p = $ (Sham $ = 0.0007$, BML-260 $ = 0.054$)) levels in the TA muscle ($n = 3$). Quantification of atrogin-1, MuRF-1, and DUSP22 levels relative to GAPDH are also shown. (G) Micrographs of MYH2-immunostained human myotubes treated as follows: 1) Vehicle alone, 2) 10 μM Dex for 24 h, 3) 10 μM Dex and 12.5 μM BML-260 for 24 h. (H) Mean myotube diameter ($n = 5,4$) ($p = $ (Vehicle $ = 0.0001$, BML-260 $ = 0.033$)). (I) Micrographs of MYH2-immunostained human myotubes treated as follows: (1) 48 h incubation in with DM plus control, scrambled siRNA; (2) 48 h incubation with DM plus DUSP22 siRNA: (3) 24 h incubation with DM plus control, scrambled siRNA and additional 24 h treatment with 10 μM Dex plus siRNA; (4) 24 h incubation with DM plus DUSP22 siRNA and additional 24 h treatment with 10 μM Dex plus siRNA. (J) Myotube diameter ($n = 7,5$) ($p = $ (siCON $ = 9.79E-06.$, siDUSP22+Dex $ = 0.0003$)). (K) qPCR analysis of DUSP22 expression in the Dex-treated human myotubes ($n = 3$) ($p = 0.0129$). (L) qPCR analysis of MuRF-1($p = $ (siCON $ = 0.0286$, siDUSP22+Dex $ = 0.049$)) expression in the Dex-treated human myotubes ($n = 3$). *$p < 0.05$, **$p < 0.01$, and ***$p < 0.001$ indicate significantly increased or decreased. (M) Working model of the effect of DUSP22 targeting on skeletal muscle atrophy: 1) In healthy muscle, Akt signaling can promote hypertrophy by increasing protein synthesis and inhibiting the activity of FOXO3a. 2) In the context of skeletal muscle wasting, the Akt pathway can become suppressed and FOXO3a signaling is upregulated. The results from this study show that targeting DUSP22 in wasting muscle downregulates JNK and reduces FOXO3a signaling. These events occur independently of Akt signaling activation, which remains suppressed. DUSP22 targeting, via pharmacology or gene knockdown, is sufficient to enhance function, improve histopathology, and lower atrogene expression in multiple forms of skeletal muscle wasting. n represents biological replicates. Error bars represent the standard error of the mean (SEM). Source data are available online for this figure.

Potential transcripts and multiple splice variants were assembled using StringTie version 2.1.3b (https://ccb.jhu.edu/software/stringtie/). The normalized sample counts were generated using DESeq2, which was also used for normalization, visualization, and differential analysis. PCA was performed using normalized sample counts. The "apeglm" type was used to calculate shrink log2 fold changes.

## Pharmacokinetics analysis

Assessment of BML-260 pharmacokinetics (PK) was carried out by the company Chaon, Yongin-si, Republic of Korea. BML-260 was administered intraperitoneally at a dose of 5 mg/kg in an injection volume of 10 mL/kg. A total of 21 6-week-old male C57BL/6J mice were divided into two groups as follows: 1) PK (plasma) group (3 mice) and 2) PK (TA muscle) group (18 mice across 6 time points).

For the PK (plasma) group, blood samples were collected at 0.5, 1, 1.5, 2, 6, 12, and 24 h after administration, using orbital sinus collection. Blood was centrifuged to obtain plasma that was then stored at −80 °C. For the PK (TA muscle) group, mice were euthanized at 0.5, 1, 1.5, 2, 6, and 12 h post-administration. The TA muscle was weighed and stored at −80 °C.

BML-260 levels in plasma and tissue samples were analyzed with liquid chromatography-tandem mass spectrometry (LC-MS/MS) using an Agilent 1200 HPLC system coupled with an AB Sciex 4000 QTrap MS/MS (Agilent Technologies, CA, USA). Quantification limits were determined based on a signal-to-noise ratio greater than 10, with precision below 20% and accuracy between 80% and 120%.

## Statistical analysis

Experiments were randomized, and investigators were not blinded to the allocation process during the experiments and outcome assessments. The sample size was determined based on the known variability of each experiment. Additionally, a power analysis was conducted to determine the appropriate sample size.

Mice displaying any signs of distress, as predefined in the Gist LARC Standard Operating Procedure, were euthanized and excluded from the study.

The Student's *t* test or ANOVA with the Dunnett's test was used to determine statistical significance for two groups, or three or more groups, respectively (GraphPad 7.0 Software, Inc., CA, USA). *p* values of less than 0.05 were deemed to be statistically significant. Unless otherwise stated, experiments were carried out in triplicate and the error bars are presented as mean ± standard error of the mean (SEM).

## Graphics

The graphics in Figs. 5A, 6A, 7A, 7I, and 9M were created with BioRender.com.

# Data availability

The dataset produced in this study is available in the following database: RNA-seq (Gene Expression Omnibus number: GSE292258).

---

### The paper explained

#### Problem
Skeletal muscle wasting is produced by aging and aging-related degenerative diseases. This disorder has become a major socio-economic issue due to population aging and there is no approved therapeutic drug.

#### Results
We discovered that the signaling phosphatase, DUSP22, is upregulated in patients and animal models of skeletal muscle wasting. Muscle-specific DUSP22 gene knockdown inhibited wasting in these models and suppressed the upregulation of FOXO3a, a master atrogene that induces multiple wasting-related signaling pathways. Targeting DUSP22 with the rhodanine-based small molecule BML-260 was also effective at preventing muscle wasting. The effect of DUSP22 targeting on FOXO3a was shown to be mediated by the stress-regulated kinase, JNK, which was previously shown to be activated by DUSP22. Interestingly, these therapeutic effects occurred without activation of the PI3K-Akt pathway.

#### Impact
The DUSP22-JNK-FOXO3a signaling axis represents a novel strategy to develop therapeutics for skeletal muscle wasting. Bypassing the PI3K-Akt pathway may help to avoid aberrant proteostasis in aged muscle. BML-260 could be an attractive lead compound for future drug development.

The source data of this paper are collected in the following database record: BioStudies (images Accession number: S-BIAD1435).

The source data of this paper are collected in the following database record: biostudies:S-SCDT-10_1038-S44321-025-00234-2.

## Peer review information

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

## Acknowledgements

This work was supported by the National Research Foundation of Korea (NRF) funded by the Korean government (MSIT) [grant no. RS-2025-00562398 and grand no. RS-2022-NR069434]. This work was also funded by the Korean government (MSIP) through the Institute for Information and Communications Technology Promotion (IITP) grant (No. RS-2019-II190567, Development of Intelligent SW systems for uncovering genetic variation and developing personalized medicine for cancer patients with unknown molecular genetic mechanisms).

## Author contributions

**Sang-Hoon Lee**: Conceptualization; Data curation; Formal analysis; Validation; Investigation; Visualization; Methodology; Writing—original draft; Writing—review and editing. **Hyun-Jun Kim**: Validation; Methodology; Writing—review and editing. **Seon-Wook Kim**: Methodology. **Hyunju Lee**: Software; Funding acquisition; Methodology; Writing—review and editing. **Da-Woon Jung**: Conceptualization; Supervision; Funding acquisition; Writing—original draft; Writing—review and editing. **Darren Reece Williams**: Conceptualization; Supervision; Funding acquisition; Validation; Writing—original draft; Project administration; Writing—review and editing.

Source data underlying figure panels in this paper may have individual authorship assigned. Where available, figure panel/source data authorship is listed in the following database record: biostudies:S-SCDT-10_1038-S44321-025-00234-2.

## Disclosure and competing interests statement

S-HL, HL, D-WJ, and DRW are named co-inventors of a pending provisional patent application based in part on the research reported in this paper.

# Expanded View Figures

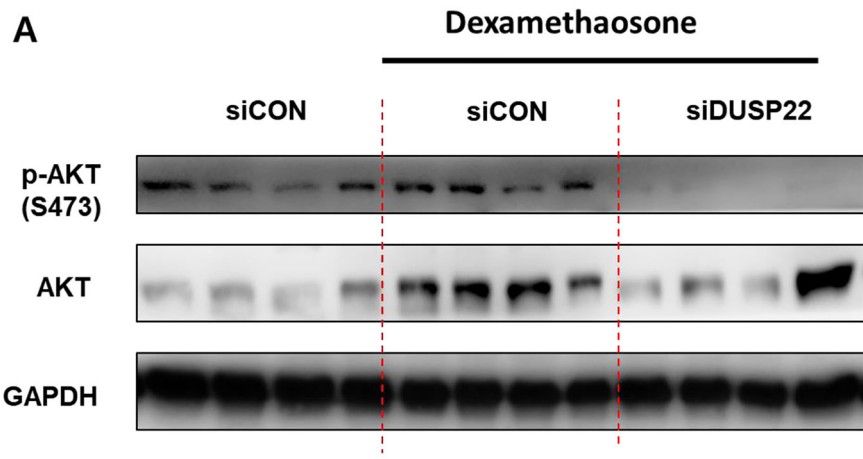

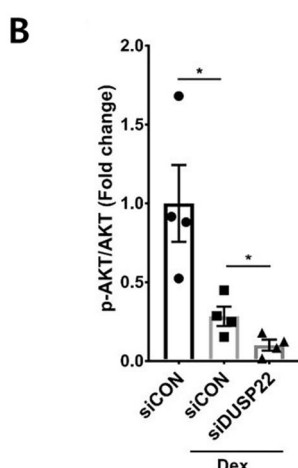

**Figure EV1. DUSP22 siRNA reduces the phosphorylation of AKT.**

(A, B) Western blot and densitometry analysis of AKT phosphorylation in the Dex-treated tibialis anterior (TA) muscle 3 d after delivery of control or DUSP22 siRNA ($n = 4$) ($p =$ (siCON = 0.029, siDUSP22+Dex = 0.0412)). GAPDH was used for normalization of expression. *$p < 0.05$ indicate significantly increased or decreased. n represents biological replicates analyzed by Student's t test. Error bars represent the standard error of the mean (SEM). Source data are available online for this figure.

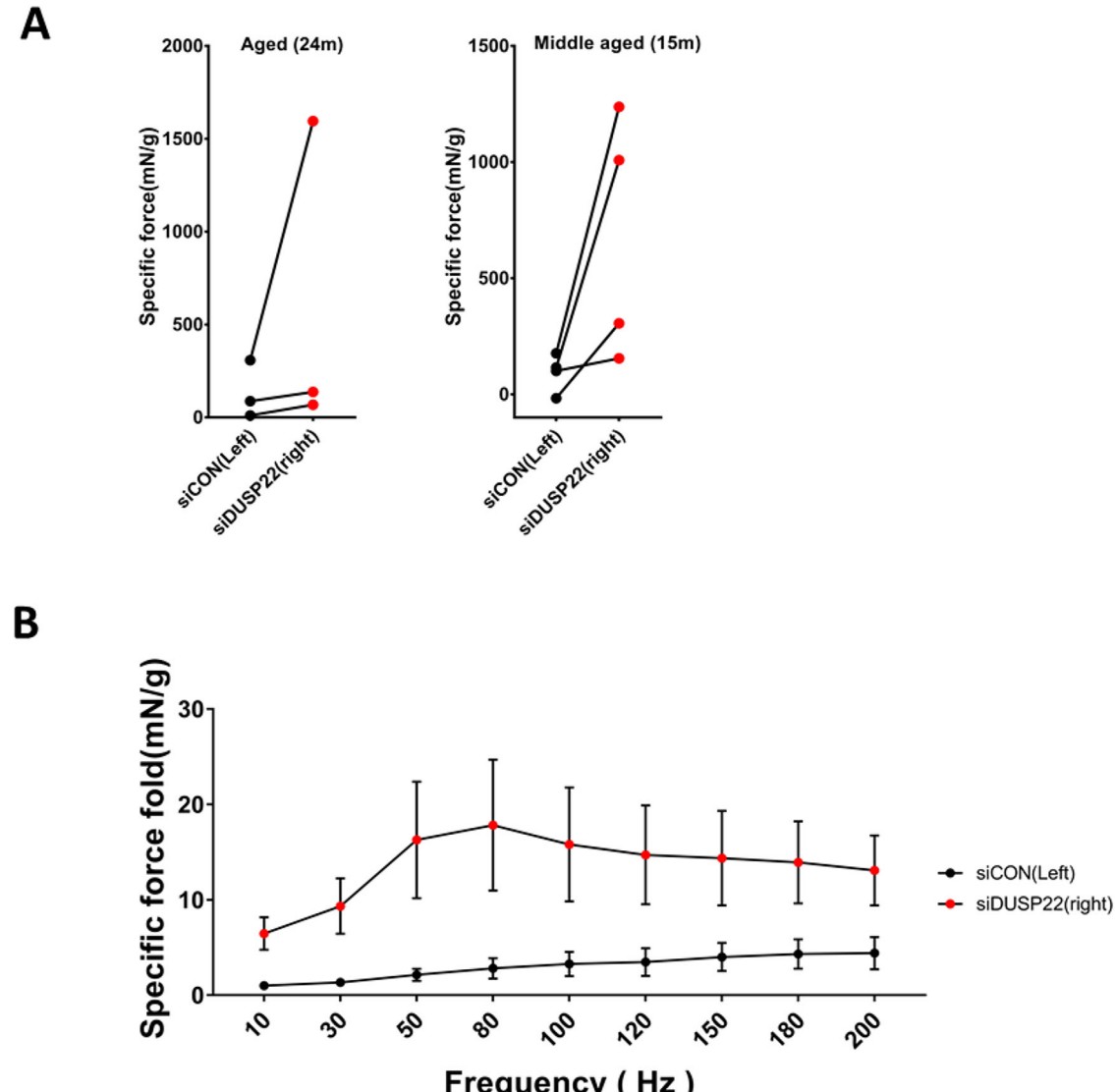

**Figure EV2. DUSP22 siRNA enhances tetanic and twitch muscle force in the tibialis anterior muscle.**

(A) Tetanic muscle contraction measurement in the TA muscle of aged ($n = 3$) ($p = 0.3974$) or middle aged mice ($n = 4$) ($p = 0.0807$) 3 d after the delivery of control or DUSP22 siRNA. (B) Twitch force measure in 15-month-old mice 3 d after the delivery of control or DUSP22 siRNA ($n = 4$) ($p = 0.1054$, $0.1677$, $0.1964$, $0.1858$, $0.2118$, $0.234$, $0.2489$, $0.2684$, $0.2947$). $n$ represents biological replicates analyzed by Student's $t$ test. Error bars represent the standard error of the mean (SEM). Source data are available online for this figure.

## BML-260 detection : Plasma

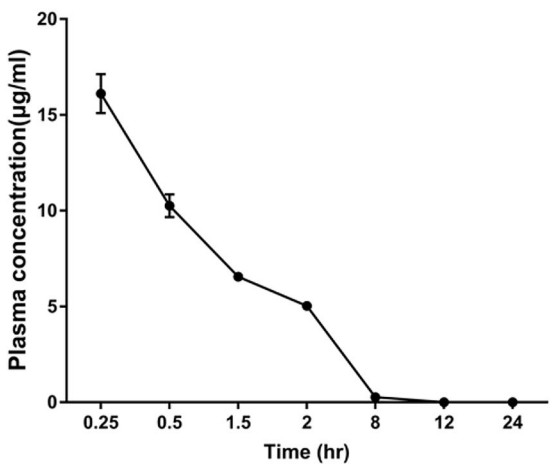

| PK Parameters | Unit | BML-260 |
|---|---|---|
| | | IP |
| Dose | mg/kg | 5.00 |
| $AUC_{last}$ | h*ug/mL | 26.14 |
| $AUC_{inf}$ | h*ug/mL | 26.70 |
| $C_{max}$ | ug/mL | 16.11 |
| $T_{max}$ | h | 0.25 |
| $V_z/F$ | L/kg | 0.38 |
| Cl/F | L/h/kg | 0.19 |
| $t_{1/2}$ | h | 1.42 |

## BML-260 detection : Skeletal muscle (tibialis anterior)

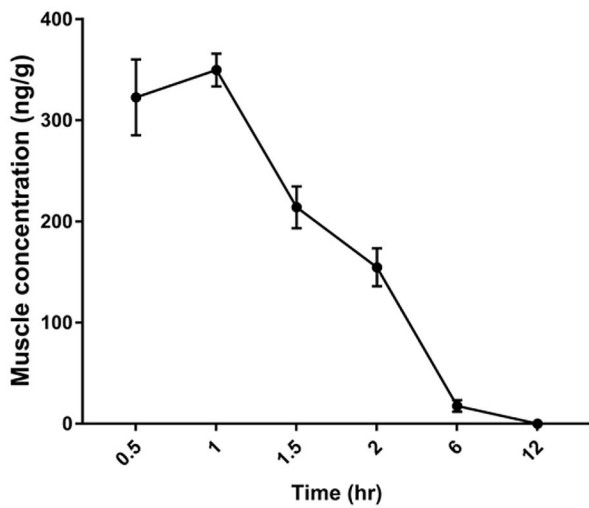

| PK Parameters | Unit | BML-260 |
|---|---|---|
| | | IP |
| Dose | mg/kg | 5.00 |
| $AUC_{last}$ | h*ug/g | 0.73 |
| $AUC_{inf}$ | h*ug/g | 0.76 |
| $C_{max}$ | ug/g | 0.35 |
| $T_{max}$ | h | 1.00 |
| $V_z/F$ | g/kg | 11871.11 |
| Cl/F | g/h/kg | 6569.42 |
| $t_{1/2}$ | h | 1.25 |

**Figure EV3. Pharmacokinetic analysis of BML-260 in 6-week-old male C57BL/6J mice.**

Pharmacokinetic analysis of BML-260 in 6-week-old male C57BL/6J mice (5 mg/kg delivered via IP injection) ($n = 3$). BML-260 detection was assessed in the plasma and tibialis anterior muscle. AUClast: Area under the concentration-time curve from time of dosing to the last measurable concentration. AUCinf: Area under the concentration-time curve extrapolated to infinity based on the last measurable concentration. Cmax: Maximum plasma concentration. Tmax: Time to reach the maximum plasma concentration. VZ/F: Apparent volume of distribution during the terminal phase. Cl/F: Apparent total body clearance after extravascular administration. t1/2: Terminal half-life. $n$ represents biological replicates. Error bars represent the standard error of the mean (SEM). Source data are available online for this figure.

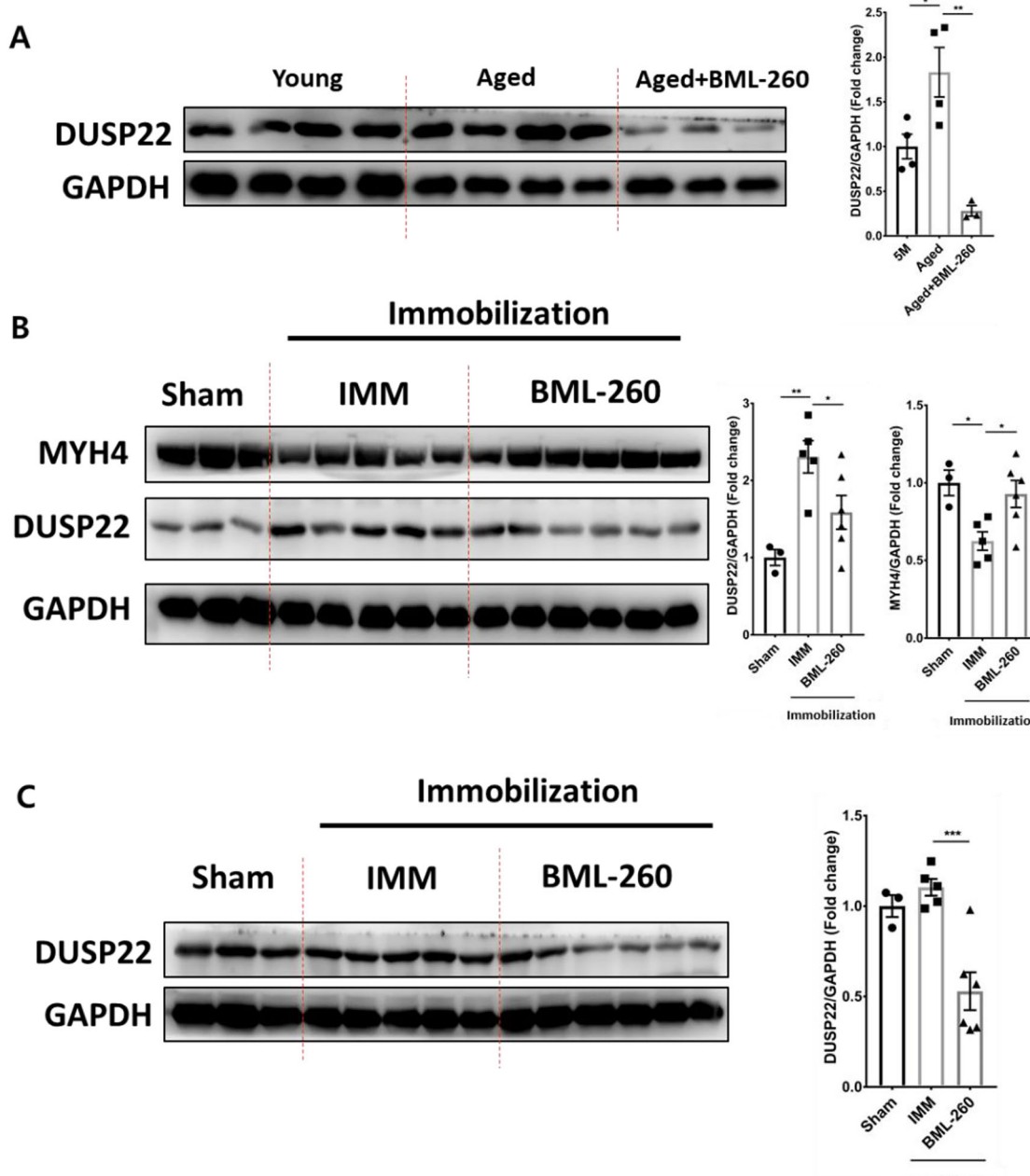

**Figure EV4. DUSP22 and MYH4 levels in skeletal muscles of aged or immobilized mice treated with BML-260.**

(A) Western blot analysis of DUSP22 expression in the TA of aged mice treated with BML-260 ($n = 4,3$) ($p = $ (Young $= 0.0362$, Aged+BML-260 $= 0.0054$)). GAPDH was used for normalization of expression. *$p < 0.05$ and **$p < 0.01$ indicate significantly increased or decreased. (B, C) Western blot analysis of DUSP22 expression in the TA ($n = 3,5,6$) ($p = $ (Sham $= 0.0051$, BML-260 $= 0.0488$)) (B) and gastrocnemius muscle ($n = 3,5,6$) ($p = $ (Sham $= 0.6845$, BML-260 $= 0.0008$)) (C) of immobilized (IMM) mice treated with BML-260. For the TA muscle, MYH4 (myosin heavy chain 2B) ($p = $ (Sham $= 0.0253$, BML-260 $= 0.0288$)) levels are also shown. GAPDH was used for normalization of expression. *$p < 0.05$, **$p < 0.01$, and ***$p < 0.001$ indicate significantly increased or decreased. n represents biological replicates analyzed by Student's t test. Error bars represent the standard error of the mean (SEM). Source data are available online for this figure.

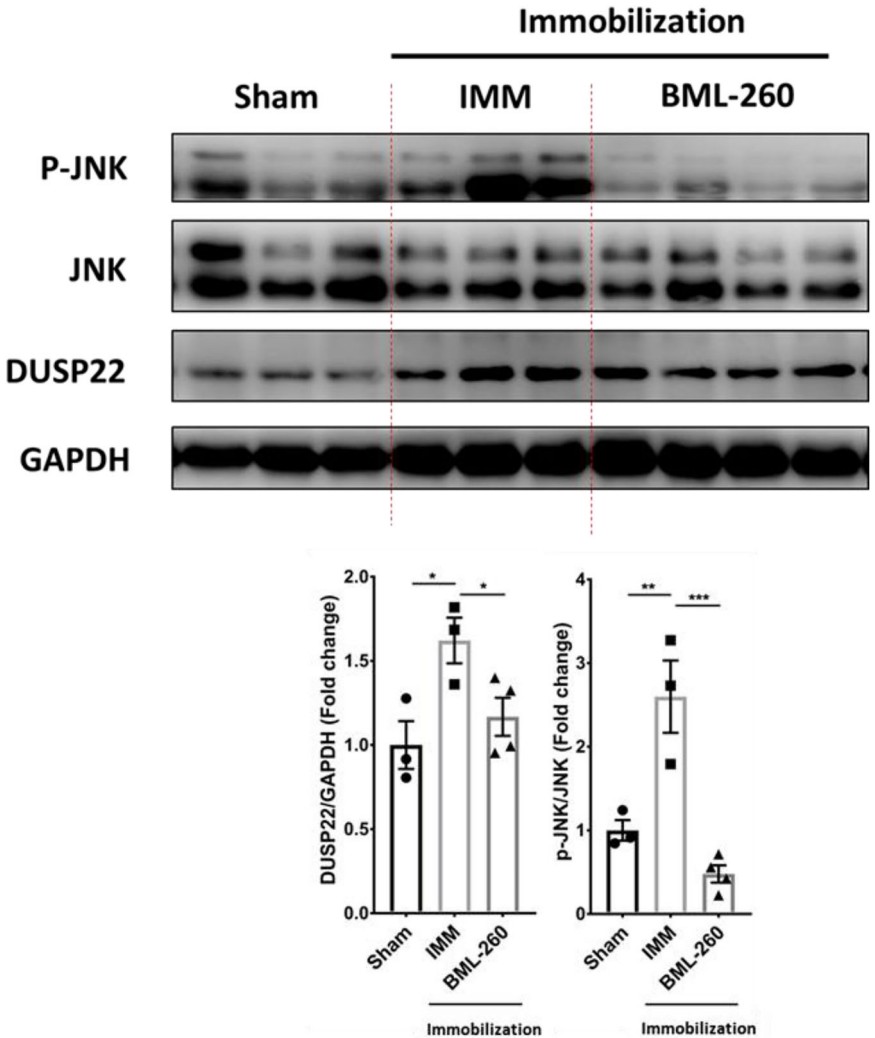

**Figure EV5.  DUSP22, JNK and phosphorylated JNK levels in the tibialis anterior of immobilized mice treated with BML-260.**

Western blot analysis of DUSP22 ($p$ = (Sham = 0.0341, BML-260 = 0.0494)), JNK and phosphorylated JNK (JNK-P) ($p$ = (Sham = 0.0052, BML-260 = 0.0007)) levels in the TA muscle of immobilized mice ($n$ = 3,4). GAPDH was used for normalization of expression. $*p < 0.05$, $**p < 0.01$, and $***p < 0.001$ indicate significantly increased or decreased. n represents biological replicates analyzed by Student's t test. Error bars represent the standard error of the mean (SEM). Source data are available online for this figure.

