## [Peer Review File · EMBO Molecular Medicine]

Modulating phosphatase DUSP22 with BML-260 ameliorates skeletal muscle wasting via Akt independent JNK-FOXO3a repression

Darren Williams, Sang Hoon Lee, Hyun-Jun Kim, Seon-wook Kim, Hyunju Lee, and Da-Woon Jung

Corresponding authors: Darren Williams (darren@gist.ac.kr) , Da-Woon Jung (jung@gist.ac.kr)

Review Timeline:

Submission Date:	9th Apr 24
Editorial Decision:	26th Apr 24
Revision Received:	5th Nov 24
Editorial Decision:	25th Nov 24
Revision Received:	25th Feb 25
Editorial Decision:	17th Mar 25
Revision Received:	24th Mar 25
Accepted:	26th Mar 25

Editor: Lise Roth

Transaction Report:

26th Apr 2024

Dear Prof. Williams,

Thank you for the submission of your manuscript to EMBO Molecular Medicine. We have now received feedback from the three reviewers who agreed to evaluate your manuscript. As you will see from the reports below, the reviewers acknowledge the interest of the study, but also raise several concerns that should be addressed in a revision of this manuscript.

Of note, while referee #3 mentioned the limited novelty of the findings based on the known link between JNK and DUSP22, referees #1 and #2 agreed (as we do) that the translational advance presented in your study warrants further consideration.

Addressing the other reviewers' concerns in full will be necessary for further considering the manuscript in our journal, and acceptance of the manuscript will entail a second round of review. EMBO Molecular Medicine encourages a single round of revision only and therefore, acceptance or rejection of the manuscript will depend on the completeness of your responses included in the next, final version of the manuscript. For this reason, and to save you from any frustrations in the end, I would strongly advise against returning an incomplete revision.

We are expecting your revised manuscript within three to six months, if you anticipate any delay, please contact us.

We require:

4) A .docx formatted letter INCLUDING the reviewers' reports and your detailed point-by-point responses to their comments. As part of the EMBO Press transparent editorial process, the point-by-point response is part of the Review Process File (RPF), which will be published alongside your paper.

5) A complete author checklist, which you can download from our author guidelines (<https://www.embopress.org/page/journal/17574684/authorguide#submissionofrevisions>). Please insert information in the checklist that is also reflected in the manuscript. The completed author checklist will also be part of the RPF.

6) Please note that all corresponding authors are required to supply an ORCID ID for their name upon submission of a revised manuscript. An ORCID identifier is missing for Prof. Da-Woon Jung.

7) It is mandatory to include a 'Data Availability' section after the Materials and Methods. Before submitting your revision, primary datasets produced in this study need to be deposited in an appropriate public database, and the accession numbers and database listed under 'Data Availability'. Please remember to provide a reviewer password if the datasets are not yet public (see <https://www.embopress.org/page/journal/17574684/authorguide#dataavailability>).

8) For data quantification: please specify the name of the statistical test used to generate error bars and P values, the number (n) of independent experiments (specify technical or biological replicates) underlying each data point and the test used to calculate p-values in each figure legend. The figure legends should contain a basic description of n, P and the test applied. Graphs must include a description of the bars and the error bars (s.d., s.e.m.). Please provide exact p values.

9) Our journal encourages inclusion of *data citations in the reference list* to directly cite datasets that were re-used and obtained from public databases. Data citations in the article text are distinct from normal bibliographical citations and should directly link to the database records from which the data can be accessed. In the main text, data citations are formatted as follows: "Data ref: Smith et al, 2001" or "Data ref: NCBI Sequence Read Archive PRJNA342805, 2017". In the Reference list, data citations must be labeled with "[DATASET]". A data reference must provide the database name, accession

number/identifiers and a resolvable link to the landing page from which the data can be accessed at the end of the reference. Further instructions are available at .

13) Author contributions: CRediT has replaced the traditional author contributions section because it offers a systematic machine readable author contributions format that allows for more effective research assessment. Please remove the Authors Contributions from the manuscript and use the free text boxes beneath each contributing author's name in our system to add specific details on the author's contribution. More information is available in our guide to authors.

16) As part of the EMBO Publications transparent editorial process initiative (see our Editorial at <http://embomolmed.embopress.org/content/2/9/329>), EMBO Molecular Medicine will publish online a Review Process File (RPF) to accompany accepted manuscripts.

In the event of acceptance, this file will be published in conjunction with your paper and will include the anonymous referee reports, your point-by-point response and all pertinent correspondence relating to the manuscript. Let us know whether you agree with the publication of the RPF and as here, if you want to remove or not any figures from it prior to publication. Please note that the Authors checklist will be published at the end of the RPF.

I look forward to receiving your revised manuscript.

Yours sincerely,

Lise Roth

***** Reviewer's comments *****

Referee #1 (Remarks for Author):

In this manuscript by Lee et al., the authors found that DUSP22 mRNA levels were upregulated in Sarcopenia people (from GEO database) and Dex-induced mouse model of muscle atrophy. They found that DUSP22 induced the expression of atrophy-related genes (atrogenes), including Atrogin-1 and MuRF-1 in skeletal muscle cells. In addition, DUSP22 also stimulated JNK and FOXO3a signaling but did not affect AKT signaling. The authors found that obliteration of DUSP22 function using DUSP22 siRNAs or a DUSP22 inhibitor (BML-260) prevented muscle wasting in muscle cells, Dex-induced atrophy animals, aged mice, immobilized skeletal muscle, and Dex-induced human myotube atrophy. Collectively, the authors proposed that DUSP22 plays an important role in skeletal muscle wasting via targeting JNK-FOXO3a signaling-induced Atrogenes, and DUSP22 may be an ideal therapeutic drug target for skeletal muscle wasting. Overall, the function of DUSP22 in skeletal muscle wasting is a novel and interesting finding. However, there are major technical concerns need to be addressed. The authors also should clarify whether DUSP22 affects FOXO3a protein stability in skeletal muscle cells.

1. In Fig. 1, only mRNA expression levels were studied. Protein levels for some of the critical genes may need to be analyzed as DUSP22 is a protein phosphatase that regulates post-translational modification of target proteins.
2. In Figure 1J, UBR2 mRNA levels were upregulated by DUSP22 in skeletal muscle cells. However, UBR2 proteins have been reported to be downregulated by DUSP22 in T cells. The authors should address this issue by performing western blotting or by discussion.
3. In Fig. 3A and 5J, Western blots and quantitation for p-FOXO3a/FOXO3a are problematic as FOXO3a protein levels were clearly downregulated by DUSP22 siRNA. Does DUSP22 induce the protein stability of FOXO3A?
4. The role of FOXO3a in DUSP22-mediated Atrogin-1 and MuRF-1 gene induction in skeletal muscle cells (as depicted in Fig. 9O) should be demonstrated using FOXO3a siRNA.
5. The Western blots for Atrogin-1 is either of poor quality or not convincing for Fig. 3E, 5M, and 6I.
6. The Western blot data for DUSP22 are missing (for Fig. 3I, 4J, 5B, 5J, 5M, 6I, 7G, 9E) or of poor quality (Fig. 7B).
7. The caveat on the specificity of the DUSP22 inhibitor BML-260 should be discussed.
8. In page 6, the sentence "FOXO3a signaling in myotubes was decreased by Dex and increased by DUSP22 knockdown (Figure 2G-H)" is confusing and seems to be misplaced.
9. Fig. 5P should be 5N.
10. The manuscript requires better proofreading for the Introduction section.
11. CSA and TA should be spelled out at least once in the Introduction or Results section.

Referee #2 (Comments on Novelty/Model System for Author):

Technical quality - Medium because some additional details are required as per my comments.

Novelty - high because this is the first evidence of a role for DUSP22 in muscle atrophy and the study is thorough including studies with a small molecule inhibitor of DUSP22.

Medical impact - medium because this is still at a very early stage of drug discovery.

Model systems adequate - mice, mouse and human muscle cells were appropriate.

Referee #2 (Remarks for Author):

Summary

Cachexia and sarcopenia remain important, common clinical problems for which there is no approved pharmacological therapy. In this manuscript, Lee et al report a collection of findings which suggest the DUSP22 dual activity phosphatase as a potential drug target for muscle wasting conditions. They report increased expression of DUSP22 in muscle in a few catabolic conditions and show using gain and loss of function (including a DUSP22 inhibitor) approaches in vivo and in muscle cells that DUSP22 can modulate muscle mass and myofiber size. In addition, they provide evidence to suggest that DUSP22 exerts its effects, at least in part, through promoting JNK activity which in turn activates FoxO3 signaling.

General comments

- 1) Overall, this is a fairly thorough study describing a novel role for DUSP22 in regulation of skeletal muscle mass. The studies appear to be generally well executed, but some issues need to be addressed as indicated below.
- 2) One of the exciting findings is the identification of BML-260, a DUSP22 inhibitor and its efficacy. However, there is almost no background information provided for this molecule. The authors provide one reference to the compound (ref 31), but the reference is to a paper that is screening for inhibitors of the PTP-1B tyrosine phosphatase, not for DUSP22. The authors do not indicate where the compound was obtained. Importantly, there is no data describing the potency or specificity of BML-260 for DUSP22. What is the K_i or IC_{50} for DUSP22? Does it inhibit a panel of other phosphatases? How was the in vivo dose of 5mg/kg chosen and is there any evidence that this achieves an appropriate concentration in muscle and engages the target? This information is important for getting some idea as to whether the inhibitor is acting on-target in the studies.
- 3) Similarly, the specificity of the siRNA oligos used in the study is not addressed. It is standard practice to test two different siRNA oligos as a minimal test for specificity. The sequence of the siRNA oligos (including the control siRNA) should be stated in the methods or supplementary table.
- 4) The authors argue that DUSP22 exerts its actions via activation of JNK kinase. Are the effects of DUSP22 silencing or inhibition lost upon inhibition or knockdown of JNK kinase?
- 5) Based on what we know about DUSP22, what are the potential adverse effects of administering an inhibitor long term as would probably be required for most muscle wasting conditions? What is the phenotype of the mouse KO of DUSP22? This point should be addressed in the discussion. See Chen M et al, BMC Med 2023 Feb 10;21(1):46. doi: 10.1186/s12916-023-02745-6 as a possible example.

Specific comments

- 1) Pg 6 - para 3 - text refers to Figures 2G-H, but this is mislabelled.
- 2) Pg 6 last para and elsewhere - the authors sometimes refer to fast type myosin and sometimes specific isoforms. For non-experts, it would be preferable to use both e.g. fast type myosin MYH# in the text and figure legends.
- 3) Pg 7 para 1 and Figure 3I-J. The text indicates that the increase in MYH2 expression upon DUSP22 knockdown did not reach statistical significance, but figure 3J indicates otherwise.
- 4) Pg 6, Fig 3A and Pg 8, para 1 and Fig 5J-K. The authors indicate that siDUSP22 has effects on FoxO3 activity - expressed as ratio of pFoxO3/FoxO3 - but the main effect seems to be on lowering total FoxO3 levels. The authors should state this.
- 5) Pg 9, para 2, Figure 7P, Q - show also the effect of BML on Type 1 fibers.
- 6) Pg 11, para 1 and Fig 9D. The effect of BML-260 on muscle mass was statistically not significant. However, $n=3$ is far too small for this type of study.
- 7) Pg 13, para 1. I did not understand the relationship of the color of rhodamine and how this relates to the potential of BML-260 to serve as a diagnostic biomarker.
- 8) Pg 16, 17 Methods - myotube diameter - myotubes are not uniform in diameter along the length of the fiber. How did they select where to measure myotube diameter? How did they select the myotubes for counting nuclei to avoid bias? Were random fields chosen? Was the observer blind to treatment conditions?
- 9) Pg 18, para 1 -Western blotting - a few details should be given which influences accuracy of quantification - What was the method of detection of secondary antibody - X ray film (poor linearity) or chemiluminescence or other?
- 10) Pg 18, para 3 - when were siRNA oligos transfected into muscle cells? If in myoblasts at what confluence? if in myotubes on what day of differentiation?
- 11) Pg 18, para 4 - At what confluence was the plasmid transfection of myoblasts done? What was the transfection efficiency? The transfected cells were subjected to selection. Were clones obtained and amplified? Were the cells differentiated into myotubes?
- 12) Pg 19 para 3, what volume was used for the TA injections of siRNAs?
- 13) Pg 27, Fig 5 legend - it is stated that Western blot signals were normalized to GAPDH, but were the phospho-proteins not mainly normalized to the particular total protein?
- 14) Figure 3I - the columns are not labelled.
- 15) Figure 5A - the schematic says that mice were given 25 mg/kg but the methods says 15 mg/kg.
- 16) Figure 6B - are the curves significantly different?
- 17) Figure 5J and 6I - blots appear overexposed.
- 18) Figure 7J, 8B, 8D, 8E, 8H, 8I - many of the labels are too small and illegible. Please increase in size. Also is fold change in Figure 8C up or down? The labels below the panels in Figure 8F are too small and do not seem necessary.

19) Figure 8O - it appears that DUSP22 activation in muscle can inhibit Akt which is not the case.

Referee #3 (Comments on Novelty/Model System for Author):

Lee SH et al. suggested targeting DUSP22 attenuates muscle atrophy via DUSP-JNK-FOXO3a signaling. However the relation between DUSP22-and JNK was already reported. The mechanism is not new. The novelty of this study could be the chemical inhibitor of DUSP22 as muscle atrophy drug. Thus, I would recommend this manuscript should be re-written focusing on the effect of BML-260.

Referee #3 (Remarks for Author):

Lee SH et al. suggested targeting DUSP22 attenuates muscle atrophy via DUSP-JNK-FOXO3a signaling. However the relation between DUSP22-and JNK was already reported. The mechanism is not new. The novelty of this study could be the chemical inhibitor of DUSP22 as muscle atrophy drug.

1. Independence of AKT phosphorylation was only shown in Fig. 3A

Thus, more evidence is needed.

2. Silence or overexpression of DUSP22 showed the expected data. Because DUSP-JNK-FOXO pathway was already reported.

3. The upregulation of DUSP22 was now observed in dex-treated mice in Fig. 1C. do you know why?

4. In Fig. 5A and 7A, the effect of siDUSP22 injection on muscle strength or physical performance should be suggested.

5. In Fig. 1B, expression of Atrogin-1 is needed.

6. In Fig. 9, measurement of myotube diameter is performed in MHC-IF stained cells for clarity.

Reference: EMM-2024-19782

“Targeting phosphatase DUSP22 ameliorates skeletal muscle wasting via Akt independent JNK-FOXO3a repression”

By Lee *et al.*

Response to reviewers' comments

We wish to thank the reviewers for their valuable feedback on our manuscript. We have tried our best to address the comments raised by the reviewers, which has significantly enhanced the quality of our submission. Our response to each comment is provided below. The changes to the manuscript text are shown using red font.

Reviewer 1

1) **Comment:** “In Fig. 1, only mRNA expression levels were studied. Protein levels for some of the critical genes may need to be analyzed as DUSP22 is a protein phosphatase that regulates post-translational modification of target proteins.”

Response and manuscript modification:

Thank you very much for this comment. In the revised manuscript, we have added western blots to measure the protein levels of some of the critical genes shown in Fig. 1. The following genes were analyzed: atrogen-1 and MuRF-1 (linked to FoxO3a-related signaling), UBR2 (linked to the ubiquitin-proteasome system), and DUSP22.

It was observed that DUSP22 overexpression increased the expression of atrogen-1, MuRF-1 and UBR2:

Figure 1: Western blot analysis of DUSP22, MuRF-1, atrogin-1 and UBR2 levels C2C12 myoblasts transfected with a DUSP22 CRISPR activation plasmid (DUSP22endoOE) or control plasmid (CONendoOE) after 96 h culture in DM. GAPDH was used for the normalization of protein levels. *= $p < 0.05$, **= $p < 0.01$ ****= $p < 0.0001$ indicate significantly increased or decreased.

This new western blot analysis has been added to the revised manuscript as Supplementary Figure 2.

2) **Comment:** “In Figure 1J, UBR2 mRNA levels were upregulated by DUSP22 in skeletal muscle cells. However, UBR2 proteins have been reported to be downregulated by DUSP22 in T cells. The authors should address this issue by performing western blotting or by discussion.”

Response and manuscript modification:

Thank you for pointing out this issue with our experimental analysis of UBR2 expression. As mentioned in our response to comment 1, above, we have performed UBR2 western blotting for the manuscript revision. It was found that DUSP22 overexpression increased the protein levels of UBR2 (inserted as Supplementary Figure 2).

As mentioned by the Reviewer, a previous study showed that DUSP22 downregulates UBR2 proteins in T cells [1]. To address this discrepancy with our results, the following text has been added to the Discussion section of the revised manuscript:

‘It was observed that DUSP22 overexpression increased the expression of UBR2 in murine myotubes. Interestingly, a previous study reported that DUSP22 downregulates UBR2 proteins in human T lymphocytes [1]. There are a number of possible explanations for this difference in the research findings. For example, the MAPK pathway has been show to maintain UBR2 expression, and DUSP22 is known to activate MAPK signaling [2, 3]. MAPK signaling status was not assessed in the human T lymphocytes, meaning the effect of DUSP22 activation on this pathway and its possible role in UBR2 expression in lymphocytes could not be ascertained. In skeletal muscle tissue, UBR2 upregulation has been previously linked to muscle atrophy and UBR2 expression was shown to be increased by FOXO3a in cancer-related muscle wasting [4, 5]. The results presented herein show that DUSP22 overexpression upregulated FOXO3a in muscle cells, which could explain the observed increase in UBR2 expression. Additionally, in skeletal muscle

tissue additional signaling mechanisms, such as the N-end rule pathway, have been shown to maintain UBR2 high expression in conditions of muscle atrophy (where UBR2 functions as a substrate recognition component) [6, 7]. It is possible that DUSP22 overexpression may also upregulate this pathway in muscle cells to increase UBR2 expression.’

This text has been inserted into the Discussion section of the revised manuscript.

3) **Comment:** “In Fig. 3A and 5J, Western blots and quantitation for p-FOXO3a/FOXO3a are problematic as FOXO3a protein levels were clearly downregulated by DUSP22 siRNA. Does DUSP22 induce the protein stability of FOXO3A?”

Response and Manuscript modification:

We are grateful to Reviewer 1 for this comment concerning our western blotting and the quantification of p-FOXO3a/FOXO3a in Fig 3A and 5J. We have revised the manuscript text to state that DUSP22 gene knockdown downregulates FOXO3a protein levels, both in vitro (Figure 3A) and in vivo (Figure 5J). The following revised sentences have been added to the Results section of the manuscript:

‘Knockdown in normal and Dex treated myotubes decreased FOXO3a protein levels, leading to an increase in the overall ratio of phosphorylated FOXO3a: total FOXO3a (Figure 3A-B).’

And:

‘Western blotting showed that FOXO3a protein levels increased after Dex treatment and were reduced by DUSP22 knockdown, which increased the overall ratio of phosphorylated FOXO3a: total FOXO3a (Figure 5J-K).’

4) **Comment:** “The role of FOXO3a in DUSP22-mediated Atrogin-1 and MuRF-1 gene induction in skeletal muscle cells (as depicted in Fig. 9O) should be demonstrated using FOXO3a siRNA.”

Response and manuscript modification:

Thank you for this advice concerning the role of FOXO3a. For the manuscript revision, we have investigated the effect of FOXO3a gene knockdown on Atrogin-1 and MuRF-1 expression in the Dex model of myotube atrophy. It was observed that FOXO3a knockdown reduced the expression of MuRF-1 and Atrogin-1 in the Dex treated myotubes:

Figure 2: Western blot analysis of FOXO3a, FOXO3a phosphorylation, atrogin-1 and MuRF-1 in C2C12 myotubes treated with control or FOXO3a siRNA. B) Quantification of FOXO3a phosphorylation, FOXO3a, atrogin-1 and MuRF-1 protein levels compared to GAPDH. *= $p < 0.05$ and **= $p < 0.01$ indicate significantly increased or decreased.

These results have been added to the revised manuscript as Supplementary Figure 3.

5) **Comment:** “The Western blots for Atrogin-1 is either of poor quality or not convincing for Fig. 3E, 5M, and 6I.”

Response and manuscript modification:

We apologize for the poor quality or not convincing western blots for Atrogin-1. For the manuscript revision, the Atrogin-1 western blots have been repeated and the following new results were obtained:

Reference: EMM-2024-19782

“Targeting phosphatase DUSP22 ameliorates skeletal muscle wasting via Akt independent JNK-FOXO3a repression”

By Lee *et al.*

Figure 3E:

Figure 5M: The atrogin-1 western has been removed from the revised manuscript, because it was not possible to get a clear signal in the DUSP22 in vivo gene knockdown condition.

Figure 6I:

6) **Comment:** “The Western blot data for DUSP22 are missing (for Fig. 3I, 4J, 5B, 5J, 5M, 6I, 7G, 9E) or of poor quality (Fig. 7B).”

Response and manuscript modification:

We are sorry for these oversights in the DUSP22 Western blotting. For the manuscript revision, DUSP22 has been added to the Western blots in Figures 3I, 4J, 5B, 5J, 5M, 6I, 7G, and 9E. In addition, we have repeated the DUSP22 Western blot in Figure 7B. The following new DUSP22 Western blotting data was obtained:

For Figure 3I:

For Figure 4J:

For Figure 5B:

Reference: EMM-2024-19782

“Targeting phosphatase DUSP22 ameliorates skeletal muscle wasting via Akt independent JNK-FOXO3a repression”

By Lee *et al.*

For Figure 5J (same samples as used in Figure 5M):

For Figure 6I:

For Figure 7G:

Figure 9E (Figure 9F in the revised manuscript):

7) **Comment:** “The caveat on the specificity of the DUSP22 inhibitor BML-260 should be discussed.”

Response and manuscript modification:

Thank you for this comment concerning the specificity of BML-260 for DUSP22. In the revised manuscript, we have added the following text that discusses the caveats of BML-260 as a specific inhibitor of DUSP22:

‘This study has shown that the small molecule BML-260 inhibits DUSP22 activity and prevents skeletal muscle atrophy. Gene knockdown of DUSP22 produced similar, beneficial effects in muscle atrophy models. DUSP22 knockdown or BML-260 treatment also downregulated FOXO3a-related atroгене expression without stimulating Akt activity. BML-260 was originally identified in a screen of rhodanine-based compounds for DUSP22 inhibition [8]. BML-260 inhibitory activity was shown to be specific for DUSP22, because there was no activity against the related atypical DUSP, VH1-related (VHR) phosphatase [8]. However, other non-specific effects of BML-260 cannot be discounted. For example, Feng, et al., reported that BML-260 treatment upregulates the expression of uncoupling protein-1 (UCP-1) in adipocytes [9]. This effect was found to be independent of DUSP22 activity and the precise mechanism could not be fully elucidated. It should be noted that while UCP-1 upregulation may be beneficial for treating obesity via mitochondrial uncoupling, reduced ATP production and heat generation in adipocytes, this may not be the case for sarcopenia, which should benefit from more efficient mitochondrial function and ATP generation in skeletal muscle [10]. In addition, increased physical activity has been shown to decrease UCP-1 expression [11]. It is also interesting to observe that in the present study DUSP22 overexpression in myotubes upregulated the expression of the uncoupling protein, UCP-3 (Figure 1H).

Rhodanine-based compounds have been described as problematic for medicinal chemistry due to non-specificity, aggregation and potential toxicity [12]. However, Epalrestat, a rhodanine-based drug for diabetic neuropathy, is safe, well-tolerated, and the

only commercially available inhibitor of aldose reductase [13]. More recently, Montañó, et al., demonstrated that rhodanine compounds could be developed as antibiotics that specifically target bacterial thymidylate kinase [14]. Progress in the medicinal chemistry of rhodanine derivative scaffolds also suggest that these could be viewed as privileged structures that may be exploited for future rational design and discovery [15]. Moreover, Mendgen, et al., assessed the biological activity of 163 rhodanine or rhodanine-like compounds and found that only 2 compounds displayed potential aggregation, with the α -carbon attachment decoration also being important in conferring selectivity and reducing promiscuous binding [16, 17]. Therefore, with a cautious approach to structural modifications and biological assays for activity, it may be possible to further develop BML-260 as a clinically relevant DUSP22 inhibitor.’

This new text has been added to the Discussion section of the revised manuscript.

8) **Comment:** “In page 6, the sentence "FOXO3a signaling in myotubes was decreased by Dex and increased by DUSP22 knockdown (Figure 2G-H)' is confusing and seems to be misplaced.”

Response and manuscript modification:

We apologize for this misplaced sentence. The sentence has been deleted from the revised manuscript.

9) **Comment:** “Fig. 5P should be 5N.”

Response and manuscript modification:

We apologize for this error. Fig. 5P has been correctly labelled as 5N in the revised manuscript.

10) **Comment:** “The manuscript requires better proofreading for the Introduction section.”

Response and manuscript modification:

Thank you for this advice. In the revised manuscript, a native English speaker has proofread the Introduction section. Changes to the text are indicated using red font.

11) **Comment:** “CSA and TA should be spelled out at least once in the Introduction or Results section.”

Response and manuscript modification:

We are grateful for this feedback. In the Results of the revised manuscript, CSA and TA have been spelled out at their first use (in the section ‘DUSP22 knockdown in skeletal muscle ameliorates wasting’ for CSA, and the section ‘DUSP22 is upregulated in skeletal muscle wasting and overexpression disrupts myogenesis’ for TA).

We wish to express our gratitude to Reviewer 1, whose insightful comments have enhanced the quality of our manuscript.

Reviewer 2:

General comments

1) **Comment:** “One of the exciting findings is the identification of BML-260, a DUSP22 inhibitor and its efficacy. However, there is almost no background information provided for this molecule. The authors provide one reference to the compound (ref 31), but the reference is to a paper that is screening for inhibitors of the PTP-1B tyrosine phosphatase, not for DUSP22. The authors do not indicate where the compound was obtained.”

Response and manuscript modification:

We apologize for not including the original reference describing BML-260 as a DUSP22 inhibitor. In the revised manuscript, reference 31 has been removed and replaced with the original reference describing the characterization of BML-260 (Cutshall, *et al.*, Rhodanine derivatives as inhibitors of JSP-1. *Bioorg Med Chem Lett.* 2005 Jul 15;15(14):3374-9.).

We have also added new text to provide background information for the BML-260 compound, as follows:

‘BML-260 was originally characterized by Cutshall, *et al.*, from a screen of rhodanine derivatives for DUSP22 inhibition using an epidermal growth factor receptor peptide P³²-

Reference: EMM-2024-19782

“Targeting phosphatase DUSP22 ameliorates skeletal muscle wasting via Akt independent JNK-FOXO3a repression”

By Lee *et al.*

based assay [8]. BML-260 was found to be a competitive inhibitor of DUSP22 with an IC₅₀ in the low micromolar range. BML-260 specificity was demonstrated by showing no inhibitory effect against VH1-related (VHR) phosphatase, which is a related, atypical DUSP [8].’

(inserted in the Results section ‘DUSP22 pharmacological targeting prevents myotube atrophy’ of the revised manuscript).

We have also updated the Materials and Methods ‘Cell culture’ section to include the BML-260 supplier, as follows:

‘BML-260 was purchased from Santa Cruz Biotechnology, TX, USA (cat. no. sc-223822).’

2) **Comment:** “Importantly, there is no data describing the potency or specificity of BML-260 for DUSP22. What is the Ki or IC50 for DUSP22? Does it inhibit a panel of other phosphatases? How was the in vivo dose of 5mg/kg chosen and is there any evidence that this achieves an appropriate concentration in muscle and engages the target? This information is important for getting some idea as to whether the inhibitor is acting on-target in the studies.”

Response and manuscript modification:

Thank you very much for these comments. The original publication describing BML-260 reported an IC₅₀ of 18 μM. This was determined using an epidermal growth factor receptor (EGF-R) peptide P³²-based assay [8]. As mentioned above, BML-260 did not show inhibitory activity against the related atypical DUSP, VH1-related (VHR) phosphatase [8].

For the manuscript revision, we have carried out an additional BML-260 inhibition study using the EnzChek™ Phosphatase Assay Kit, which is based on the 6,8-difluoro-4-methylumbelliferyl phosphate (DiFMUP) substrate (Invitrogen). Using this assay, it was observed that BML-260 dose-dependently inhibited DUSP22 activity with an IC₅₀ of 54 μM:

Figure 3: Phosphatase activity assay for DUSP22 and inhibition by BML-260. The assay is based on the 6,8-difluoro-4-methylumbelliferyl phosphate (DiFMUP) reagent and GST-tagged-DUSP22.

Inserted into the revised manuscript as Supplementary Figure 5A.

To gain further insight into the specificity of BML-260 for DUSP22, additional molecular docking studies were carried out for BML-260 bound to DUSP22, DUSP22 with an active site mutation (at C88S) and DUSP15, which is a DUSP member with the highest homology with DUSP22 (as assessed by sequence alignment) [18]. It was observed that only DUSP22 showed BML-260 binding to the active site region:

Figure 4: Phosphatase assay CB-Dock2 molecular docking analysis of BML-260 binding to the active site of DUSP22, DUSP22 with an active site mutation (at C88S), and DUSP15, which is a DUSP member with the highest homology with DUSP22 (as assessed by sequence alignment)

[18]). DUSP2 (C88S mutation) and DUSP15 showed altered BML-260 binding. BML-260 bound DUSP15 at the interface of the A and B chain, distinct from the predicted active site at position 85.

Inserted into the revised manuscript as Supplementary Figure 5B.

The 5 mg/kg in vivo dose was selected based on a previous study that carried out bilateral injection of BML-260 into fat pads of male 8 week-old C57BL/6J mice [9]. This information has been added to the Methods section ‘Dex treatment model of skeletal muscle atrophy and DUSP22 knockdown’ in the revised manuscript.

Concerning BML-260 pharmacokinetics, a previous study reported that, based on the HPLC results, BML-260 is predicted to be stable at pH values within normal physiological conditions (pH range 3 to 7.4) [9]. In addition, we utilized a company to carry out pharmacokinetic analysis of BML-260 (Chaon, Yongin-si, Republic of Korea). BML-260 was found to be detectable in the plasma immediately after intraperitoneal injection. After an approximately 20 min delay, peak detection of BML-260 was then observed in the skeletal muscle tissue (based on analysis of the tibialis anterior):

Reference: EMM-2024-19782

“Targeting phosphatase DUSP22 ameliorates skeletal muscle wasting via Akt independent JNK-FOXO3a repression”

By Lee *et al.*

Figure 5: Pharmacokinetic analysis of BML-260 in 6 weeks old male C57BL/6J mice (5 mg/kg delivered via IP injection). BML-260 detection was assessed in the plasma (A) and tibialis anterior muscle (B).

The pharmacokinetic data has been inserted into the revised manuscript as Supplementary Figure 9.

3) **Comment:** “Similarly, the specificity of the siRNA oligos used in the study is not addressed. It is standard practice to test two different siRNA oligos as a minimal test for specificity. The sequence of the siRNA oligos (including the control siRNA) should be stated in the methods or supplementary table.”

Response and manuscript modification:

We are very grateful for this comment concerning the siRNA oligos used in our study. In the revised manuscript, we have repeated the DUSP22 knockdown experiment using a different DUSP22 siRNA. In addition, we confirmed that the second siRNA oligo can suppress expression of the key muscle atrophy-related gene, atrogin-1. It was observed that the second DUSP22 siRNA downregulated both DUSP22 and atrogin-1 expression in C2C12 myotubes:

Figure 6: Western blot analysis of atrogin-1 and DUSP22 in C2C12 myotubes treated with control or two distinct DUSP22 siRNAs (termed siDUSP22-1 and siDUSP22-2).

Reference: EMM-2024-19782

“Targeting phosphatase DUSP22 ameliorates skeletal muscle wasting via Akt independent JNK-FOXO3a repression”

By Lee *et al.*

The pharmacokinetic data has been inserted into the revised manuscript as Supplementary Figure 4.

The sequence of the second DUSP22 have been included in the revised manuscript as Supplementary Table 4. We were informed by the company supplier (Thermo Fisher Scientific) that the sequence of the control siRNA could not be provided.

4) **Comment:** “The authors argue that DUSP22 exerts its actions via activation of JNK kinase. Are the effects of DUSP22 silencing or inhibition lost upon inhibition or knockdown of JNK kinase?”

Response and manuscript modification:

Thank you for this advice concerning DUSP22 and JNK activation. For the manuscript revision, we have assessed the effect of JNK inhibition on myotube atrophy using the small molecule ATP-competitive inhibitor, SP600125, which has a >20-fold selectivity versus a range of kinases and enzymes tested [19]. In the presence of JNK inhibition, DUSP22 inhibition by BML-260 did not have any significant effect on myotube atrophy compared to BML-260 treatment alone:

Figure 7: A) Fast myosin (MYH2) immunostaining of C2C12 myoblasts cultured as follows: (1) DM for 120 h (vehicle alone); (2) DM for 96 h and DM plus 10 µM Dex for 24 h; (3) DM for 96 h and DM plus 10 µM Dex and 12.5 µM BML-260 for 24 h; (4) DM for 96 h and DM plus 10 µM Dex and 20 µM SP600125 for 24 h; (5) DM for 96 h and DM plus 10 µM Dex and 12.5 µM BML-260 plus 20 µM SP600125 for 24 h. B) Mean myotube diameter. ****= $p < 0.0001$ indicate significantly increased or decreased.

This data has been inserted into the revised manuscript as Supplementary Figure 6.

5) **Comment:** “Based on what we know about DUSP22, what are the potential adverse effects of administering an inhibitor long term as would probably be required for most muscle wasting conditions? What is the phenotype of the mouse KO of DUSP22? This point should be addressed in the discussion. See Chen M et al, BMC Med 2023 Feb 10;21(1):46. doi: 10.1186/s12916-023-02745-6 as a possible example.”

Response and manuscript modification:

We appreciate this advice concerning the long-term administration of BML-260 and the phenotype of DUSP22 knockout mice. The previous study by Chen *et al.*, observed that DUSP22 knockout mice spontaneously developed syndesmophytes (inflammatory growths in the spinal vertebrae and may produce joint fusion) [20]. In addition, Li *et al.*, reported that aged JKAP-knockout mice can spontaneously develop inflammation and autoimmunity, with higher serum levels of anti-nuclear antibodies and anti-dsDNA, and expansion of white pulps in spleen [21]. Consequently, small molecule inhibition by BML-260 should, ideally, lower DUSP22 activity without recapitulating the immune-related phenotype observed in knockout mice. It should be noted DUSP22 has additional functions in cells that are independent of its enzyme activity, such as a scaffold protein for JNK regulation by signaling proteins such as apoptosis signal-regulating kinase 1 (ASK1) [3]. These non-enzymatic functions should still be preserved in the presence of BML-260 treatment.

For the manuscript revision, we have investigated the effect of 6 weeks BML-260 treatment in 15 months-old mice on liver, heart and kidney mass (three major organs assessed for potential drug toxicity [22]). It was observed that organ mass showed no significant change after the treatment period:

Figure 7: A) Photographs of the kidneys, liver, and heart after 6 weeks treatment with 5 mg/kg in 15 month-old mice. B) Kidney, liver, and heart mass. ns=not statistically significant.

This result has been inserted into the revised manuscript as Supplementary Figure 11. The possible toxic effects of DUSP22 treatment are also mentioned in the revised Discussion section.

Specific comments

1) **Comment:** “Pg 6 - para 3 - text refers to Figures 2G-H, but this is mislabelled.”

Response and manuscript modification:

We apologize for this error. The text referring to the mislabeled Figures has been deleted from the revised manuscript.

2) **Comment:** “Pg 6 last para and elsewhere - the authors sometimes refer to fast type myosin and sometimes specific isoforms. For non-experts, it would be preferable to use both e.g. fast type myosin MYH# in the text and figure legends.”

Response and manuscript modification:

We are grateful for this advice. In the revised manuscript, the following text is used to refer to the myosin isoforms: ‘fast type myosin (MYH isoform)’.

3) **Comment:** “Pg 7 para 1 and Figure 3I-J. The text indicates that the increase in MYH2 expression upon DUSP22 knockdown did not reach statistical significance, but figure 3J indicates otherwise.”

Response and manuscript modification:

We apologize for this error in the Results section for DUSP22 knockdown. The incorrect graph was used for MYH2 when constructing the final version of Figure 3J. This graph has been corrected in the revised manuscript ($p=0.051$).

4) **Comment:** “Pg 6, Fig 3A and Pg 8, para 1 and Fig 5J-K. The authors indicate that siDUSP22 has effects on FoxO3 activity - expressed as ratio of pFoxO3/FoxO3 - but the main effect seems to be on lowering total FoxO3 levels. The authors should state this.”

Response and manuscript modification:

We are sorry for this oversight in reporting the effect of siDUSP22 on FoxO3 activity. For the manuscript revision, the text in Pg 6, Pg8, Fig 3A, and Fig 5J-K has been revised to state that the main effect of siDUSP22 is lowering total FoxO3 levels. The following text has been added to the corresponding sections of the Results section:

“Knockdown in normal and Dex treated myotubes decreased FOXO3a protein levels, leading to an increase in the overall ratio of phosphorylated FOXO3a: total FOXO3a (Figure 3A-B).”

And:

“Western blotting showed that FOXO3a protein levels increased after Dex treatment and were reduced by DUSP22 knockdown, which increased the overall ratio of phosphorylated FOXO3a: total FOXO3a (Figure 5J-K).”

5) **Comment:** “Pg 9, para 2, Figure 7P, Q - show also the effect of BML on Type 1 fibers.”

Response and manuscript modification:

Thank you for this comment. We have carried out immunostaining for Type 1 fibers in the gastrocnemius muscle of aged mice treated with BML-260. It was observed that the CSA of Type 1 fibers did not significantly change in response to aging or BML-260 treatment:

Figure 8: A) Type 1 myofiber and laminin staining in the TA and gastrocnemius muscles (5M=5 months-old). Scale bar=100 μ m. B) CSA of the type 1 myofibers.

This new data has been added to the revised manuscript as figures Supplementary Figure 10.

6) **Comment:** “Pg 11, para 1 and Fig 9D. The effect of BML-260 on muscle mass was statistically not significant. However, n=3 is far too small for this type of study.”

Response and manuscript modification:

We appreciate this feedback concerning our immobilization model and measurement of muscle mass. For the manuscript revision, we have repeated the immobilization study to increase the n number from n=3 to n=10 to analyze muscle mass. It was observed that BML-260 treatment increased TA muscle mass in the immobilization model and achieved statistical significance:

This new result has been added to the revised manuscript as Figure 9E.

7) **Comment:** “Pg 13, para 1. I did not understand the relationship of the color of rhodamine and how this relates to the potential of BML-260 to serve as a diagnostic biomarker.”

Response and manuscript modification:

Thank you for pointing out this issue with discussion of BML-260 as a diagnostic marker. Our reasoning was based on the ability of rhodamine compounds to change color upon metal ion binding, which may be useful for tissue sample from patients with metal accumulation disorders, such as Wilson’s disease, or aging related iron accumulation [23-25]. However, with hindsight we believe that this discussion of BML-260 as a diagnostic marker is beyond the scope of our manuscript, because metal ion binding to BML-260 has not been assessed. Consequently, the discussion of BML-260 as a diagnostic marker has been deleted from the revised manuscript.

8) **Comment:** “Pg 16, 17 Methods - myotube diameter - myotubes are not uniform in diameter along the length of the fiber. How did they select where to measure myotube diameter? How did they select the myotubes for counting nuclei to avoid bias? Were random fields chosen? Was the observer blind to treatment conditions?”

Response and manuscript modification:

We apologize for these omissions in describing the experimental methods for measuring myotube diameter. In the revised manuscript, these methods have been expanded to explain more clearly the measurement of myotube diameter. The following text has been inserted into the revised Materials and Methods section:

“Myotubes were primarily examined in the central area of each well, as opposed to randomly selected fields. The mid-point of the myotube was used to measure the diameter. For nuclear counts, we chose myotubes containing three or more nuclei. The observer was aware of the treatment conditions.”

9) **Comment:** “Pg 18, para 1 -Western blotting - a few details should be given which influences accuracy of quantification - What was the method of detection of secondary antibody - X ray film (poor linearity) or chemiluminescence or other?”

Response and manuscript modification:

We are sorry for not fully describing the Western blotting detection method. Chemiluminescence was used to detect the secondary antibody. These details have been inserted into the Western blotting section of the revised Materials and Methods.

10) **Comment:** “Pg 18, para 3 - when were siRNA oligos transfected into muscle cells? If in myoblasts at what confluence? if in myotubes on what day of differentiation?”

Response and manuscript modification:

Thank you for pointing out these oversights in our description of the transfection protocol. siRNA transfection was performed on day 3 of myoblast differentiation into myotubes. Accordingly, we have included this information in the revised ‘siRNA-mediated gene knockdown in myotubes’ section of the Materials and Methods.

11) **Comment:** “Pg 18, para 4 - At what confluence was the plasmid transfection of myoblasts done? What was the transfection efficiency? The transfected cells were subjected to selection. Were clones obtained and amplified? Were the cells differentiated into myotubes?”

Response and manuscript modification:

We apologize for not fully explaining the transfection protocol for the plasmid. Transfection was performed at 50% confluence. The transfection efficiency was approximately 5%. Selection was conducted using puromycin dihydrochloride (2 µg/mL), hygromycin B (200 µg/mL), and blasticidin S HCl (20 µg/mL). After approximately two weeks of selection, the selected myoblasts were differentiated into myotubes.

These details have been added to then ‘CRISPR/Cas9-mediated gene overexpression’ section of the revised Materials and Methods.

12) **Comment:** “Pg 19 para 3, what volume was used for the TA injections of siRNAs?”

Response and manuscript modification:

We apologize for this oversight. The volume for the TA injections of the siRNAs was 50 µL. This has been added to the ‘Dex treatment model of skeletal muscle atrophy and DUSP22 knockdown’ section of the revised Materials and Methods.

13) **Comment:** “Pg 27, Fig 5 legend - it is stated that Western blot signals were normalized to GAPDH, but were the phospho-proteins not mainly normalized to the particular total protein?”

Response and manuscript modification:

Thank you for pointing out this discrepancy in our description of the western blot data. The phospho-proteins were also normalized to the particular protein of interest in the Fig 5. The Fig 5 legend in the revised manuscript has been updated to describe this analysis.

14) **Comment:** “Figure 3I - the columns are not labelled.”

Response and manuscript modification:

We apologize for this error. The columns in Figure 3I have been labelled in the revised manuscript.

15) **Comment:** “Figure 5A - the schematic says that mice were given 25 mg/kg but the methods says 15 mg/kg.”

Response and manuscript modification:

We apologize for this inconsistency in reporting the dose of dexamethasone administered to the C57BL/6 young mice. The correct dose for the BML-260 treatment study was 15 mg/kg dexamethasone. The correct dose for the siRNA study was 25 mg/kg. The ‘Dex treatment model of skeletal muscle atrophy and DUSP22 knockdown’ section of the Materials and Methods in the revised manuscript has been updated to correctly state these treatment doses.

16) **Comment:** “Figure 6B - are the curves significantly different?”

Response and manuscript modification:

Thank you for pointing out is issue with Figure 6B. We have re-checked the original data and found that there was no statistical difference between the curves in Figure 6B.

17) **Comment:** “Figure 5J and 6I - blots appear overexposed.”

Response and manuscript modification:

We apologize for the overexposed blots in Figure 5J and 6I. In the revised manuscript, the MurF-1 and atrogen-1 blots in Figure 6I have been updated and DUSP22 included. For Figure 5J, the exposure of the blots have been adjusted to try and reduce the level of exposure.

18) **Comment:** “Figure 7J, 8B, 8D, 8E, 8H, 8I - many of the labels are too small and illegible. Please increase in size. Also is fold change in Figure 8C up or down? The labels below the panels in Figure 8F are too small and do not seem necessary.”

Response and manuscript modification:

Thank you for these comments and we are sorry for the errors in the labelling for Figures 7J, 8B, 8D, 8E, 8H, 8I. In the revised manuscript, these labels have been resized to increase legibility. For Figure 8C, the control was set as the baseline value of 1. A value of 2 would indicate a 2-fold increase, and 0.5 would indicates a 2-fold decrease. The Figure legend for 8C has been revised to explain the fold change more clearly. In addition, the labels below the panels in Figure 8F have been removed. To increase the fold size of the labels, Figures 8G, H and I have been moved to Supplementary Figure 12.

19) **Comment:** “Figure 8O - it appears that DUSP22 activation in muscle can inhibit Akt which is not the case.”

Response and manuscript modification:

Reference: EMM-2024-19782

“Targeting phosphatase DUSP22 ameliorates skeletal muscle wasting via Akt independent JNK-FOXO3a repression”

By Lee *et al.*

We apologize for this error in Figure 8O. In the revised manuscript, this diagram has been revised and the interaction between DUSP22 activation and Akt has been removed.

We wish to express our gratitude to Reviewer 2, whose insightful comments have enhanced the quality of our manuscript.

Reviewer 3:

(Remarks for Author):

Lee SH *et al.* suggested targeting DUSP22 attenuates muscle atrophy via DUSP-JNK-FOXO3a signaling. However, the relation between DUSP22-and JNK was already reported. The mechanism is not new. The novelty of this study could be the chemical inhibitor of DUSP22 as muscle atrophy drug.

Response and manuscript modification:

Thank you very much for these suggestions concerning the novelty of our manuscript. We have updated parts of the manuscript text to aim to emphasize the novelty of the DUSP22 chemical inhibitor as a muscle atrophy drug. Accordingly, the following text has been added to the revised Discussion of the manuscript:

‘An interesting aspect of the current study is the discovery of DUSP22 as an upstream regulator of the relationship between JNK and FOXO3a, along with the availability of BML-260 as a novel small molecule inhibitor that prevents skeletal muscle atrophy.’

We have also revised the Abstract and Title of the manuscript to mention the discovery of BML-260 as a therapeutic small molecule for skeletal muscle atrophy. BML-260 has also been added as a key word for the manuscript.

1) **Comment:** “Independence of AKT phosphorylation was only shown in Fig. 3A Thus, more evidence is needed.”

Response and manuscript modification:

We are grateful for this advice related to the independence of AKT phosphorylation. To provide more experimental evidence that DUSP22 inhibition does not increase AKT phosphorylation and activation, the following Western blot results have been added to the revised manuscript as Supplementary Figure 7:

Figure 9: A-B) Western blot and densitometry analysis of AKT phosphorylation in in the Dex-treated tibialis anterior (TA) muscle 3d after delivery of control or DUSP22 siRNA. GAPDH was used for normalization of expression. $*=p<0.05$ indicate significantly increased or decreased.

2) **Comment:** “Silence or overexpression of DUSP22 showed the expected data. Because DUSP-JNK-FOXO pathway was already reported.”

Response and manuscript modification:

Thank you for this comment. In the revised manuscript, we have added the following text to further explain our rationale for demonstrating that DUSP22 influences FOXO levels via its effect on JNK activity:

‘The relationship between JNK activation and FOXO3a-mediated signaling has previous been reported. In addition, the effect of DUSP22 activation on JNK signaling is already known. The novel aspect of this current study is the demonstration that DUSP22 targeting can downregulate FOXO3a in skeletal muscle and produce beneficial effects on muscle

atrophy. As major signaling molecules in cells, both JNK and FOXO3a activity can be regulated by numerous factors (reviewed in [26, 27]), the results herein demonstrate that targeting DUSP22 alone in skeletal muscle is sufficient to downregulate FOXO3a and produce therapeutically desirable effects in myofibers.’

The text has been inserted into the revised Discussion section.

3) **Comment:** “The upregulation of DUSP22 was now observed in dex-treated mice in Fig. 1C. do you know why?”

Response and manuscript modification:

We are grateful for this comment. In the revised manuscript, we discuss the effect of dex on DUSP22 upregulation and the possible underlying mechanisms using following text:

“The results in this study show that DUSP22 expression is upregulated in Dex-treated mice. It is known that a number of DUSPs are up-regulated by glucocorticoids [28]. This regulatory mechanism has mainly been studied for DUSP1, but it is possible that similar mechanisms govern DUSP22 expression. Moreover, the upstream DNA enhancer sequence of the DUSP22 gene contains binding sites for the transcription factor Krueppel-like factor 9 (KLF9) [29]. Glucocorticoids have been shown to induce KLF9 expression in human epithelial cells [30]. It may be possible that muscle cells also upregulate KLF9 expression in response to Dex treatment and KLF9 binds to the DUSP22 upstream enhancer to induce gene expression. Further experiments would be needed to validate this hypothesis, which may be a rewarding area for future investigation.”

Inserted into the revised Discussion section.

4) **Comment:** “In Fig. 5A and 7A, the effect of siDUSP22 injection on muscle strength or physical performance should be suggested.”

Response and manuscript modification:

Thank you very much for this comment concerning muscle performance. For the manuscript revision, we have repeated the siDUSP22 injection experiment and tested the effect on muscle strength using an 820MS muscle strip system. It was observed that siDUSP22 knockdown increased the contraction force:

Figure 10: A) Tetanic muscle contraction measurement in the TA muscle of aged or middle aged mice 3d after the delivery of control or DUSP22 siRNA. B) Twitch force measure in 15 months-old mice 3d after the delivery of control or DUSP22 siRNA (n=4).

This data has been added to the revised manuscript as Supplementary Figure 8.

5) **Comment:** “In Fig. 1B, expression of Atrogin-1 is needed.”

Response and manuscript modification:

Thank you for this suggestion about Fig. 1B. In the revised manuscript, we have analyzed Atrogin-1 expression and found that it was also upregulated:

Figure 11: Atrogin-1 expression in C2C12 murine myotubes treated with vehicle or dexamethasone (Dex) to induce atrophy. Expression was measured using RNA Seq. TPM=transcript per million.

This data has been added to the revised manuscript as Supplementary Figure 1.

6) **Comment:** “In Fig. 9, measurement of myotube diameter is performed in MHC-IF stained cells for clarity.”

Response and manuscript modification:

We are grateful for this comment. For the manuscript revision, we have repeated the DUSP22 knockdown and BML-260 treatment experiments in human primary myotubes, with MHC-IF being utilized for measuring myotube diameter. The following data was obtained using this staining technique:

Figure 12: A) Micrographs of MYH2-immunostained human myotubes treated as follows: 1) Vehicle alone, 2) 10 μ M Dex for 24 h, 3) 10 μ M Dex and 12.5 μ M BML-260 for 24 h. G) Mean myotube diameter. B) Myotube diameter distribution. C) Micrographs of H MYH2-immunostained human myotubes treated as follows: (1) 48 h incubation in with DM plus control, scrambled siRNA; (2) 48 h incubation with DM plus DUSP22 siRNA; (3) 24 h incubation with DM plus control, scrambled siRNA and additional 24 h treatment with 10 μ M Dex plus siRNA; (4) 24 h incubation with DM plus DUSP22 siRNA and additional 24 h treatment with 10 μ M Dex plus siRNA. D) Myotube diameter. $*=p<0.05$, $***=p<0.001$, and $****=p<0.0001$ indicate significantly increased or decreased.

These results have been inserted into the revised manuscript as Supplementary Figure 9G-J.

We wish to express our gratitude to Reviewer 3, whose insightful comments have enhanced the quality of our manuscript.

References for the response to reviewers' comments

1. Shih, Y.C., et al., *The phosphatase DUSP22 inhibits UBR2-mediated K63-ubiquitination and activation of Lck downstream of TCR signalling*. Nat Commun, 2024. **15**(1): p. 532.
2. Villa, E., et al., *The E3 ligase UBR2 regulates cell death under caspase deficiency via Erk/MAPK pathway*. Cell Death Dis, 2020. **11**(12): p. 1041.
3. Ju, A., et al., *Scaffold Role of DUSP22 in ASK1-MKK7-JNK Signaling Pathway*. PLoS One, 2016. **11**(10): p. e0164259.
4. Hockerman, G.H., et al., *The Ubr2 Gene is Expressed in Skeletal Muscle Atrophying as a Result of Hind Limb Suspension, but not Merg1a Expression Alone*. Eur J Transl Myol, 2014. **24**(3): p. 3319.
5. Judge, S.M., et al., *Genome-wide identification of FoxO-dependent gene networks in skeletal muscle during C26 cancer cachexia*. BMC Cancer, 2014. **14**: p. 997.
6. Gao, S., et al., *UBR2 targets myosin heavy chain IIb and IIx for degradation: Molecular mechanism essential for cancer-induced muscle wasting*. Proc Natl Acad Sci U S A, 2022. **119**(43): p. e2200215119.
7. Kwak, K.S., et al., *Regulation of protein catabolism by muscle-specific and cytokine-inducible ubiquitin ligase E3alpha-II during cancer cachexia*. Cancer Res, 2004. **64**(22): p. 8193-8.
8. Cutshall, N.S., C. O'Day, and M. Prezhdo, *Rhodanine derivatives as inhibitors of JSP-1*. Bioorg Med Chem Lett, 2005. **15**(14): p. 3374-9.

9. Feng, Z., et al., *Identification of a rhodanine derivative BML-260 as a potent stimulator of UCP1 expression*. *Theranostics*, 2019. **9**(12): p. 3501-3514.
10. Harper, C., V. Gopalan, and J. Goh, *Exercise rescues mitochondrial coupling in aged skeletal muscle: a comparison of different modalities in preventing sarcopenia*. *J Transl Med*, 2021. **19**(1): p. 71.
11. Brandao, C.F.C., et al., *Physical training, UCP1 expression, mitochondrial density, and coupling in adipose tissue from women with obesity*. *Scand J Med Sci Sports*, 2019. **29**(11): p. 1699-1706.
12. Tomasic, T. and L. Peterlin Masic, *Rhodanine as a scaffold in drug discovery: a critical review of its biological activities and mechanisms of target modulation*. *Expert Opin Drug Discov*, 2012. **7**(7): p. 549-60.
13. Ramirez, M.A. and N.L. Borja, *Epalrestat: an aldose reductase inhibitor for the treatment of diabetic neuropathy*. *Pharmacotherapy*, 2008. **28**(5): p. 646-55.
14. Montano, E.T., et al., *Bacterial Cytological Profiling Identifies Rhodanine-Containing PAINS Analogs as Specific Inhibitors of Escherichia coli Thymidylate Kinase In Vivo*. *J Bacteriol*, 2021. **203**(19): p. e0010521.
15. Chaurasya, A., et al., *Rhodanine derivatives: An insight into the synthetic and medicinal perspectives as antimicrobial and antiviral agents*. *Chem Biol Drug Des*, 2022.
16. Jones, A.M., *2.05 - Privileged Structures and Motifs (Synthetic and Natural Scaffolds)*, in *Comprehensive Medicinal Chemistry III*, S. Chackalamannil, D. Rotella, and S.E. Ward, Editors. 2017, Elsevier: Oxford. p. 116-152.
17. Mendgen, T., C. Steuer, and C.D. Klein, *Privileged scaffolds or promiscuous binders: a comparative study on rhodanines and related heterocycles in medicinal chemistry*. *J Med Chem*, 2012. **55**(2): p. 743-53.
18. Huang, C.Y. and T.H. Tan, *DUSPs, to MAP kinases and beyond*. *Cell Biosci*, 2012. **2**(1): p. 24.
19. Bennett, B.L., et al., *SP600125, an anthrapyrazolone inhibitor of Jun N-terminal kinase*. *Proc Natl Acad Sci U S A*, 2001. **98**(24): p. 13681-6.
20. Chen, M.H., et al., *Dual-specificity phosphatases 22-deficient T cells contribute to the pathogenesis of ankylosing spondylitis*. *BMC Med*, 2023. **21**(1): p. 46.
21. Li, J.-P., et al., *The phosphatase JKAP/DUSP22 inhibits T-cell receptor signalling and autoimmunity by inactivating Lck*. *Nature Communications*, 2014. **5**(1): p. 3618.
22. Lin, Z. and Y. Will, *Evaluation of drugs with specific organ toxicities in organ-specific cell lines*. *Toxicol Sci*, 2012. **126**(1): p. 114-27.
23. Akram, D., I.A. Elhaty, and S.S. AlNeyadi, *Synthesis and Antibacterial Activity of Rhodanine-Based Azo Dyes and Their Use as Spectrophotometric Chemosensor for Fe³⁺ Ions*. *Chemosensors*, 2020. **8**(1): p. 16.
24. Langner, C. and H. Denk, *Wilson disease*. *Virchows Arch*, 2004. **445**(2): p. 111-8.
25. Alves, F.M., et al., *Iron accumulation in skeletal muscles of old mice is associated with impaired regeneration after ischaemia-reperfusion damage*. *J Cachexia Sarcopenia Muscle*, 2021. **12**(2): p. 476-492.
26. Zeke, A., et al., *JNK Signaling: Regulation and Functions Based on Complex Protein-Protein Partnerships*. *Microbiol Mol Biol Rev*, 2016. **80**(3): p. 793-835.

Reference: EMM-2024-19782

“Targeting phosphatase DUSP22 ameliorates skeletal muscle wasting via Akt independent JNK-FOXO3a repression”

By Lee *et al.*

27. Nho, R.S. and P. Hergert, *FoxO3a and disease progression*. World J Biol Chem, 2014. **5**(3): p. 346-54.
28. Clark, A.R., J.R. Martins, and C.R. Tchen, *Role of dual specificity phosphatases in biological responses to glucocorticoids*. J Biol Chem, 2008. **283**(38): p. 25765-9.
29. *DUSP22 Gene - Dual Specificity Phosphatase 22*. 10/20/2024]; Available from: <https://www.genecards.org/cgi-bin/carddisp.pl?gene=DUSP22>.
30. Mostafa, M.M., et al., *Genomic determinants implicated in the glucocorticoid-mediated induction of KLF9 in pulmonary epithelial cells*. Journal of Biological Chemistry, 2021. **296**.

25th Nov 2024

Dear Prof. Williams,

Thank you for the submission of your revised manuscript to EMBO Molecular Medicine.

We have now received the reports from the referees, and as you will see below, while referees #1 and #3 are satisfied with the revisions, referee #2 still raises a major concern that the inhibitor is exerting effects that are independent of muscle inhibition of DUSP22. I further cross-commented with the other referees, and referee #1 added:

"The concern on DUSP22 inhibitor is valid. One way to address the concern is to use skeletal muscle-specific DUSP22 conditional knockout mice in dex-induced muscle wasting."

After discussion within the team, we agreed that this was an essential point and would need to be addressed for further consideration of the manuscript in EMM. However, we would not necessarily request in vivo experiments if you provide alternative convincing results to address this concern.

As EMBO Press usually encourages one single round of revisions, please be aware that this will be the last chance for you to address the referees' concerns. The revised manuscript will once again be subjected to review, and we cannot guarantee a positive outcome at this stage.

Moreover, please address the following editorial requests:

1/ Manuscript text:

- We can accommodate a maximum of 5 keywords, please adjust accordingly.
- Please remove "data not shown" (p.13). As per our guidelines, the journal does not permit citation of "Data not shown". All data referred to in the paper should be displayed in the main or Expanded View figures.
- Acknowledgements: please note that funding information provided in the manuscript should match the information provided in the submission system (currently, Korean government (MSIP) through the Institute for Information and Communications Technology Promotion (IITP) grant (No. 2019-0-00567, Development of Intelligent SW systems for uncovering genetic variation and developing personalized medicine for cancer patients with unknown molecular genetic mechanisms) and "GIST Research Institute(GRI) IIBR" grant funded by the GIST are missing in the submission system).
- Author contributions: CRediT has replaced the traditional author contributions section because it offers a systematic machine readable author contributions format that allows for more effective research assessment. Please remove the Authors Contributions from the manuscript and use the free text boxes beneath each contributing author's name in our system to add specific details on the author's contribution. More information is available in our guide to authors.
- Please rename the conflict of interests: "Disclosure Statement and Competing Interests".
- The main figure legends should be placed after the References, at the very end of the manuscript file.
- BioRender should be acknowledged at the end of the Methods section in the following way:

Graphics:

(some of the... OR Figure #... OR synopsis) Graphics were created with BioRender.com.

2/ Reagents and Tools Table: Please download and fill our Reagents and Tools Table template (.docx), which you can find in our author guidelines: <https://www.embopress.org/page/journal/14693178/authorguide#structuredmethods>.

3/ Figures and Appendix:

- The main and EV figures (if any) should be provided as production quality individual Figure files (.eps, .tif, .jpg).
- We note that you have 13 suppl. Figures and 4 suppl. Tables: these suppl. figures and tables can be provided in a single PDF with a title page and page numbers for each item; the nomenclature should be Appendix Figure S1, etc. Appendix Table S1, etc.; We can also accommodate 5-6 Expandable View figures that would need to be provided as separate files.
- Figure legends for suppl. figures need to be removed from the manuscript and should be provided right after each Appendix figure; if you wish to have EV figures, then the legends can stay in the manuscript, but the nomenclature should be Figure EV1, etc; all callouts in the manuscript need to be updated accordingly.
- Please make sure that all figures/figure panels are referenced in the manuscript text. Currently, callouts are missing for Figure 1G, Suppl. Tables 1-4.
- Please address the queries from the copy editors:
 1. Please note that the box plots need to be defined in terms of minima, maxima, centre, bounds of box and whiskers, and percentile in the legends of figures 1b; 8d-e.
 2. Please note that information related to n is missing in the legends of figures 1a, f-k; 2e, g-h; 3b-d, j; 4d-e; 5e-f, k-l, n; 6d, f, h;

7c, f, h, j, l, p-q; 8b, d-e; 9b, i, k, m.

3. Although 'n' is provided, please describe the nature of entity for 'n' in the legends of figures 1b, e, l; 2c; 3f-h; 4c, f, h-j; 5c; 6b-c, j; 7d, k, m-n; 9a, e-g.

5. Please note that the error bars are not defined in the legends of figures 1a, c-d, f-l; 2c, e, g-h; 3b-d, f-h, j; 4c, e-f, h-i, k-m; 5c-d, f, k-l, n; 6b-d, f, h, j; 7c, f, j-n, p-q; 9a-b, e-g, i, k, m-n.

4/ Thank you for providing Source Data. We note that blots and microscopy images are missing for Figures 3A, 3E, 3I, 4B, 4G, 4J, 5B, 5E, 5M, 5P, 6I, 7B, 7G.

5/ Checklist: please also fill the top left corner (corr. author's name, journal and ms ID#).

6/ Please note that all corresponding authors are required to supply an ORCID ID for their name upon submission of a revised manuscript. An ORCID ID is currently missing Da-Woon Jung.

7/ The paper explained: EMBO Molecular Medicine articles are accompanied by a summary of the articles to emphasize the major findings in the paper and their medical implications for the non-specialist reader. Please provide a draft summary of your article highlighting

8/ As part of the EMBO Publications transparent editorial process initiative (see our Editorial at <http://embomolmed.embopress.org/content/2/9/329>), EMBO Molecular Medicine will publish online a Review Process File (RPF) to accompany accepted manuscripts.

In the event of acceptance, this file will be published in conjunction with your paper and will include the anonymous referee reports, your point-by-point response and all pertinent correspondence relating to the manuscript. Let us know whether you agree with the publication of the RPF and as here, if you want to remove or not any figures from it prior to publication. Please note that the Authors checklist will be published at the end of the RPF.

I look forward to receiving your revised manuscript.

Yours sincerely,

Lise Roth

***** Reviewer's comments *****

Referee #1 (Remarks for Author):

The authors have adequately addressed my previous concerns on the manuscript.

Referee #2 (Comments on Novelty/Model System for Author):

I rated technical quality high as overall the studies appear to be carried out well from a technical perspective, especially with the improvements in this revision (e.g. using 2 siRNA oligos). I rated the novelty as only medium because of my serious concerns about whether BML-260 is indeed acting via inhibition of DUSP22 (see remarks). Removing these data would decrease the novelty significantly. I rated medical impact as low as this impact was largely based on the potential of BML-260 or future inhibitors of DUSP22 to be therapies for muscle wasting. The model systems (e.g. dex induced muscle wasting in vivo and in myotubes) were appropriate, but limited mainly to dexamethasone induced atrophy.

Referee #2 (Remarks for Author):

A major concern that I had in the original review was the near complete lack of information on the small molecule inhibitor BML260. The authors have provided much more information and provided data in this revision. However, this information and data raises a significant concern that the inhibitor is exerting effects that are independent of muscle inhibition of DUSP22. Consider the following:

(1) The IC₅₀ of the inhibitor for DUSP22 has been reported as 18 μ M in the original description of the inhibitor (ref 31) and in new supplementary fig 5A as 54 μ M. These are measured in vitro in cell free assays.

(2) In supplementary figure 9, plasma concentrations after injection peak at 16 μ g/ml, decrease with a half life of ~60 minutes and are undetectable by ~ 8 hours. A concentration of 10 μ g/ml is only 29 μ M (MW 341) suggesting that for most (>23 hrs/day) of the time after injection, the inhibitor is subtherapeutic in the plasma.

(3) Muscle levels after injection peak at ~ 1 hour at ~0.35 μ g/g tissue and are undetectable somewhere between 6 and 12 hours. If one crudely estimates 1 g tissue = 1 ml, the half maximal inhibitor concentration in muscle would be ~0.5 μ M, far below the IC₅₀. The BML-260 is unlikely to be inhibiting DUSP22 in muscle at any time.

(4) The inhibitor was only given once per day, so unlikely to accumulate over time. Therefore, there is no evidence that muscle levels are ever inhibitory.

(5) The inhibitor is competitive (ref 31) and so the possibility of permanent inhibition following brief exposure is eliminated.

(6) In cell culture studies, BML-260 was added to media at 12.5 μ M. Since the intracellular concentration is likely lower, it is unlikely that the effects seen are due to inhibition of DUSP22.

Although I have no reason to dispute the data on beneficial effects of BML-260 on muscle atrophy, the effects appear unlikely to be due to inhibition of DUSP22.

A minor concern with the revision is that the conclusion is now too long.

Otherwise, the authors have responded adequately to my other criticisms.

Referee #3 (Remarks for Author):

Suitable for publication.

Reference: EMM-2024-19782-V2

“Targeting phosphatase DUSP22 with BML-260 ameliorates skeletal muscle wasting via Akt independent JNK-FOXO3a repression”

By Lee *et al.*

Reviewer 2:

1) Comment: “A major concern that I had in the original review was the near complete lack of information on the small molecule inhibitor BML260. The authors have provided much more information and provided data in this revision. However, this information and data raises a significant concern that the inhibitor is exerting effects that are independent of muscle inhibition of DUSP22. Consider the following:

(1) the IC50 of the inhibitor for DUSP22 has been reported as 18 uM in the original description of the inhibitor (ref 31) and in new supplementary fig 5A as 54 uM. These are measured in vitro in cell free assays.

(2) In supplementary figure 9, plasma concentrations after injection peak at 16 ug/ml, decrease with a half life of ~60 minutes and are undetectable by ~ 8 hours. A concentration of 10 ug/ml is only 29 uM (MW 341) suggesting that for most (>23 hrs/day) of the time after injection, the inhibitor is subtherapeutic in the plasma.

(3) Muscle levels after injection peak at ~ 1 hour at ~0.35 ug/g tissue and are undetectable somewhere between 6 and 12 hours. If one crudely estimates 1 g tissue = 1 ml, the half maximal inhibitor concentration in muscle would be ~0.5 uM, far below the IC50. The BML-260 is unlikely to be inhibiting DUSP22 in muscle at any time.

(4) The inhibitor was only given once per day, so unlikely to accumulate over time. Therefore, there is no evidence that muscle levels are ever inhibitory.

(5) The inhibitor is competitive (ref 31) and so the possibility of permanent inhibition following brief exposure is eliminated.

(6) In cell culture studies, BML-260 was added to media at 12.5 uM. Since the intracellular concentration is likely lower, it is unlikely that the effects seen are due to inhibition of DUSP22.

Although I have no reason to dispute the data on beneficial effects of BML-260 on muscle atrophy, the effects appear unlikely to be due to inhibition of DUSP22.

Response and manuscript modification:

We thank Reviewer 2 for their comment about the relationship between BML-260 about DUSP22 in the context of skeletal muscle atrophy. It is known that DUSP22 functions as an activator of JNK, which promotes muscle atrophy. In Appendix Figure S6, the effect of small molecule SP600125 (a JNK inhibitor) on myotube atrophy was compared with BML-260. It was observed that BML-260 had no additive effect on myotube atrophy in the presence of SP600125 (also shown in Figure 1 below). This result indicates that BML-260 prevents myotube atrophy via inhibition of the DUSP22 target.

Figure 1 (reproduced from Appendix Figure S6): A) Fast myosin (MYH2) immunostaining of C2C12 myoblasts cultured as follows: (1) DM for 120 h (vehicle alone); (2) DM for 96 h and DM plus 10 µM Dex for 24 h; (3) DM for 96 h and DM plus 10 µM Dex and 12.5 µM BML-260 for 24 h; (4) DM for 96 h and DM plus 10 µM Dex and 20 µM SP600125 for 24 h; (5) DM for 96 h and DM plus 10 µM Dex and 12.5 µM BML-260 plus 20 µM SP600125 for 24 h. (n=6) B) Mean myotube diameter. ****= $p < 0.0001$ indicate significantly increased or decreased. n represents biological replicates. Error bars represent the standard error of the mean (SEM).

The gene knockdown data presented in this study shows that reducing DUSP22 expression prevents skeletal muscle atrophy. In Figure 5N of the manuscript, BML-260 treatment was observed to significantly reduce DUSP22 protein levels in the dexamethasone (Dex) model of skeletal muscle atrophy (reproduced as Figure 2 below).

Figure 2 (reproduced from Figure 6I-J): Western blot analysis of DUSP22 expression in the gastrocnemius muscle. GAPDH was used for normalization of expression. *= $p < 0.05$ and **= $p < 0.01$ indicate significantly increased or decreased.

To further investigate that relationship between BML-260 treatment and DUSP22, we assessed DUSP22 levels in aged mouse skeletal muscle. BML-260 treatment was also found to reduce DUSP22 protein levels (Figure 3).

Figure 3: Western blot analysis of DUSP22 expression in the TA of aged mice. GAPDH was used for normalization of expression. *= $p < 0.05$ and **= $p < 0.01$ indicate significantly increased or decreased. n represents biological replicates. Error bars represent the standard error of the mean (SEM).

This figure has been added to the revised manuscript as Appendix Figure S13A.

In the immobilization model, it was shown that BML-260 treatment reduced DUSP22 protein levels, although this failed to reach statistical significance ($p = 0.54$; $n = 3$). We repeated the immobilization model with an increased n number for BML-260 treatment.

It was observed that BML-260 treatment significantly reduced DUSP22 levels in the tibialis anterior (TA) muscle (Figure 4). To validate that this effect was not only restricted to the TA muscle, DUSP22 protein levels were also assessed in the gastrocnemius muscle. It was found that BML-260 treatment also reduced DUSP22 levels in the gastrocnemius muscle (Figure 4).

Figure 4: Western blot analysis of DUSP22 expression in the TA (n=3,5,6) (A) and gastrocnemius muscle (n=3,5,6) (B) of immobilized (IMM) mice treated with BML-260. For the TA muscle, MYH4 (myosin heavy chain 2B) levels are also shown. GAPDH was used for normalization of expression. *= $p < 0.05$, **= $p < 0.01$, and ***= $p < 0.001$ indicate significantly increased or decreased. n represents biological replicates. Error bars represent the standard error of the mean (SEM).

This figure has been added to the revised manuscript as Appendix Figure S13B-C.

Accordingly, BML-260 treatment also decreased activity of the DUSP22 therapeutic target, JNK, in the immobilization model (Figure 5).

Figure 5: Western blot analysis of DUSP22, JNK and phosphorylated JNK (JNK-P) levels in the TA muscle of immobilized mice (n= 3,4). GAPDH was used for normalization of expression. *= $p < 0.05$, **= $p < 0.01$, and ***= $p < 0.001$ indicate significantly increased or decreased. n represents biological replicates. Error bars represent the standard error of the mean (SEM).

This figure has been added to the revised manuscript as Appendix Figure S14.

In conjunction with our results showing that DUSP22 gene knockdown prevents skeletal muscle wasting, the finding that BML-260 reduces DUSP22 levels in the aging,

Dex, and immobilization models supports the hypothesis that BML-260 can prevent muscle wasting by regulating DUSP22.

To further assess the relationship between BML-260 therapeutic activity and DUSP22 expression, we repeated the DUSP22 siRNA study in Dex-treated myotubes subsequently treated with BML-260. Compared to DUSP22 gene knockdown or BML-260 alone, BML-260 treatment in the presence of DUSP22 gene knockdown had no significant additional effect on MuRF-1 and atrogenin-1 expression (Figure 5), suggesting that the effect of BML-260 on atrogenes expression is dependent on DUSP22 expression.

Figure 6: qPCR analysis of A) atrogenin-1, B) MuRF-1, and C) DUSP22 expression in dexamethasone-treated C2C12 myotubes treated with BML-260 alone, DUSP22 siRNA, or DUSP22 siRNA plus BML-260 (n=5). *= $p < 0.05$, **= $p < 0.01$, ***= $p < 0.001$ and ****= $p < 0.0001$ indicate significantly increased or decreased. ns=not significant. n represents biological replicates. Error bars represent the standard error of the mean (SEM).

This figure has been added to the revised manuscript as Appendix Figure S15.

Taken together, the above results, along with the known binding interaction between BML-260 and DUSP22, can support a role for DUSP22 regulation in the anti-muscle atrophy effects of BML-260. To attempt to more accurately describe this relationship, we have replaced 'Targeting' DUSP22 with 'Modulating' DUSP22 in the title of the manuscript. Additionally, we have added the following text to the Discussion section to describe these effects of BML-260 treatment, as follows:

‘BML-260 was developed as an enzyme inhibitor of DUSP22 activity. In myotubes undergoing atrophy, BML-260 had no additional effect on atrogene downregulation compared to DUSP22 gene knockdown, suggesting that BML-260 activity is dependent on the presence of DUSP22. In addition, the observation of reduced levels of DUSP22 after BML-260 treatment in the skeletal muscle of Dex-treated, aged, and immobilized mice indicates that BML-260 also suppresses DUSP22 expression. The effect on DUSP22 levels, rather than reduced enzyme activity, may also be responsible for the therapeutic effects of BML-260 in muscle atrophy. This aspect of BML-260 bioactivity could be an interesting area for future investigation in other disease contexts linked to DUSP22 and JNK suppression.’ (shown using red font).

2) **Comment:** “A minor concern with the revision is that the conclusion is now too long.”

Response and manuscript modification: Thank you for this feedback about the manuscript conclusion. The following passages have been removed, because we believe that they are not essential for the Discussion section: 1) Previous studies linking DUSP4 expression to denervation-induced muscle atrophy, 2) DUSP5 and DUSP6 expression patterns in acute exercise, and 3) The text related to assessing other E3 ligases, such as, MUSA1 and SMART.

We wish to express our gratitude to Reviewer 2, whose expert comments have further increased the quality of our manuscript.

17th Mar 2025

Dear Prof. Williams,

Thank you for submitting your revised study, which was re-reviewed by referee #2. As you will see below, this referee acknowledges the work that was done, but remains unconvinced that BML-260 acts as a direct DUSP22 inhibitor. Taking these comments into consideration, we further discussed your manuscript within the team, and would like to invite minor revisions of the manuscript. Please note that we do NOT ask for additional experiments, but rather for adequate discussion in the manuscript (as suggested by the referee) and toning down the claim that BML-260 is an inhibitor of DUSP22 (including in the abstract and synopsis).

Please also address the following editorial minor issues:

1/ Manuscript text:

- Please remove the red font and only keep in track changes mode any new modification.
- "Materials and Methods" should be renamed "Methods":
 - o Cells: please provide a statement on mycoplasma contamination and authentication.
 - o Mouse models: please provide the origin of the mice, as well as housing and husbandry conditions.
 - o Statistics: please provide a statement on randomization, blinding, sample size and inclusion/exclusion criteria.
- Data availability: Please note that only datasets that were produced in this study should be listed here (not previously published datasets). Please note that new RNAseq datasets must be deposited in a public repository.
- Disclosure statement and competing interests: please note that it is understood that by publishing a paper in this journal, the authors agree to make available to colleagues in academic research all new reagents, including organisms (or means to produce them), viruses, cells, nucleic acids and antibodies, that were used in the research reported and that are not available from public repositories or commercial suppliers.

2/ Figures and Appendix:

- Exact p values should be provided in the figures or their legends.
- As mentioned previously, we would strongly encourage you to make some of your appendix figures EV figures, that are collapsible/expandable online. EV Figures should be cited as 'Figure EV1, Figure EV2' etc... in the text and their respective legends should be included in the main text after the legends of regular figures.

3/ Checklist:

- please fill the subsection cell authentication/mycoplasma contamination.
- please check the section "Animals observed in or captured from the field", as I do not think it applies to your study.
- please check the section study protocol, as I am not sure both sections apply to your study.
- please fill in the subsection about blinding (statistics).
- please check the section on data availability/human clinical and genomic datasets, as well as computational models, as I am not sure it applies to your study

4/ Synopsis:

Thank you for providing a graphical abstract. I have cropped a small portion to serve as thumbnail on our electronic table of content (attached, 115 x 70 pixels), please let me know if you agree or provide an alternative image in the right dimensions.

We note that you agree with the publication of Review Process File (RPF) and that you do not wish to exclude any figure from the point-by-point rebuttal letters.

I look forward to receiving your revised manuscript.

Yours sincerely,

Lise Roth

***** Reviewer's comments *****

Referee #2 (Comments on Novelty/Model System for Author):

The manuscript is a large body of work that overall appears to be well carried out. As I stated in my previous review, the novelty is to a large degree due to the studies with the BLM-260 inhibitor which offered translational potential. Due to my significant doubts that BLM-260 is exerting its actions as an inhibitor of DUSP 22, I consider the novelty only medium and the medical impact low. BLM-260 appears to be an early stage rhodamine based compound which at this stage of development is probably a 'dirty' inhibitor with many off target effects. Since translational potential appears to be important for EMBO Mol Medicine, I don't think it meets the bar despite the authors' hard work.

Referee #2 (Remarks for Author):

The authors have provided additional results to try to bolster their argument that BML-260 is a pharmacological inhibitor of DUSP22. The observations that BML-260 did not exert any additional protective effect on dexamethasone induced myotube atrophy in vitro beyond that seen with a JNK inhibitor is consistent with BML-260 acting on DUSP22 but not proof that it does as there are many upstream regulators of JNK. The observations that BML-260 administration in vivo decreases levels of DUSP22 protein in muscle indicates that BML-260 exerts some effect on DUSP22, but the mechanism remains unclear. It is unusual, but not unheard of, for an enzyme inhibitor to also trigger a protein's degradation. These data, taken together with the pharmacokinetics of the compound, the authors cannot conclude that BML-260 is functioning as a pharmacological inhibitor. A number of possible explanations remain. BML-260 seems to be an early stage inhibitor with no evidence of significant medicinal chemistry optimization. So evidently, it remains possible that it has off target pleotropic effects that alter multiple targets/pathways leading to its muscle protective effects. Some of these effects may be in the same pathway as DUSP22 and JNK and so generate the epistatic results observed. In addition, does DUSP2 exist in a complex in cells and in vivo such that in the complex, there is much higher affinity binding to BML-260 than with the enzyme alone in an in vitro assay?

I suggest that the authors include the pharmacokinetic data of BML-260 as supplementary data and discuss the challenge raised by these results and the possible explanations. Although the authors may prefer the conclusion that BML-260 is functioning as an inhibitor of DUSP2 enzyme activity, they should also acknowledge that off target effects may also be contributing.

In view of the overall uncertainty of the mechanism of action of BML-260, I am personally dubious as to whether the manuscript is suitable for EMBO Mol Medicine

The authors addressed the remaining editorial issues.

26th Mar 2025

Dear Prof. Williams,

Thank you for submitting your revised files. I am pleased to inform you that your manuscript is accepted for publication and is now being sent to our publisher to be included in the next available issue of EMBO Molecular Medicine.

If you have any questions, please do not hesitate to contact the Editorial Office.

Thank you for your contribution to EMBO Molecular Medicine!

Yours sincerely,

Lise Roth
